# 3D bioprinted multilayered cerebrovascular conduits to study cancer extravasation mechanism related with vascular geometry

Wonbin Park [1], Jae-Seong Lee[2], Ge Gao [3], Byoung Soo Kim [2,4] ✉ &
Dong-Woo Cho [1] ✉

Cerebral vessels are composed of highly complex structures that facilitate blood perfusion necessary for meeting the high energy demands of the brain. Their geometrical complexities alter the biophysical behavior of circulating tumor cells in the brain, thereby influencing brain metastasis. However, recapitulation of the native cerebrovascular microenvironment that shows continuities between vascular geometry and metastatic cancer development has not been accomplished. Here, we apply an in-bath 3D triaxial bioprinting technique and a brain-specific hybrid bioink containing an ionically crosslinkable hydrogel to generate a mature three-layered cerebrovascular conduit with varying curvatures to investigate the physical and molecular mechanisms of cancer extravasation in vitro. We show that more tumor cells adhere at larger vascular curvature regions, suggesting that prolongation of tumor residence time under low velocity and wall shear stress accelerates the molecular signatures of metastatic potential, including endothelial barrier disruption, epithelial–mesenchymal transition, inflammatory response, and tumorigenesis. These findings provide insights into the underlying mechanisms driving brain metastases and facilitate future advances in pharmaceutical and medical research.

Brain blood vessels are composed of specialized structures that facilitate the cerebral perfusion necessary for the high energy demands of neuronal activity[1]. The native brain, which constitutes only approximately 2% of the body's weight, receives approximately 15–20% of the total blood supply in the body. This substantial blood flow allows the delivery of approximately 49 mL of oxygen per minute, utilizing 750 mL of blood per minute[2,3]. The vasculature originating from the heart grooves on the cortex surface, extends deep into the brain, and forms complex networks. The cerebral vasculature should retain high molecular selectivity to regulate homeostasis of the brain microenvironment. By integrating with supportive cells, such as pericytes, glial cells, and neuronal cells, brain endothelial cells build a highly organized barrier that restricts crossing of pathogens[4]. However,

despite the presence of the physicochemical barrier, metastases are predominantly observed in certain regions of the brain, such as the vascular border zone or gray–white matter boundary, where the barrier begins to bend and forms coiled, looped, and spiral structures[5–8]. Decades of research have shown that such significant tortuosity can govern the biophysical behavior of circulating tumor cells by altering factors that control the hemodynamic distribution, including blood flow velocity and wall shear stress (WSS)[9,10]. Meanwhile, according to the seed and soil hypothesis, the intercellular communication with resident brain cells shapes the brain tropism of metastatic tumor cells[11,12]. Therefore, researchers speculate that the biomechanical, biochemical, and cellular interplay between cerebral vasculature and tumor cells contribute to the progression of brain metastases.

[1]Department of Mechanical Engineering, Pohang University of Science and Technology, Pohang, Republic of Korea. [2]School of Biomedical Convergence Engineering, Pusan National University, Yangsan, Republic of Korea. [3]School of Medical Technology, Beijing Institute of Technology, Beijing, China. [4]Medical Research Institute, Pusan National University, Yangsan, Republic of Korea. ✉e-mail: bskim7@pusan.ac.kr; dwcho@postech.ac.kr

Numerous in vitro models have been developed to understand the mechanisms underlying brain metastases. For example, the Transwell system-based coculture models have provided insights into intercellular communications in the survival and growth of tumor cells in the brain and could potentially provide a reproducible high-throughput screening method[13–15]. However, these models fail to recapitulate rate-regulating events, including the circulation, adhesion, and extravasation of tumor cells in the three-dimensional (3D) microenvironment[16]. To overcome this limitation, 3D microfluidic devices have recently emerged as an alternative approach for recreating such metastatic events in the vascular tissue in vitro[17–20]. Furthermore, by incorporating 3D printing and computing technologies, structurally various brain vasculatures have been modeled and analyzed to verify their effects on tumor dissemination[21–26]. Nonetheless, most of the reported tissue engineering strategies still rely on quite complicated processes, such as scaffold fabrication, cell seeding, and perfusion-tube connection, thus reducing the manufacturing yield

of devices. Moreover, only a few attempts have been made to engineer multicellular brain blood vessels with various cerebrovascular curvatures. Consequently, advances in understanding the key biomechanical driving factors in brain metastases have been few and the development of preclinical research in cerebrovascular disease has been limited.

In this study, we propose a direct bioprinting strategy to construct multilayered cerebrovascular conduits (MCCs) with geometrically varying curvatures for brain metastasis studies (Fig. 1). The MCCs were developed via an in-bath 3D triaxial bioprinting technique employing an ionically cross-linkable brain-specific bioink laden with multiple cell sources, including brain endothelial cells (BECs), brain pericytes (BPCs), and neural progenitor cells (NPCs), and a sacrificial material (calcium-added PF-127; CPF-127) to investigate the effects of vessel curves and cellular interactions in circulating tumor cells (CTCs) dissemination. Modeling a series of metastatic cascades from tumor adhesion and extravasation in vitro enabled multifaceted analysis of

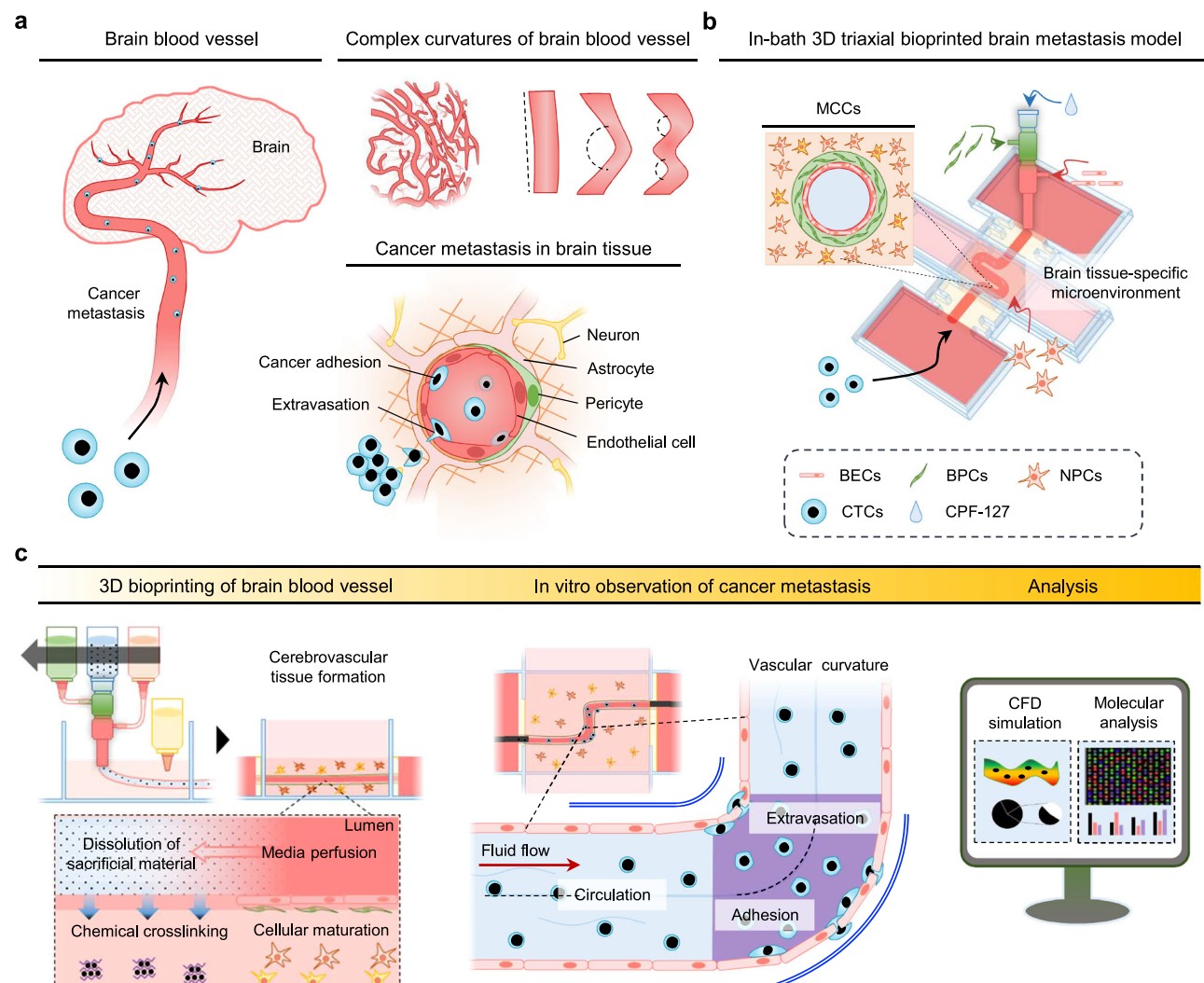

**Fig. 1 | Schematic illustration for development of in vitro brain metastasis model. a** Brain metastasis usually occurs through blood vessels connecting distant organs to brain. In the brain, circulating tumor cells (CTCs) disseminating through the complex vascular network are influenced by cerebrovascular geometry and other cellular components during metastatic progression. **b** To recapitulate the pathophysiological mechanisms in brain metastasis of CTCs in vitro, multilayered cerebrovascular conduits (MCCs) are constructed using in-bath 3D triaxial bioprinting technology. Multiple cell types, such as neural progenitor cells (NPCs), brain endothelial cells (BECs), and brain pericytes (BPCs), are utilized to

fabricate triple layered vascular structures with diverse curvatures. **c** Through rapid crosslinking between hybrid brain-derived decellularized extracellular matrix bioink and calcium ion from the core sacrificial material (calcium-added Pluronic F-127; CPF −127), the hollow cerebrovascular conduits are constructed with high structural stability. CTCs are introduced into a fully mature brain blood vessel, and metastatic pattern changes depending on hemodynamic variants and biological interactions are investigated. The model allows multifaceted investigation of the underlying mechanisms of brain metastasis, demonstrating the interplay of cerebrovascular curvatures and cellular interactions during tumor cell adhesion and extravasation.

the key molecular and hemodynamic drivers resulting from vascular geometrical features in brain metastasis events. The findings mark a technological breakthrough in studying the physical and molecular mechanisms of cancer extravasation for scientific and medical advances.

## Results

### Optimal formulation of hybrid BdECM bioink for in-bath 3D triaxial bioprinting

An appropriate bioink is a prerequisite for successful in-bath 3D triaxial bioprinting. To elaborate 3D constructs with high printability and structural stability, bioinks exhibiting suitable rheological characteristics should be used. Therefore, a hybrid brain-derived decellularized extracellular matrix (BdECM) bioink comprising BdECM and alginate was developed (Fig. 2a). Fundamentally, BdECM bioink was selected as an essential printing material for elaborating the brain-mimetic microenvironment. In the decellularized material, vital proteinous ingredients, including glycosaminoglycans (GAGs, $78.70 \pm 4.14\%$) and collagen ($2864.81 \pm 357.61\%$), were preserved without most cellular components presented with dsDNA ($13.24 \pm 2.62$ ng mg$^{-1}$) (Supplementary Fig. 1). In addition, an in-depth proteomics analysis of BdECM revealed more than 2,000 types of proteins, including basement membrane components, interstitial matrix proteins, and growth factors, implying that it has the potential to create a more biologically relevant microenvironment than other single materials (Supplementary Fig. 2). However, while the intracellular or signaling molecules were removed during decellularization process, the structural proteins residing at basement membrane (type IV collagens) and interstitial matrix (type I collagens) remained[27]. Therefore, the amounts of collagen in BdECM were significantly higher than that in native tissue. The increased portion of fibrillary collagens can provide load-bearing mechanical properties to the material[28]. Our previous studies have shown the superior functionalities of BdECM and in-bath bioprinting technology to construct reliable drug-testing platforms targeting cerebral and vascular diseases[29, 30]. To combine the two source technologies for fabricating a cerebrovascular tissue equivalent, the optimal condition of the hybrid BdECM bioink was established by comparing five different formulations with BdECM (B) and alginate (A) (0.5B, 0.5 wt.% BdECM; 1.0B, 1.0 wt.% BdECM; 1.5B, 1.5 wt.% BdECM; 1.0B0.5A, 1.0 wt.% BdECM mixed with 0.5 wt.% alginate; 1.0B1.0A, 1.0 wt.% BdECM mixed with 1.0 wt.% alginate).

In extrusion-based bioprinting, BdECM-based bioinks should exhibit a stable sol−gel transition capability depending on the viscosity, viscoelastic shear moduli, and elastic recovery. The results of a viscosity assay indicate shear-thinning flow of the pre-gel solutions: the increasing shear rates led to a decrease in the viscosity when the shear rate was in the range 0.1−1000 s$^{-1}$ (Fig. 2b). Furthermore, a higher concentration ratio of BdECM and alginate increased the viscosity of the bioinks, ensuring printability and structural stability. A complex modulus assay demonstrated that all groups, except for the 0.5B group, displayed clear yield and flow points indicating the transition from the solid plateau region to the fluid region, which was dependent on the increased strain at 15 °C (Fig. 2c, Supplementary Fig. 3a). Fluidic-dominant behavior was exhibited by 0.5B: the elastic modulus (G") was higher than the viscous modulus (G') even under 0.1% strain. Furthermore, the shear stress-dependent hydrogels showed rapid sol−gel transitions, confirming their potential as supportive bath materials and printing materials (Fig. 2d). In practice, the fluidic-dominant bioink could not sustain the printed hollow structure, and the tube collapsed instantly; by contrast, the solid-dominant bioinks provided a structurally supportive environment (Fig. 2e). However, when the conduits were printed with tortuosity, repetitive leakage was observed in the curved portion of the pure BdECM bath (Fig. 2f), whereas sleek curvatures could be created with high printing fidelity in the hybrid BdECM bath. Alginate in the bath material was immediately

crosslinked with calcium ions from calcium-added pluronic F-127 (CPF-127); this relieved the surface tension from the bias bioink deposit owing to the printhead velocity gap at the inner and outer walls. Chemical enhancement enabled the direct printing of structurally stable multilayered hollow conduits with highly curved structures.

Another critical requirement of the bioink is the ability to support cell growth. The capacity was evaluated by comparing the viability and proliferation rates of human neural progenitor cells (NPCs) for various formulations. The growth of NPCs was generally nurtured by day 7, indicating that the increase in BdECM concentration plays a biochemically supportive role in brain cells (Fig. 2g). However, an excessive BdECM concentration impeded biological activity, which was suggested by the cell proliferation rate and mechanical strength results for 1.5B (Supplementary Fig. 3b). Namely, a physically suitable environment should be ensured for the encapsulated cells to function. On the other hand, the cells in 1.0B1.0A showed the lowest metabolic activity, which might be due to the high concentration of alginate with no cell binding sites. The viability of NPCs and brain endothelial cells also decreased with increasing alginate concentrations (1.0B, $91.15 \pm 1.60\%$; 1.0B0.5 A, $84.28 \pm 1.12\%$; 1.0B1.0 A, $79.48 \pm 3.85\%$) (Fig. 2h, Supplementary Fig. 4). Additionally, under the assumption that the presence of alginate can affect the cell morphology, pure BdECM (1.0B) and hybrid BdECM (1.0B0.5 A) were compared[31]. Although the cells encapsulated in 1.0B0.5 A exhibited relatively thin and long shapes with many pseudopodia on day 7 compared with those in 1.0B, the morphological gap decreased as the cells proliferated (Supplementary Fig. 5). In addition, the cell-laden hybrid BdECM exhibited lower physical strength than native brain tissue (Supplementary Fig. 6). This may have resulted from the difference in cell density between the two systems (experimental condition, ~$2 \times 10^6$ cells mL$^{-1}$; native brain, >$5 \times 10^6$ cells mL$^{-1}$ [32]; weight/volume density of porcine brain, $1.05 \pm 0.02$ g mL$^{-1}$) (Supplementary Fig. 7). Thus, a printing material with a minimum alginate concentration that can support suitable printing fidelity and biological support was necessary. Overall, the 1.0B0.5 A hybrid BdECM bioink was selected as the main printing material in this study.

### Parametrical control to generate dimensionally tunable vascular structure

A triple-layered perfusable conduit was successfully constructed using the optimal hybrid BdECM bioink. To elaborate the dimensional controllability, a parametric study was conducted. Based on the assumption that the printhead velocity and programming commands govern the geometry, the diameter and curvature of the conduits were successfully adjusted (Fig. 3a). To establish a parametrically precise control process, the nozzle size and pneumatic pressure for tube printing were optimized in advance (Fig. 3b). Because minimizing the wall thickness of the innermost layer of the tube is important for generating the endothelial monolayer, the minimum pressure was applied to each of the three nozzles (13G/17G/22G) for the construction of dual-layered conduits: core nozzle, 500 kPa; middle nozzle, 35 kPa; outer nozzle, 35 kPa. Using this setup, the number of layers could be adjusted to produce single or dual layers without fracture.

In particular, as the printhead velocity increased from 300 mm min$^{-1}$ to 900 mm min$^{-1}$, the inner diameter and total wall thickness of the multilayered tubes decreased from $904.05 \pm 40.72$ μm to $490.44 \pm 33.08$ μm and from $219.81 \pm 18.42$ μm to $74.77 \pm 5.14$ μm, respectively (Fig. 3c, d). Importantly, when the printhead velocity was over 600 mm min$^{-1}$, the wall thickness of the innermost layer was less than the 50 μm needed to provide a supportive structural environment for endothelial monolayer formation[30]. Therefore, 600 mm min$^{-1}$ was selected as the optimal printhead velocity in this study. A curvilinear motion of the printhead, which was executed in response to G-code commands, generated vessel curvatures with a spatial frequency of 200−600 cycles m$^{-1}$ (Fig. 3e). Consequently, straight, narrowed,

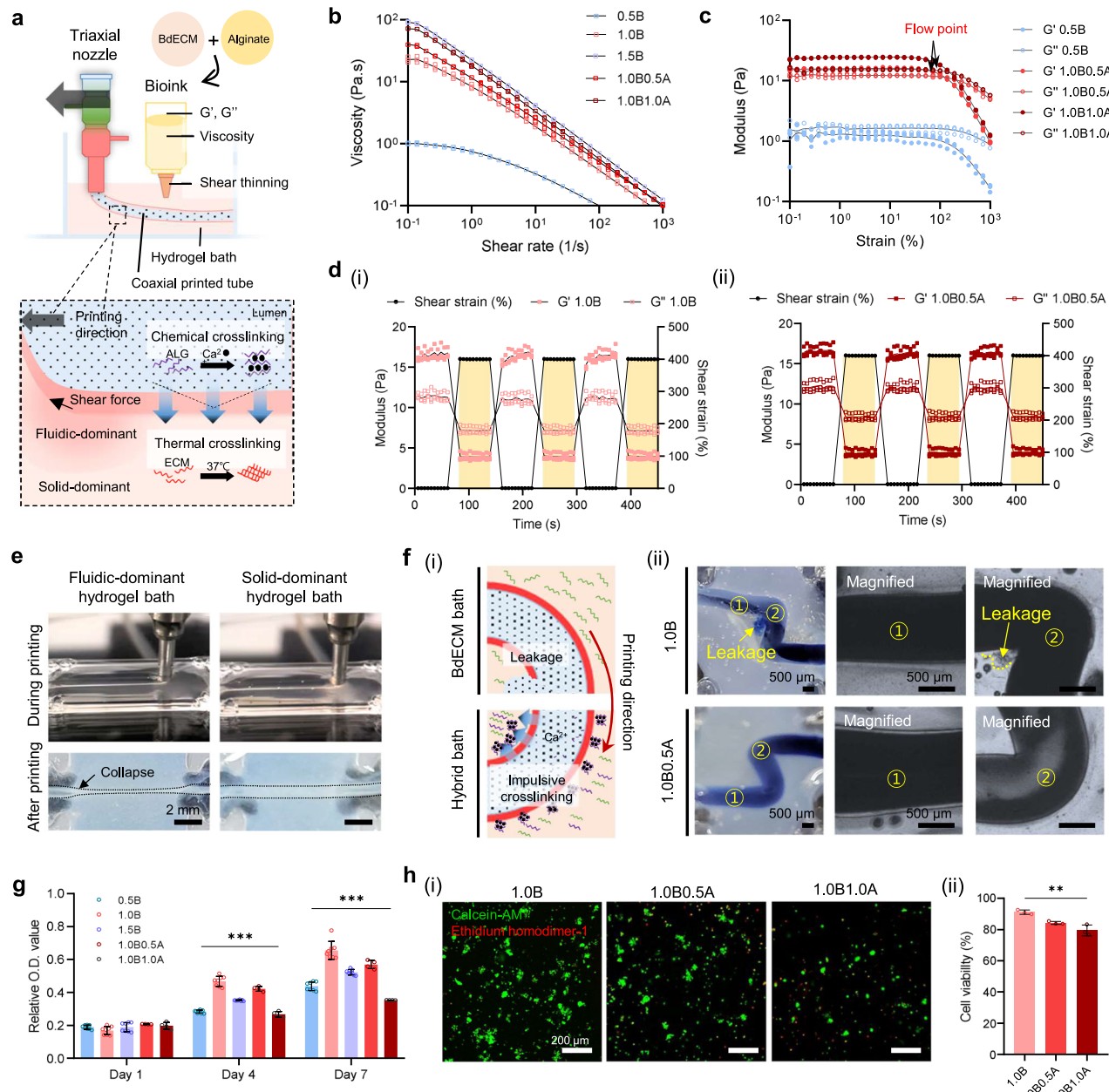

**Fig. 2 | Rheological, physiochemical, and biological assessment of hybrid brain-derived extracellular matrix (BdECM) bioink. a** Considering printability and shape fidelity, a hybrid bioink is formulated using BdECM and alginate. BdECM and alginate are labeled as B and A, respectively. **b** Most of BdECM-based bioinks show shear-thinning behavior and **c** stable sol–gel transition with clear flow points, where the storage modulus (G′) and loss modulus (G″) reverse at the threshold shear stress, but 0.5B bioink exhibits fluidic-dominant property. The data are compiled from $n = 3$ samples. Lines connect mean values. **d** BdECM-based bioinks including (i) pure BdECM bioink (1.0B) and hybrid BdECM bioink (1.0B0.5A) with concentrations above 1% show rapid shear recovery, implying the bioinks recover their shape after the applied stress is removed. **e** The fluidic-dominant hydrogel cannot sustain the hollow tubular structure, but the solid-dominant hydrogels exhibit suitable structural stability for applicability as bath materials and bioinks. Scale bars, 2 mm. **f** Rapid ionic crosslinking due to the presence of alginate prevents leakage at the curved portion of tubes. Scale bars, 500 µm. **g, h** Because metabolic activity and cell viability of neural progenitor cells decrease with increasing alginate concentration, 1.0B0.5A is selected as the optimal bioink considering the rheological, chemical, and biological properties. The results show mean ± SD from $n = 3$ samples. The significance is determined using two-sided t-test and one-way ANOVA (**$p \leq 0.01$; ***$p \leq 0.001$). Scale bars, 200 µm. Source data are provided as a Source Data file.

double-narrowed, and curved vasculatures with geometrical diversity were created (Fig. 3f).

**In-bath 3D triaxial bioprinting of in vitro brain metastasis model**
With the established bioprinting parameters, an in vitro brain metastasis model with multilayered cerebrovascular conduits was generated. The entire fabrication process could be automated by activating a single code. Through the gradual printing steps from housing fabrication to cerebral vessel construction, triple layered MCCs composed of human brain microvascular endothelial cells (HBMECs)-, human brain vascular pericytes (HBVPs)-, and NPCs-laden hybrid BdECM bioinks were successfully created (Fig. 4a). The model featured perfusable and geometrically controllable blood vessels (Fig. 4b). Owing to the crossing chamber design, the optimal condition could be generated after printing by circulating endothelial cell medium from the first medium chamber to the conduit for HBMECs' maturation;

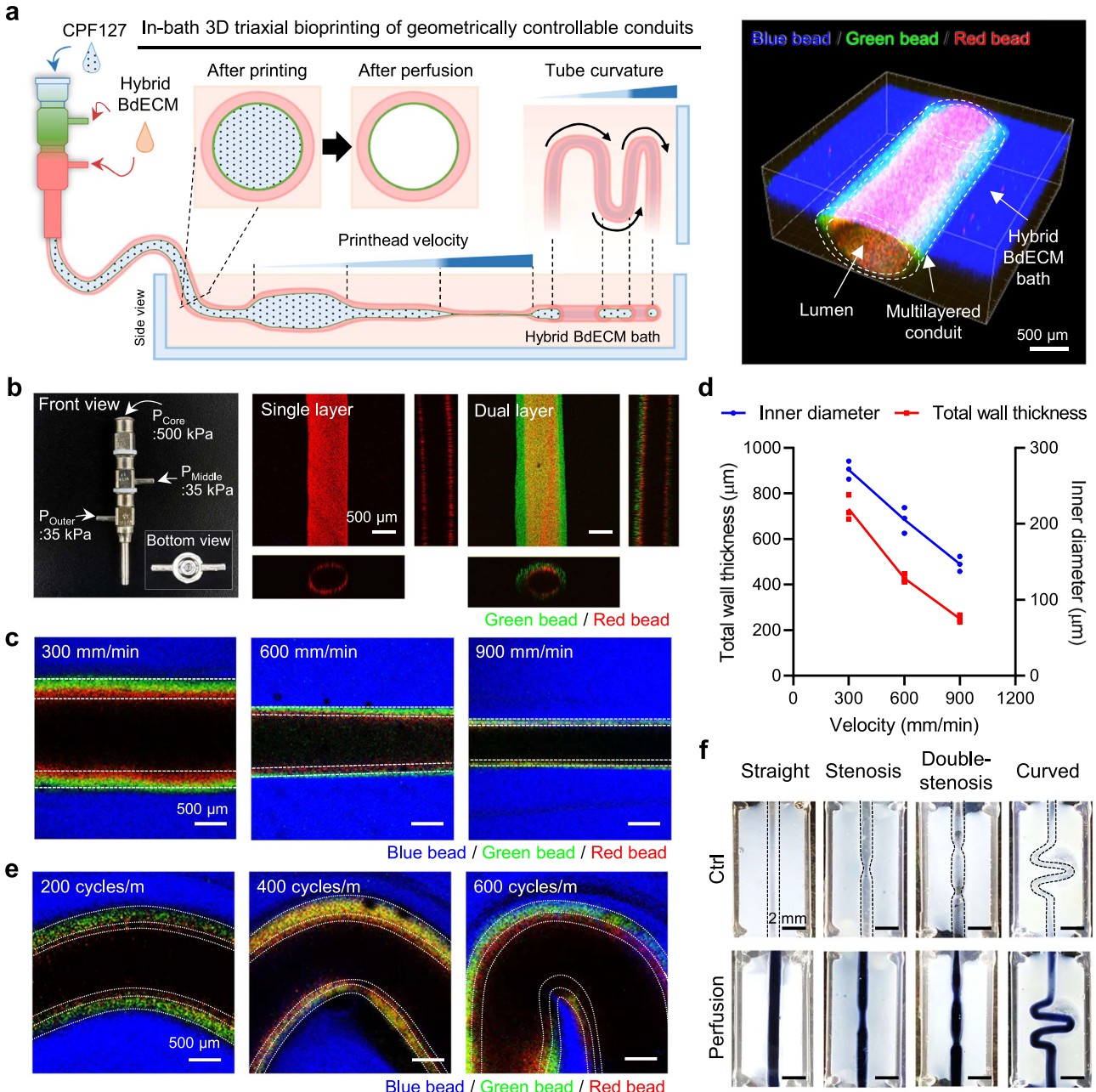

**Fig. 3 | Parameter study for geometrical control of a multilayered cerebrovascular conduit using in-bath 3D triaxial bioprinting technology. a** For direct fabrication of multilayered conduits, a triaxial nozzle, hybrid brain-derived extracellular matrix (BdECM) bioink, and calcium-added pluronic F −127 (CPF-127) were utilized. Dimensionally controllable conduits are constructed by managing pneumatic pressure, printhead velocity, and programming commands. Scale bar, 500 μm. **b** Using the triaxial nozzle with suitable pneumatic pressures, multiple layers of the hollow tube are easily organized. Scale bars, 500 μm. **c**, **d** As the printhead velocity increases, the inner diameter and wall thickness of the construct significantly decrease. Scale bars, 500 μm. The results are compiled from n = 3 samples. Lines connect mean values. **e** By appropriate G-code commands, tubes with different angles can be elaborated. Scale bars, 500 μm. **f** Based on the parametric study results, fabrication processes are designed for conduits with various geometries, including straight, stenosis, double-stenosis, and curved structures. Scale bars, 2 mm. Source data are provided as a Source Data file.

neural cell medium was loaded in the second medium chamber on top of the construct.

Fully mature MCCs were obtained after 14 days of culture (Supplementary Fig. 8). During maturation, the importance of each element, including the polymer housing, bath materials, and cellular components, was identified. The MCCs fully matured without any structural distortion resulting from the intercellular junctional force, whereas unexpected deformation was observed in the absence of housing or bath material (Supplementary Fig. 9). The encapsulated cells stretched their bodies and formed endothelial barriers (Fig. 4c,

Supplementary Fig. 10). Confluent CD31 and ZO-1 were detected in large, small, and curved models, indicating that the encapsulated HBMECs formed the selective cellular barriers regardless of the diameter and curvature (Fig. 4d(i)). Furthermore, the progenitor cells differentiated into astrocytes and neurons over time, expressing GFAP, which is an astrocytic marker (Fig. 4d(ii)), and TUJ1 and MAP2, which are neuronal markers (Fig. 4d(iii)). In addition, the expression of the cell maturation markers increased (Supplementary Fig. 11). The three-cell structure formed a mature cerebral vasculature composed of four types of the brain cells. In addition, vigorously stretched pericytes

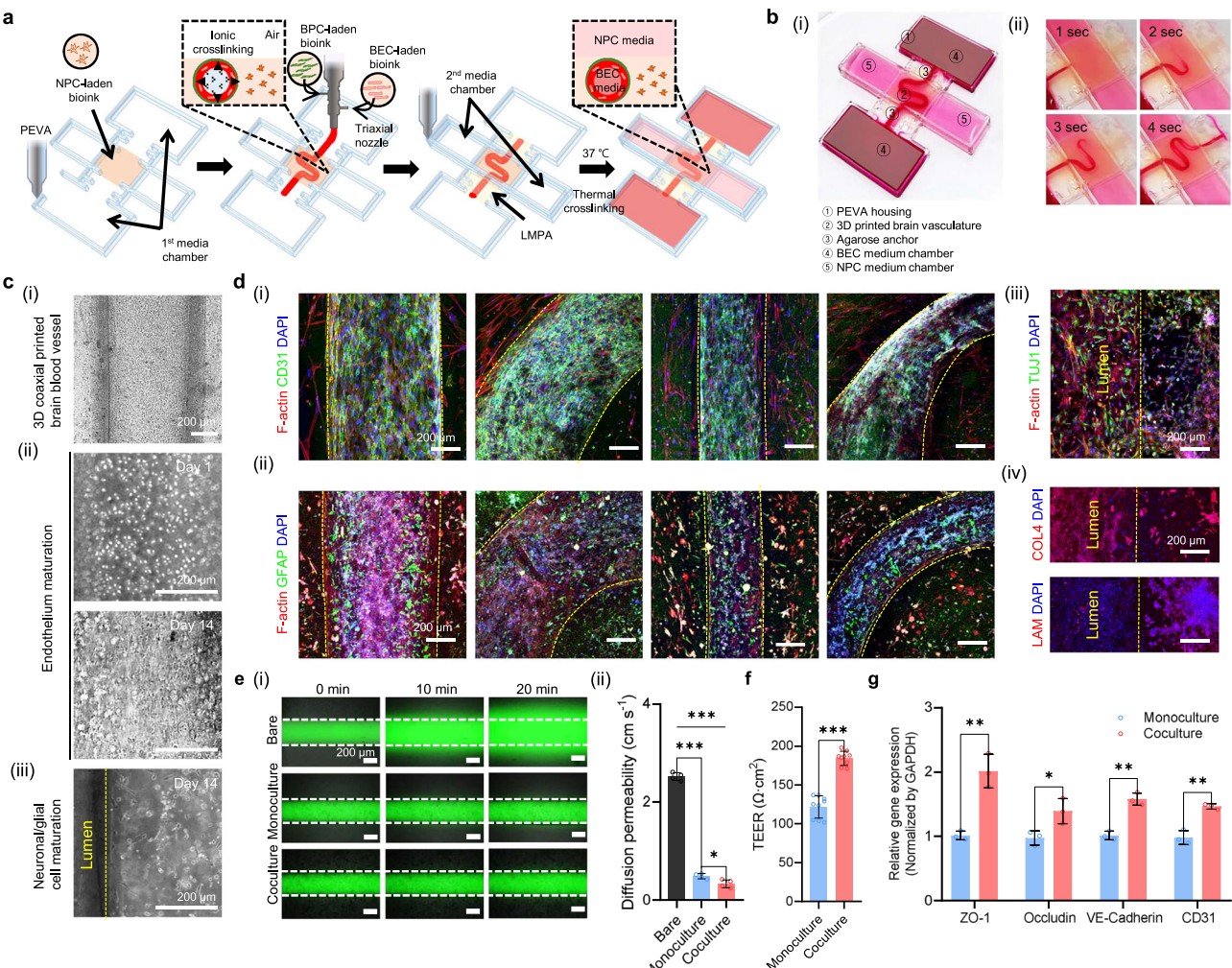

**Fig. 4 | In-bath 3D triaxial bioprinting of fully mature multilayered cerebrovascular conduits (MCCs). a** The strategy to construct an in vitro brain metastasis model containing MCCs was designed. **b** Through the gradual printing steps, perfusable brain blood vessels can be fabricated with diverse vessel curvatures. **c** The printed constructs are cultivated for 14 days to generate confluent cerebrovascular endothelium with the additional mural, glial, and neuronal cells. Scale bars, 200 μm. **d** Immunostaining results show that mature MCCs express (i) endothelial junctional marker (CD31), (ii) glial cell marker (GFAP), and (iii) neuronal marker (TUJ1). (iv) In addition, de novo production of extracellular matrix components including collagen IV (COL4) and laminin (LAM) is indicated. Scale bars,

200 μm. **e** Endothelial barrier function is verified via diffusion permeability assay. The results show mean ± SD from *n* = 4 samples. The significance is determined using one-way ANOVA (***, $p \le 0.001$; ns, no significance). Scale bars, 200 μm. **f, g** Coculture condition of endothelial cells with pericytes and glial/neural cells enhances trans-epithelial electrical resistance (TEER) and elevated the expression of the markers including the tight junction makers (*ZO–1, Occludin*), the adherens junction marker (*VE-cadherin*), and the endothelial marker (*CD31*). The results show mean ± SD from *n* = 3 samples. The significance is determined using two-sided t-test (***, $p \le 0.001$; **, $p \le 0.01$; *, $p \le 0.05$). Source data are provided as a Source Data file.

were observed using the filamentous actin (F-actin) marker and α-SMA, and de novo extracellular matrix (ECM) production of collagen IV and laminin was reproduced (Fig. 4d(iv)).

The generation of functional cerebrovascular tissue was evaluated using the diffusion permeability test (Fig. 4e(i)). For comparative assessment, three experimental groups were prepared: a bare model with no cells, a monoculture model with only endothelial cells, and a coculture model with MCCs. Compared with the bare group ($2.52 \pm 0.09$ cm s⁻¹), the endothelial cell-containing groups showed a significantly reduced diffusion permeability (monoculture, $0.49 \pm 0.07$ cm s⁻¹; coculture, $0.33 \pm 0.09$ cm s⁻¹), although there was no critical difference between the monoculture and coculture group (Fig. 4e(ii)). Meanwhile, the straight and curved MCCs exhibited no significant difference in the fluorescence intensity level of the diffused permeability probes (Supplementary Fig. 12). However, the assessment of trans-epithelial electrical resistance (TEER) values indicated that the coculture condition significantly improved the barrier function of the cerebrovascular conduits (Fig. 4f, Supplementary Fig. 13).

Furthermore, the coculture group exhibited higher gene expression levels of endothelial markers including the tight junction makers (*ZO-1, Occludin*), the adherens junction marker (*VE-Cadherin*) and the endothelial marker (*CD31*), indicating that the synergistic interactions among brain cells enhanced the endothelial functions (Fig. 4g). This demonstrates the morphological maturation and the brain tissue-specific microenvironment elaboration functionality of the developed cerebrovascular tissue construct.

## Cancer dissemination pattern changes depending on cerebrovascular curvature

To investigate the geometrical effects of vessel curves on brain metastasis, a 90°-bent MCC was prepared. Interestingly, when tumor cells were introduced into the tube, the cells preferentially adhered to the curved portion ($114.8 \pm 78.8$ cells mm⁻²) rather than the straight segment ($13.2 \pm 7.8$ cells mm⁻²) (Fig. 5a). To understand the causes of this phenomenon, multiparametric analysis considering the interplay of biochemical and biomechanical factors was required. Therefore,

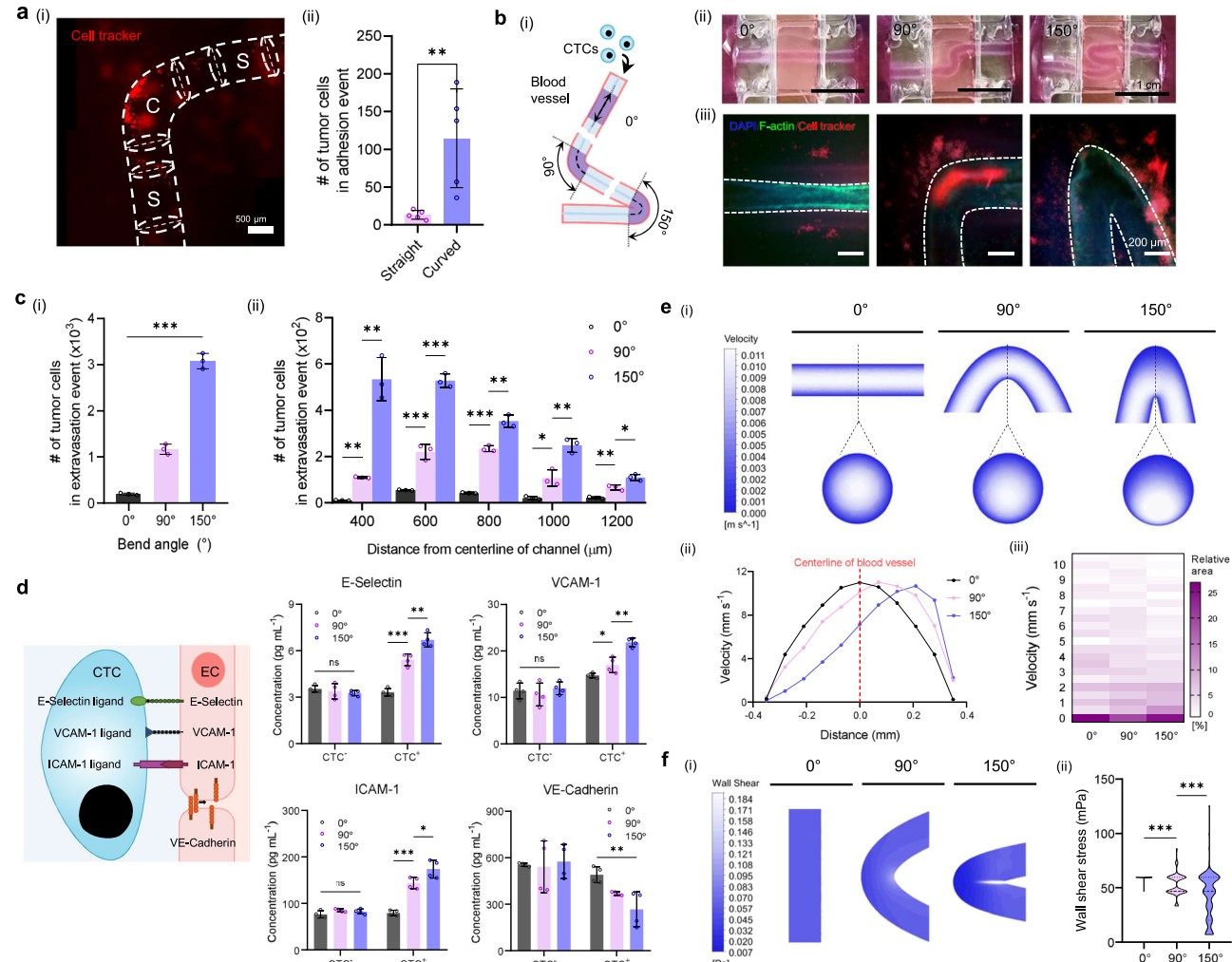

**Fig. 5 | Investigation of effects of vascular geometry on tumor extravasation.**
**a** The introduced tumor cells preferentially adhered to the endothelium at the curved, rather than the straight, portion. Scale bar, 500 μm. The results show mean ± SD from *n* = 5 samples. The significance is determined using two-sided t-test (**\*\*p ≤ 0.01*). **b** Fluorescence-labeled tumor cells were perfused into multilayered cerebrovascular conduits with three different bending angles (0°, 90°, and 150°), and tumor dissemination was visualized (**\*\*p ≤ 0.01*). Scale bars, 1 mm and 200 μm. **c** With the increase in the vascular bending angle, larger number of invading tumor cells are detected. The results show mean ± SD from *n* = 3 samples. The significance is determined using one-way ANOVA and two-sided t-test (**\*\*\*p ≤ 0.001; \*\*p ≤ 0.01; *\*p ≤ 0.05*). **d** The protein expression levels of cell adhesion molecules including E-selectin, VCAM−1, ICAM−1, PECAM−1, and VE-Cadherin are quantified. The results

show mean ± SD from *n* = 4 samples. The significance is determined using one-way ANOVA and two-sided t-test (**\*\*p ≤ 0.01; *\*p ≤ 0.05; ns, no significance). **e** Fluid-flow patterns for different vessel curves are investigated by computational fluid dynamics simulation, and (i) velocity contours are obtained. (ii) The average flow velocity is ~6 mm s⁻¹ and the biased velocity distribution increases as the vascular curvatures increase. (iii) The area of relatively low velocity (0.5–2.5 mm s⁻¹) is enlarged with the increase of vessel curve. **f** Similarly, (i) the distribution of wall shear stress biased and its gradient is generated following the increase of the vascular curvatures. (ii) In addition, the area with the low wall shear stress in the elbow portion is increased. The significance is determined using one-way ANOVA and two-sided t-test (**\*\*\*p ≤ 0.001; \*\*p ≤ 0.01; *\*p ≤ 0.05; ns, no significance). Source data are provided as a Source Data file.

MCCs were fabricated at three different angles (0°, straight angle; 90°, right angle; 150°, obtuse angle), and the tumor cells were perfused into the conduits for 24 h at an average velocity of 6 mm s⁻¹ under laminar flow conditions (Fig. 5b). The introduced metastatic cells adhered to the endothelial monolayer after 3 h of the perfusion, extravasated from the vasculature, and invaded the ECM, causing slight vasoconstriction (Supplementary Figs. 14–17). In particular, the number of extravasating tumor cells increased, following an increase in the vascular bending angle (195 ± 23, 1168 ± 111, and 3080 ± 169 cells) (Fig. 5c).

To verify intercellular communication between tumor cells and brain endothelial cells, adhesion molecules were quantitatively analyzed (Fig. 5d, Supplementary Table 1). After perfusion of CTCs, the expression of cell adhesion molecules (E-Selectin, VCAM-1, and ICAM-1) increased, whereas a key marker for the barrier function of blood vessel walls (VE-Cadherin) decreased. Significantly, the group of the larger MCC bending angles corresponded to higher measured

molecular concentrations, indicating the inflammatory response of brain endothelial cells. Meanwhile, blocking the function of the cell adhesion molecules demonstrated that these surface proteins assist tumor cell adhesion (Supplementary Fig. 18). On the other hand, VE-Cadherin was downregulated depending on the increase of vessel curve, demonstrating endothelial barrier disruption. This implies that the vessel curve provides a space for cellular interactions between tumor cells and brain endothelial cells during metastatic progression. Additionally, three different types of lung tumor cells including human lung circulating tumor cells, A549 cells, and H1299 cells were circulated to investigate the variations in metastatic behaviors based on the lung cancer cell types. However, there was no significant difference in the expression level of the adhesion proteins among the cells (Supplementary Fig. 19). However, to better understand the reasons for this phenomenon, a different approach was required.

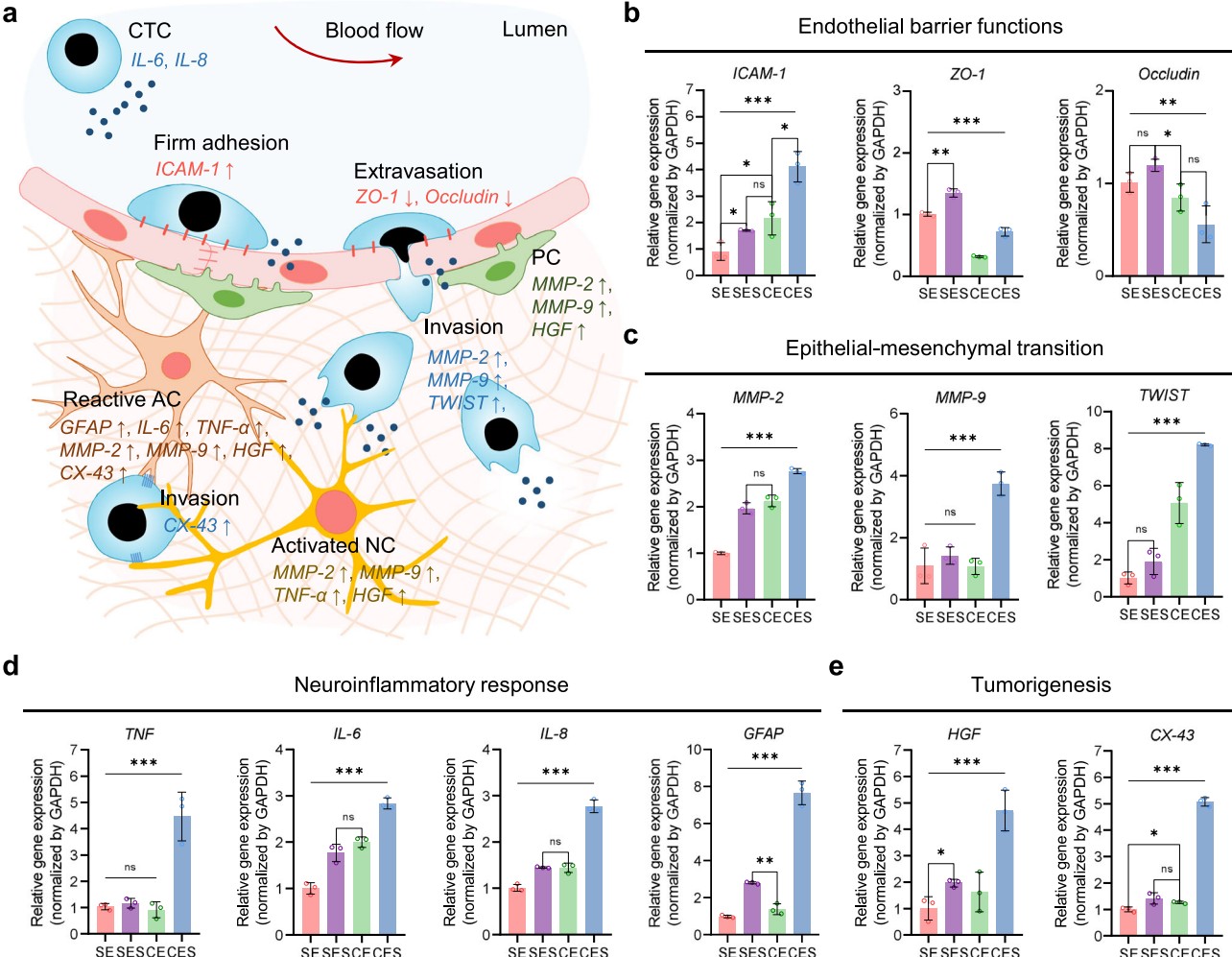

**Fig. 6 | Investigation of underlying molecular mechanisms in brain metastasis through comparative analysis, considering coculture conditions and vascular geometries of the multilayered cerebrovascular conduits (MCCs). a** Brain metastasis occurs through stepwise progression from tumor cell adhesion, extravasation, and invasion in brain tissue, accompanied by complex molecular interactions. Quantitative reverse-transcription PCR results demonstrate the effects of coculture and vascular curvatures on metastatic brain cancer development in four experimental groups: straight MCCs with endothelial cell monoculture (SE), straight MCCs with endothelial cell–stromal cell coculture (SES), curved MCCs with endothelial cell monoculture (CE), and curved MCCs with endothelial cell–stromal cell coculture (CES). The expression of key markers associated with **b**) endothelial barrier function (*ICAM-1*, *ZO-1*, and *Occludin*), **c** epithelial–mesenchymal transition (*MMP–2*, *MMP-9*, and *TWIST*), **d** neuroinflammatory responses (*TNF*, *IL-6*, *IL-8*, and *GFAP*), and **e** tumorigenesis (*HGF*, *CX-43*) are analyzed. The results show mean ± SD from $n = 3$ samples. The significance is determined using one-way ANOVA and two-sided t-test (***$p ≤ 0.001$; **$p ≤ 0.01$; *$p ≤ 0.05$; ns, no significance). Source data are provided as a Source Data file.

Accordingly, hemodynamic factors, including fluid velocity and WSS, were analyzed through computational fluid dynamics (CFD) simulations. Different fluid velocity distributions were predicted based on the vascular bending angle (Fig. 5e(i)). A velocity gradient of the perfused medium (Reynolds number, ~4.61) was generated due to the centripetal force of the fluid flow in the curved conduits[33]. Quantitatively, the flow velocity ranged from 0.1 to 11.02 mm s[-1] (Fig. 5e(ii)). The bias of the radial flow gradient intensified along with the curvature degree; the low velocity (0.5–2.5 mm s[-1]) area increased, whereas the high-velocity area decreased (Fig. 5e(iii)). Similarly, an elongated WSS gradient was generated by the curvature of the vessel (Fig. 5f(i)). The WSS at the extrados was lower than that at the intrados, implying that the WSS was proportional to the velocity gradient[34]. In particular, the range in which the WSS < 60 mPa was expanded for the curved groups (Fig. 5f(ii)). On the other hand, the fluid pressure gradient along with the longitudinal direction was observed, and the pressure change was slower on the extrados compared to the intrados (Supplementary Fig. 20). Additionally, no radial pressure gradient was indicated because of the low fluid velocity and low inertial force under the

experimental condition (Supplementary Fig. 21)[35]. For MCCs with different bending degrees but the same inner diameter and blood flow velocity, higher curvature may result in regions of low shear stress and velocity, promoting metastasis. These findings imply that various biomechanical and cellular factors can influence cancer metastasis through the prolongation of the residence time of tumor cells.

**Regulation mechanisms of brain metastases**
Underlying regulatory mechanisms of a series of hallmark events during brain metastasis development, including the collapse of endothelial barrier functions, acceleration of epithelial–mesenchymal transition, neuroinflammatory responses, and tumorigenesis in brain tissue, were investigated using our MCC-based approach (Fig. 6a). In particular, the effects of the anatomic features of brain blood vessels on cancer cell adhesion and extravasation were examined, considering the cellular composition and geometrical shape of the cerebral vasculature. For comparative analysis of gene expression on mRNA level, four types of models were prepared: straight MCCs with endothelial cell monoculture (SE), straight MCCs with endothelial cell–stromal cell

coculture (SES), 150°-bent MCCs with endothelial cell monoculture (CE), and 150°-bent MCCs with endothelial cell–stromal cell coculture (CES).

Circulating tumor cells first encounter the endothelial barrier, which is crucial for the regulation of tumor invasion. By secreting inflammatory cytokines, the tumor cells activate the vascular barrier and destroy endothelial junctions. The expression of the cell adhesion molecule, *ICAM-1*, and the endothelial junctional markers, *ZO-1* and *Occludin*, reflects the endothelial activation. Compared with the SE, the SES showed the enhanced *ICAM-1, ZO-1*, and *Occludin* gene expression, demonstrating that stromal cell-coculture condition enhanced the endothelial barrier function but also facilitated proinflammatory responses (Fig. 6b). Meanwhile, the CE and CES showed upregulated levels of *ICAM-1*, while *ZO-1* and *Occludin* were downregulated. The results indicated that the vessel curve promoted cancer cell dissemination and impaired the cellular integrity.

Following tumor penetration across the brain endothelial barrier, the tumor cells invaded the ECM, as indicated by several epithelial–mesenchymal transition markers, including endopeptidases, *MMP-2* and *MMP-9*, and an invasion marker, *TWIST* (Fig. 6c). The differences among SE, SES, and CE had relatively low significance levels. However, substantial upregulation of the migration markers was detected in the CES, representing accelerated tumor infiltration, implying that coculture and biased flow can provide synergistic effects on tumor cell migration.

Moreover, brain tissue remodeling is associated with brain metastases. The anti-tumor and pro-tumorigenic activities of brain-resident cells play an important role in tumor dissemination, which can be reflected in the expression of neuroinflammatory cytokines. The results showed significantly enhanced gene expression of inflammatory cytokines, including *TNF*, *IL-6*, and *IL-8*, and a reactive astrocyte marker, *GFAP*, in the CES (Fig. 6d). Meanwhile, tumorigenic markers, including *HGF* and *CX-43*, were also upregulated in the CES (Fig. 6e), proving that increasing vascular curvature promotes the invasion of tumor cells and intercellular communication with brain-resident cells.

Additionally, A549 exhibited a similar expression pattern to that of CTCs in the key markers (Supplementary Fig. 22). However, more substantial differences in the expression levels of some genes (*ICAM-1*, *MMP-2*, *MMP-9*, *IL-6*, and *CX-43*) were observed, indicating the potential influence of the cancer cell type on secondary tumor progression. Consequently, with comprehensive consideration, it could be concluded that cancer metastasis is a complex outcome arising from the dynamic interplay between various mechanical and biological factors.

## Discussion

We propose a direct bioprinting strategy to create anatomically and physiologically biomimetic cerebrovascular conduits to study brain metastases using a physiochemically enhanced BdECM-based hybrid bioink. By simply activating the printing code, the entire construct can be created within a remarkably short timeframe of less than 10 minutes. Moreover, ECM is an ultimate biomaterial that has evolved by nature to support the sophisticated and robust structures of tissues and organs; using its essential complexities would be the ideal[36–40]. Unfortunately, restoring the properties of the original tissues is difficult during the decellularization and solubilization process because of missing components including intercellular junctions and macromolecular chains. It demands the technological advancement to maintain the ECM proteins and innate characteristics of the tissue and an in-depth study for the correlation between protein components and the mechanical and biological performances of the tissue-derived materials[37,41,42]. On the other hand, to compensate for this destruction, we utilized alginate and formulated a hybrid BdECM bioink to improve the fine-tuned shape fidelity of the constructs in a brain-tissue-specific microenvironment. As well as its biological and mechanical suitability for the cerebrovascular cells in our platform, its chemical reactivity

that allowed the immediate ionic crosslinking between alginate and multivalent cations in CPF-127 for the elaboration of vessel curvatures with sharp angles. Utilizing the in-bath triaxial nozzle printing technique allowed the direct construction of MCCs with cellular complexity with four types of compartmentalized brain cells.

To develop a more physiologically relevant in vitro model representing native brain tissue, several factors have been considered. To recapitulate the cellular complexity of the brain, coculture with more diverse cells is required. However, this raises issues related to the medium composition and cellular duration[43,44]. To address these issues, neural progenitor cells have been utilized, and their function in constructing artificial tissue equivalents has been verified[45–47]. According to these techniques utilizing neural progenitor cells, we fabricated a multilayered cerebrovascular tissue containing four different types of cells under a simple culture condition. To evaluate the tissue formation of the 3D bioprinted cerebrovascular system, we used specific molecular markers indicating the endothelial tight junction development, pericyte function, and neuronal/glial differentiation. In particular, the expression of the GFAP marker suggests that the neural progenitor cells underwent astrocyte differentiation[47–49], but it also possibly indicates astrocytic activation in the in vitro culture environment, warranting further investigation[50]. Furthermore, the design of the crossing medium chamber in the platform supports hybrid perfusion systems including pump-based perfusion and pump-less perfusion, allowing high-throughput studies with better experimental efficiency.

The impact of cerebrovascular curvatures on metastatic cancer extravasation can investigated from multiple perspectives. For decades, biomechanical studies on metastatic cancer have primarily focused on physical entrapment in capillaries with a small diameter (<10 μm) and a low fluid velocity (<1.5 mm s$^{-1}$), and there is a growing need to investigate various hemodynamic factors affecting cancer dissemination[51,52]. To comprehend the multifactorial mechanisms involved, we established a physiologically relevant in vitro hemodynamic microenvironment that has the potential to facilitate tumor metastasis but no dominant biomechanical influencer. Instead, the variables under our control can accelerate the complex cascade. In the human circulatory system, small blood vessels have fluid flow velocities ranging from 0.5 to 10 mm s$^{-1}$ and WSSs varying from 0 to 30 dynes cm$^{-2}$ [22,53–57]. Cancer adhesion mainly occurs in the low WSS range below 15 dynes cm$^{-2}$ [58]. The adhesion efficiency is relatively high at WSS < 0.5 dynes cm$^{-2}$ but decreases as the WSS increases[59–62]. Considering these factors, our model was designed to investigate how various vascular curvatures influence cancer metastasis within human brain blood vessels. A control group was created with straight vessels (0° curvature, a diameter of 700 μm) having a fluid velocity of 6 mm s$^{-1}$ (corresponding to a flow rate of 2.31 mm$^3$ s$^{-1}$) and WSS of -0.6 dynes cm$^{-2}$. Subsequently, experimental groups were introduced with right-angle (90°) and obtuse-angle (150°) curves. Finally, this research demonstrated that the vascular geometries influence cancer metastasis. Despite the findings, a parametric study about the other hemodynamic factors, such as fluid velocity, flow rate, and blood pressure have not been fully investigated and the related in-depth study will be considered as a future work. On the other hand, the model replicated a series of metastatic cascades, including tumor circulation, adhesion, and extravasation, demonstrating the interplay of hemodynamic and molecular drivers in brain metastasis events. By bridging the gap between cellular biology and vascular biomechanics, our study provides insights into how vascular architectural complexity impacts the spread of cancer cells.

Although molecules such as cellular adhesion proteins and endothelial growth factors have been linked to cancer dissemination for decades, there are currently no drugs or therapies targeting cancer extravasation. The model presented in this study may, therefore, hold promise for future biomarker discovery in cancer extravasation, and it has the potential for transcriptome analysis and high-resolution, real-

time imaging in metastatic cancer research. Moreover, the model can be also utilized for the study of other cerebrovascular diseases, such as stroke, aneurysm, atherosclerosis, or even senescence, in which structural, mechanical, and biological changes can be a critical cause. Our technology's capability to fabricate the blood vessels with various structures, such as the curved, stenotic and swollen-shape, opens new possibilities for investigating these diseases. Additionally, the technology can be extended to incorporate diverse immune cell types, such as microglia and monocytes, allowing for a deeper exploration of the underlying mechanisms in these conditions[30,63–65]. Finally, this model has strong potential to provide not only a platform for real-time mechanism study of multiple brain diseases but also as a drug-testing platform for evaluation of the efficacy and safety of biopharmaceuticals.

In summary, this study substantiates the effects of cerebrovascular geometry on metastatic cancer extravasation. The study achieved four significant goals: (1) in-bath 3D triaxial bioprinting with physiochemically enhanced hybrid BdECM bioink was developed to directly construct multicellular cerebrovascular conduits; (2) dissemination pattern changes of circulating tumor cells were visualized in vitro; (3) local hemodynamic pattern changes were analyzed through CFD simulation to identify the critical values of WSS and flow velocity; (4) intercellular interactions and underlying molecular mechanisms were successfully investigated using the MCC model. Consequently, modeling complex cerebrovascular geometries could illuminate the biological and physical interplay of circulating tumor cells and native vasculatures and thus the underlying mechanisms governing brain metastasis. These findings mark a major technological breakthrough for advanced pharmaceutical and medical research of cerebrovascular diseases.

## Methods

### Hybrid BdECM Bioink preparation

The hybrid BdECM bioink was prepared by mixing BdECM with alginate. First, BdECM was extracted from porcine brains through a sequential physicochemical decellularization process[29,65]. Briefly, frozen porcine brains were chopped into $5 \times 5 \times 5$ mm$^3$ cubes and washed with phosphate-buffered saline (PBS; Tech and Innovation, South Korea) for 48 h. Then, chemical and enzymatic agents were treated to eliminate the cellular components: 0.5% (w/v) sodium dodecyl sulfate solution (Thermo Fisher Scientific, USA) for 24 h at 4 °C, 50 U mL$^{-1}$ deoxyribonuclease for 12 h at 37 °C, 0.5% (w/v) Triton X-100 (Sigma-Aldrich, USA) for less than 96 h at 4 °C, and 0.1% (v/v) peracetic acid in 4% (v/v) ethanol for 2 h at 4 °C. After lyophilization for 5 days, the acellular matrix was solubilized with 0.5% (w/v) pepsin (Sigma-Aldrich, USA) in 0.01 M HCl (Duksan, South Korea) for 3 days at 1000 rpm to obtain 5% (w/v) BdECM bioink. Subsequently, the BdECM pregel was mixed with 6% (w/v) alginate to formulate hybrid BdECM bioink of the desired concentration. Before the bioink was used, its pH was adjusted with 0.1 N sodium hydroxide solution (Samchun Chemical, South Korea).

### Biochemical analysis

The GAG contents in native brain tissue and BdECM were compared using a GAG quantification kit (Biocolor, UK) according to the manual. Briefly, to extract GAG components from 5 mg of the lyophilized samples, papain extraction reagent was treated for 3 h at 65 °C. After the sample tubes were centrifuged, the pellets were dissolved in deionized water. The test samples and reference standard aliquots were incubated with dye reagent for 30 min. After centrifugation of the tubes, dissociation reagent was added, and the absorbance was measured at 656 nm using a microplate reader (Thermo Fisher Scientific, USA) ($n = 3$). The collagen content was quantified using a total collagen assay kit (BioVision, USA) according to the developer's protocol. Briefly, 5 mg of native brain tissue and BdECM were homogenized using deionized water, and the samples and the collagen I standard

were incubated with ~12 M hydrochloric acid for 3 h at 120 °C. 10 min after Chloramine T reagent was added to each solution at room temperature, DMAB reagent was supplemented for 90 min at 60 °C. The absorbance of all samples was measured at 560 nm using a microplate reader (Thermo Fisher Scientific, USA). The source data generated in this study are provided in the Source Data file.

### Proteomics analysis

The protein composition was identified by applying the liquid chromatography-mass spectrometry (LC-MS/MS) analysis using Orbitrap Exploris 480 (Thermo Fisher Scientific, US), conducted by a biotechnology company (ebiogen, Korea). Briefly, after quantifying the protein content using the Bicinchoninic Acid Protein (BCA) assay, the proteins were digested using the filter aided sample preparation (FASP) method. The samples were sequentially treated with 500 mM tris (2-carboxyethyl) phosphine, 500 mM iodoacetamide, and 8 M urea for 15 min each. The trypsinized proteins was desalted using desalting column, and the remained solutions were evaporated using speed vacuum concentrator. The LC-MS/MS data was qualified and quantified based on spectral library of Uniprot Sus scrofa protein sequence database, and the discovered proteins were selected based on precursor q-value (≤0.01) and protein group q-value (≤0.01). The relative composition of each protein was calculated as a percentage of the total content. The source data generated in this study are provided in the Source Data file. The mass spectrometry proteomics data have been deposited to the ProteomeXchange Consortium via the PRIDE[66] partner repository with the dataset identifier PXD046191.

### Cell viability and proliferation assay

Cell viability was investigated using a live/dead assay kit (Invitrogen, USA). The ratio between live and dead cells was measured according to the developers' protocol. Briefly, the assay reagent was mixed with 5 μL of 4 mM calcein AM, and 2 μL of 2 mM EthD-1 solution in 1 × PBS was added on top of the hydrogel-encapsulated cells and incubated for 30 min at 37 °C. The fluorescence intensity was detected using a fluorescence microscope (ZEISS, Germany), and the ratio between the green and red signals was calculated ($n = 3$). The cell proliferation rate was measured using a Cell Counting kit-8 assay (Dojindo, Japan). The rate was calculated according to the guidelines by detecting dehydrogenase activity for 7 days. Briefly, 100 μL of CCK-8 solution in 1 mL of cell culture medium was added to the sample and incubated at 37 °C in the dark. After 4 h, the absorbance of the supernatant was measured at 450 nm using a microplate reader (Thermo Fisher Scientific, USA) ($n = 4$). The source data generated in this study are provided in the Source Data file.

### Rheological and mechanical characterization

The rheological and mechanical performances of BdECM-based bioinks and native brain tissue were validated using an Advanced Rheometric Expansion System (TA Instruments, USA). The testing gap between 20-mm-diameter geometry and bioinks was adjusted to 250 μm after loading 250 μL of the materials. Viscosity was measured by increasing shear rate from 0.1 to 1000 s$^{-1}$ at 15 °C ($n = 3$). Yield and flow point were investigated by measuring complex modulus under shear strain from 0.1 to 1000% at 15 °C ($n = 3$). The shear recovery properties were tested by repetitively changing shear strain from 10 to 1000% at 15 °C. The analysis of gelation kinetics was performed with temperature sweep oscillatory test by increasing temperature from 4 to 37 °C. Compressive modulus was measured under shear strain ranging from 0.1 to 1000% at 15 °C ($n = 3$). The source data generated in this study are provided in the Source Data file.

### Cell culture

HBMECs (Innoprot, Spain, P10361) were cultured using an Endothelial Cell Culture Medium kit according to the manufacturer's instructions.

HBVPs (ScienCell, USA, #1200) were maintained in Pericyte Medium (ScienCell, USA). ReNcells (Sigma-Aldrich, USA) were cultivated in a growth medium composed of DMEM/F-12 (Gibco, USA), B27 (Thermo Fisher Scientific, USA), heparin (Stemcell Technologies, USA), penicillin-streptomycin, amphotericin, EGF, and bFGF on a laminin-coated cell culture dish. After 5 days, the differentiation medium without EGF and bFGF was added. 1% penicillin-streptomycin (Gibco, USA) was supplemented to culture media. The cells were cultured in a humidified incubator at 37 °C, 5% $CO_2$. The human lung circulating tumor cells (Celprogen, USA, 36107-34CTC, Supplementary Table 2) were cultured using a Human Lung Circulating Tumor Cell Extracellular Matrix and a Human Lung Cancer Stem Cell Complete Growth Media according to the manufacturer's instructions. A549 and H1299 cells (Korean Cell Line Bank, Korea, 91299) were cultivated in a growth medium composed of RPMI1640 with L-glutamine (Hyclone, USA), 25 mM HEPES (Sigma, USA), 25 mM sodium bicarbonate (Sigma, USA), and 5% fetal bovine serum (Gibco, USA), following the protocol of Korean Cell Line Bank.

## Construction of in vitro brain metastasis model
The brain metastasis model was automatically fabricated by utilizing our in-house built hybrid 3D bioprinting system (ICBS)[67] in an isolated 3D printing dedicated room. The internal temperature of the room was maintained at 18°C to inhibit the crosslinking of the printed hydrogel, and the humidity was maintained at 40-50%. After the first medium chamber was printed with poly (ethylene/vinyl acetate) (PEVA; Polysciences, USA), an NPCs-laden hybrid BdECM bioink was printed as a bath material. Afterward, a customized triaxial nozzle (Ramé-Hart Instrument, USA) was submerged in the bath after loading materials into each part: core part, 35% (w/v) Pluronic F127 in 100 mM calcium chloride solution; intermediate part, HBMEC-laden (final cell density: $1 \times 10^7$ cells $mL^{-1}$) hybrid BdECM bioink; shell part, HBVP-laden (final cell density: $1 \times 10^6$ cells $mL^{-1}$) hybrid BdECM bioink. The tubular structures were created in the bath following G-code commands with printhead velocity ranged from 300 to 900 mm $min^{-1}$. After anchoring the construct using low-melting-point agarose (Thermo Fisher Scientific, USA), a second medium chamber was printed on top of the first layer of the housing. After 30 min of thermal crosslinking at 37 °C in an incubator, a rocker shaker with a tilting angle of 9° and 5 rpm was used to stabilize the chip until the sacrificial material was removed and the cells stretched their bodies (Supplementary Fig. 23a). Suitable cell culture media were provided from the separated medium chambers in platform during this time. Following the stabilization process, the chip was connected to a peristaltic pump, and a laminar flow was perfused for tissue maturation (Supplementary Fig. 23b).

## Permeability test
The endothelial barrier function was quantified by measuring the fluorescence intensity of 70 kDa FITC-labeled dextran (Sigma-Aldrich, USA) that diffused from the printed vascular channel with 25 μg $mL^{-1}$ concentration of the particles in endothelial cell-growth medium at a flow rate of 20 μL $min^{-1}$ for 20 min. The intensity was detected by a fluorescence microscope (ZEISS, Germany) and quantitatively calculated using ImageJ software (National Institute of Health, USA) using the following equation.

$$P_d = \frac{1}{I_1 - I_b}\left(\frac{I_t - I_1}{t}\right)\frac{d}{4} \qquad (1)$$

which relates the diffusion permeability coefficient ($P_d$) considering the initial fluorescence intensity ($I_1$), background fluorescence intensity ($I_b$), final fluorescence intensity ($I_t$) at a certain time ($t$), and diameter of the channel ($d$) ($n = 4$)[68]. The source data generated in this study are provided in the Source Data file.

## Tumor cell perfusion analysis
Tumor cell perfusion analysis was conducted using a peristaltic pump to achieve laminar flow with an average velocity of 6 mm $s^{-1}$ and an average wall shear stress of 60 mPa inside the fabricated conduits. The tumor cells suspended in a cell culture medium were perfused for 24 h, and the chip was harvested at time points of 3 h and 24 h to observe the tumor dissemination pattern.

## Computational fluid dynamics simulation analysis
The 3D geometries of the models were generated using a commercial software, Solidworks (Dassault Systems, FR). The conduits were modeled with the diameter of 700 μm and the length of 16 mm. Subsequently, fluid flow simulations were performed using a commercial software, ANSYS (2023 R1, ANSYS Inc., USA), following a user guide and established methods. The fluid flow in the channel was assumed to be Newtonian fluid with a density of 1000 kg $m^{-3}$ and a viscosity of 0.00093 Pa·s. The vascular walls were assumed to be rigid and a no-slip boundary condition was applied to them. The initial velocity was ~6 mm $s^{-1}$ at the inlet and the initial pressure was 0 Pa at the outlet. The length of time step was 0.001 s. The fluid flow velocity, WSS, and pressure gradient were calculated employing the shear stress transport k-ω model, according to Navier-Stokes equations,

$$\nabla \cdot u = 0 \qquad (2)$$

$$\rho \frac{\partial u}{\partial t} + \rho u + \nabla u + \nabla p = \mu \nabla^2 u \qquad (3)$$

where $u$ represents the velocity, $\rho$ represents the fluid density, $\mu$ represents the dynamic viscosity, $t$ represents the time, and $p$ represents the pressure[69,70].

## Quantitative reverse transcription-polymerase chain reaction
Comparative gene expression levels were determined via the quantitative reverse transcription-polymerase chain reaction method. The total RNA was isolated through the trizol method as follows. Homogenized samples in trizol were phase-separated by adding chloroform (TaKaRa, Japan). RNA in the aquatic phase was precipitated using isopropanol and 75% ethanol. The extracted RNA was reversely transcribed to produce complementary DNA (cDNA) using a cDNA synthesis kit (Thermo Fisher Scientific, USA). The primers were designed for the specific targets (Supplementary Table 3) and added with SYBR-green for comparative analysis of gene expression level. The fluorescent signals released was quantitatively monitored using a StepOne Plus Real-Time PCR System (Applied Biosystems, USA) ($n = 3$). The source data generated in this study are provided in the Source Data file.

## Enzyme-linked immunosorbent assay
Comparative expression levels of cell adhesion molecules were measured using quantitative assay kits (R&D Systems) following the manufacturer's instructions. Briefly, conjugate, standard, and samples were added to plate and incubated at room temperature for 2 h. After rinsing four times with wash buffer, substrate solution was subsequently added at room temperature for 30 min. After adding stop solution, the absorbance was measured at 450 nm using a microplate reader (Thermo Fisher Scientific, USA) ($n = 3$). The source data generated in this study are provided in the Source Data file.

## Immunostaining
Protein expression was visualized using an antibody-based method. The printed constructs were fixed in 10% neutral buffered formalin (Biosesang, South Korea) for 3 h at room temperature. After washing three times with 1 × PBS, the fixed samples were permeabilized using 0.1% Triton X-100 (Sigma-Aldrich, USA) for 10 min. Unspecific binding

was blocked with 1% bovine serum albumin (BSA) (Thermo Fisher Scientific, USA) solution for 1 h, and the primary antibodies (anti-CD31 antibody, ab9498, Abcam, UK; anti-GFAP antibody, MA5-12023, Invitrogen, USA; Anti-TuJ1 antibody, 32-2600, Invitrogen, USA; Anti-Collagen IV antibody, ab6586, Abcam, UK; Anti-MAP2 antibody, Sigma-Aldrich, M4403; Anti-MAP2 antibody, Abcam, ab32454; Anti-ZO-1 antibody, Invitrogen, 33-9100; Anti-ICAM-1 antibody, R&D systems, BBA3; Anti-VCAM-1 antibody, R&D systems, BBA5; Anti-Laminin antibody, L9393, Sigma-Aldrich, USA) in 1% BSA-PBS (dilution 1:200) were subsequently incubated at 4 °C overnight. A mixture of 4',6-diamidino-2-phenylindole (DAPI), Alexa 488- and Alexa 546-conjugated secondary anti-mouse or anti-rabbit antibodies (Thermo Fisher Scientific, USA), and TRITC- or FITC-labeled phalloidin was added and incubated at room temperature for 3 h. Finally, immunofluorescence signals were detected using a fluorescence microscope (ZEISS, Germany).

### Statistics and reproducibility

All quantitative data are presented as the mean ± SD from more than three independent samples at least. Statistical differences in the data were verified using GraphPad Prism v. 8.0.1 (GraphPad Software Inc, San Diego, CA). Two-tailed Student's t-test was conducted for two groups, whereas one-way analysis of variance was performed for multiple groups. The statistical significance was set at $p < 0.05$.

### Reporting summary

Further information on research design is available in the Nature Portfolio Reporting Summary linked to this article.

## Data availability

The mass spectrometry proteomics data have been deposited to the ProteomeXchange Consortium via the PRIDE[66] partner repository with the dataset identifier PXD046191. All other data supporting the findings of this study are available within the article and its supplementary files. Any additional requests for information can be directed to, and will be fulfilled by, the corresponding authors. Source data are provided with this paper.

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

## Acknowledgements

This work was supported by the Alchemist Project 1415180884 (20012378, Development of Meta Soft Organ Module Manufacturing Technology without Immunity Rejection and Module Assembly Robot System) funded by the Ministry of Trade, Industry & Energy (MOTIE, Korea) (D.-W. Cho and W. Park), Korean Fund for Regenerative Medicine funded by Ministry of Science and ICT, and Ministry of Health and Welfare (22A0106L1, Republic of Korea) (D.-W. Cho and W. Park), and the National Research Foundation of Korea (NRF) grant funded by the Korean Government (MSIT) (No.2022R1C1C1004803 and No. 2022R1A5A2027161) (B. S. Kim and J.-S. Lee).

## Author contributions

W.P., B.S.K., and D.-W.C. conceived the concept of the study. W.P. performed most of the experiments and wrote the manuscript. J.-S.L. performed the computational fluid dynamics simulation. G.G. contributed to initiate the technological development. D.-W.C. and B.S.K. provided overall guidance and supervision.

## Competing interests

The authors declare no competing interests.
