## [Peer Review File · Nature Communications]

REVIEWER COMMENTS

Reviewer #1 (Remarks to the Author):

Park and co-worker demonstrated a mature three-layered cerebrovascular channel with erratic curvatures using in-bath 3D triaxial bioprinting technique and a brain-specific hybrid bioink containing an ionically crosslinkable hydrogel for studying the physical and molecular mechanisms of cancer extravasation in vitro. In this manuscript they showed that more tumor cells are arrested in the curved vascular channels and might cause under low velocity and wall shear stress accelerates the molecular signatures of metastatic potential. The authors claimed that the proposed in-bath 3D triaxial bioprinting technique in conjunction with hybrid bioink technological to develop erratic curvatures of vascular follow that driving brain metastases might help pharmaceutical and medical research.

Comments:

- a) Overall, this work done in this manuscript will improve the current 3D field in marginal level as there are so many other papers already showed similar approach where they described the 3D printing using dECM from many different organs including brain (<https://doi.org/10.1021/acsbiomaterials.9b01512>). Moreover, the authors claimed that their bioinks provided a structurally supportive environment for hollow structure. On contrary, there are reports to achieve the same using 3D bioprinting of heterogeneous bi- and tri-layered hollow channels within gel scaffolds using scalable multi-axial microfluidic extrusion nozzle (DOI 10.1088/1758-5090/aaf7c7). Considering the outcome of the present work it is hard for the reader to find a substantial improvement in the technological aspects where are currently available in the field.
- b) Another philosophical question was not addressed here is that the "seed and soil" hypothesis where the influence in abluminal space is also important. In addition, the overall flow rate as well as oxygen concentration were not addressed.
- c) Figure 3E and 3F were not mentioned in results and discussion.
- d) Higher magnification of fluorescent images will substantiate the claims: mature endothelium, HBMECs formed mature vascular tissue, the progenitor cells differentiated into astrocytes and neurons over time.
- e) Figure 4G and 4 H should be marked as 4E and 4F respectively.
- f) The authors should test diffusion permeability using curved structure developed with triple layered MCCs composed of human brain microvascular endothelial cells (HBMECs), human brain vascular pericytes (HBVPs), and NPCs.
- g) The authors should define CMM.
- h) The authors should justify how these three different angles (0°, 90°, and 150°) were studied?
- i) Why the fluid velocity restricted to 6mm/min?
- j) Controls for Figure 5D are missing. The authors should provide proteins expression in cases of three different angles without cancer cells.
- k) It is not clearly articulated in the results, the angle for curved MCC in Figure 6A.
- l) The authors should involve CTC from metastatic and non-metastatic tumors and present in Fig 5D, 6B-6E to prove their hypothesis.

Reviewer #2 (Remarks to the Author):

Park et al. utilize 3D bioengineered models to analyze the biology of cells that comprise the neurovascular unit and regulate the BBB. In addition, mechanisms of adhesion and communication between neurovascular cells and arrested metastatic cells in the circulation and during the extravasation process are investigated. A particular focus is placed on the effect of curvature in the ex vivo cerebrovasculature and how this impacts BBB permeability as well as metastatic seeding and dissemination into the modeled brain parenchyma. The manuscript is well written. The scientific premise is solid and the data are presented in a rational manner. Overall, the results are of high

quality. However, there are several significant conceptual and technical issues that limit enthusiasm for the study.

1. The major criticism of the manuscript is that, as presented, the data are largely proof-of-concept, confirmatory in nature, and incremental in impact. The data in the first four figures are largely illustrating/confirming the basic premise for design and use the 3D system. The results in Figures 4 and 5 include gene expression data for adhesion and communication between the tumor cells and the vascular endothelial cells, including analysis of effects of vessel curvature on circulating tumor cell arrest and extravasation. However, these cell adhesion molecules and endothelial junction proteins are already established as important for BBB integrity and cell rolling and arrest. There are very few new findings that add novelty to the study. It is unclear how this 3D system is distinct from other similar published systems for the neurovascular unit.

2. Figures 1 and 2

- The protein composition of the porcine BdECM remains unclear. The authors extract proteins from pieces of decellularized brain. Presumably they are working with ECM proteins from basement membranes as well as the interstitial matrix. However, they are using the bioink/algenate mixtures mostly as a vascular basement membrane model, although there are likely interstitial matrix components present. In Supp Fig. 1 they show GAGs, suggesting interstitial matrix, as well as collagens although the exact collagen composition is unclear. Also, there are likely growth factors and other bound proteins in the BdECM blend. A more detailed experimental characterization of the BdECM is important.

3. Figure 3 and 4

- The authors use neural progenitor cells (presumably neurospheres) in the co-culture system. It is unclear why they do not use differentiated astrocytes, since these cells are important components of the neurovascular unit and regulate BBB development and physiology. The NPCs probably express basal levels of GFAP. Tuj1 is a marker for immature neurons, emphasizing the questionable physiological relevance of using NPCs in the system. As presented, this is more a model for the neural stem cell niche, which makes it distinct from a general model of the neurovascular unit.

- The authors mention use of pericytes in the co-culture system but contributions by these cells to BBB integrity and tumor cell extravasation are not sufficiently analyzed. There is insufficient inclusion of pericyte markers.

- In Figure 4E the authors show that single cultures of endothelial cells are sufficient to form a barrier to fluorescent tracers, bringing into question the functions of the additional mural cells (pericytes) and glial/neuronal cells. Perhaps an additional assay like TEER would add more insight. They do show that junctional proteins are elevated in the co-culture systems; however, it is uncertain why PECAM/CD31 is included since it is a marker for endothelial cells outside of the brain and is not directly linked to brain endothelial barrier integrity.

Figures 5 and 6

- The authors show that the tumor cells selectively home to bends in the vessel wall. How does this relate to the capillary network in the brain when a tumor cell arrests? Also, there are very few details about the metastatic tumor cells being utilized. More technical specifics should be included. Also, related to the fluorescent tracers in Figure 4E, are the curves/bends in the vessel wall correlative with increased endothelial barrier integrity and how does that relate to tumor cell arrest?

- The expression of several cell adhesion molecules (ICAM, VCAM, etc.) with known functions in cancer and immune cell arrest in the circulation are analyzed. However, functional assays are not included. The authors should use function-blocking antibodies or gene silencing efforts in cancer cells to determine which pathways are important for arrest at the endothelial lumen. Also, including all the details for the cell adhesion molecule expression in the results section is very distracting to the reader. These numbers should be shown in the figure or as supplemental data.

- The authors show that tumor cells induce a down-regulation in various endothelial cell junction proteins and an increase in neuroinflammatory molecules. What does this mean functionally? Is BBB

integrity impacted in their system? What new and important mechanistic information could be gained from this system, rather than simply confirming what is known? For example, is the system amenable to transcriptome analysis or high-resolution, real time imaging to analyze the steps in tumor cell extravasation?

Reviewer #3 (Remarks to the Author):

Summary:

The authors utilized a direct 3D bioprinting strategy to construct multilayered cerebrovascular conduits (MCCs) with varying geometries and curvatures to study brain metastasis. First, a hybrid BdECM bioink formulation was optimized for printing fidelity and cell viability. An in vitro brain metastasis model was constructed using triple layered MCCs consisting of HBMECs (endothelial), HBVPs (pericytes) and NPCs (neural progenitor cells) with perfusable and geometrically tunable blood vessels. Using their platform, the authors observed that vascular curvature results in the upregulation of endothelial adhesion molecules and downregulation of endothelial tight junction markers, as well as the upregulation of EMT and tumorigenesis markers in cancer cells. Ultimately, this platform has the potential to be used as a preclinical model for studying the molecular mechanisms involving brain metastases.

Major Comments

Comment 1: The authors state that most tissue engineering strategies still rely on complex techniques to fabricate advanced platforms to study brain vasculature. However, the fabrication process involved in this study still appears quite rigorous. How does this platform compare with advanced 3D microfluidic models in terms of ease of use and technical robustness? Can devices be more reliably established here? How this platform differs from others in this regard should be further specified.

Comment 2: The authors stated that their model can recapitulate the cerebral mechanical microenvironment but should compare the mechanical/rheological properties of their platform to that of native tissue if possible in order to come to their conclusion.

Comment 3: (I could be missing something here but...) The cancer cell line/type as well as the methods to maintain cancer cells are not specified anywhere in the manuscript. Do these cancer cells have an invasive or non-invasive phenotype? What cancer cells (and from what organ) are being used in the metastasis model? Are these patient-derived circulating tumour cells?

Comment 4: (related to the previous comment) With regards to the cancer cell type being used for these studies, does the model capture the behaviour of invasive/non invasive cancer cell phenotypes?

Comment 5: The authors demonstrate that changing the vessel curvature influences hemodynamic factors such as wall shear stress and fluid velocity gradients. Do/would these changes in curvature and fluid profile directly affect cancer cell EMT? That is, is cancer cell EMT directly controlled by the expression endothelial cell markers or do mechanical factors, or the mechanical environment play a role in this as well?

Comment 6: From fluid mechanics point of view, it is easy to understand the velocity close to convex side is higher compared to velocity close to the concave side. However, as a result of the velocity gradient, given the conservation of mechanical energy, the pressure gradient should be the opposite, which is higher pressure close to the concave side and lower pressure to the convex side, which intuitively should "push" cells towards the convex side, which seems contradictory to what have been observed here. Has the authors quantify/simulate the pressure gradient close to the curved area? How the pressure gradient, shear stress play a role on regulating cancer metastasis?

Comment 76: Cancer cell EMT adhesion markers (cadherins) also play a significant role especially in the context of adhesion/extravasation/metastasis. Are these factors also upregulated in addition to the ones already presented (MMP-2, MMP-9, TWIST)?

Comment 87: The authors should specify how many repeats were performed for each experiment.

Minor Comments

Comment 98: An experimental timeline including all aspects of the experiment involving cancer cells and endothelial cells would be nice to include to allow readers to visualize the procedures performed.

Comment 910: What environment (temp., humidity(?)) is the bioprinting taking place? Is this performed at room temp. in sterile conditions (that is, in a biosafety cabinet)? Conditions should be specified.

Comment 110: Figure 5C(ii.) – Statistical significance between groups in this figure is missing.

Comment 121: The authors mention that this model could be used to study other cerebrovascular conditions such as stroke or aneurysm. How would aspects of these diseases be incorporated into the platform?

Comment 13: The implications of the shear thinning behaviour is not clear, why it is important and how that compared to native ECM need to be provided.

Comments 14. Many figures, for example, Figure 2C, 2D, 5D, etc., are poor quality, with very blur images of the curve and the legends, leading to great difficulty to understand the figures.

Comments 15: Figure 4G and 4H are missing. On page 4 line 152, Fig. 4G is mentioned but this reviewer couldn't find it in Figure 4, instead, it seems the authors are referring to 4E.

Comments 16: In figure 6, SE, CES etc. are not explained, what are these cells?

Reviewer #4 (Remarks to the Author):

The research article by W. Park et al. introduces a new class of 3D bioprinted in vitro models of cancer, consisting of various three-layered vascular geometries, to study mechanisms of cancer extravasation. The team used an innovative triaxial bioprinting technique in a support bath, as well as a hybrid brain-specific bioink formulation comprising of brain dECM and alginate to better mimic the native tissue. This platform was used to correlate cerebrovascular geometry/curvature, and flow hemodynamics, to the molecular signatures of cancer cells adhesion and metastatic behavior in the brain. The work is innovative, impactful, and offers great potential for clinical/therapeutic translation of the possible mechanisms that could be identified using this platform. There are however a number of major and minor issues that need to be addressed to further improve the flow and the strength of the submitted work:

Major Comments:

- MCC maturation is not really supported by the minimal data presented on each cell phenotype. For EC maturation, more in-depth data from gene expression profile or more extensive IHC staining will be required to assess maturity of endothelium. Same need exists for other cells/tissue involved. The NPC differentiation certainly needs more in-depth analysis to claim maturation of brain tissue. The IHC images in Figure 4D are too vague and not really providing much insight to the cells phenotype. I

strongly recommend presenting more extensive results, at different magnifications, to clearly show cellular structure and junctions.

- The perfusion bioreactor set-up used in this study is not fully elaborated. Schematics and photos of the assembly, how they connect to the inlet and outlet of the printed endothelialized channel, the flow parameters (flow rate, wave form, etc.) should be all explained. How the prescribed rate compares with the clinical/in vivo levels of flow in the brain vasculature should be discussed as well. Does the flow rate match with in vivo? This is particularly critical in order to be able to correlate the hemodynamics factors to the cell response. If the WSS levels are not within the same range, the study on cell response does not seem to be logical.

- The cell maturation is pretty important in this study, especially for the endothelial layer, as authors mainly focus on the extravasation of cancer cells. Not having a full continuous endothelium, could readily 'leak' the CTCs out of the printed channels and this may not have anything to do with the tumor extravasation in vivo. In Figure 5, they assessed CTC migration distance based on the angle of the channel. The formation of the endothelium is typically not ideal in tortuous channel geometries, leaving some fenestrations/gaps in the endothelium, which could in turn increase the CTC escape. This concern is of more significance for this study, since the IHC images shown in Figure 5 do not show a strong and continuous GFP signal from the ECs. So, authors must provide more careful and zoomed-in analysis of endothelial layer in the three angle groups to exclude the potential effect of channel geometry on endothelium formation.

- There is some ambiguity about the specific bioinks that used for each and different parts of the study plan: the ink for BPE and BEC printing (were cells encapsulated in same hybrid ink?) and then the ink used to print the bath with NPCs. Authors must clarify what inks were used for each. Also, did authors print the bath or just cast/inject the NPC/hydrogel into the molding space? There does not seem to be a need for printing that bath material.

- Figure captions and presentation and elaboration of results in the text have some major issues and are hard to follow. See the several minor comments listed below for details.

- The flow biomechanics simulation and analyses presented in the Figure 5 and in the text are rather superficial and incomplete. The relevance of the predicted WSS to the biological levels in the brain vascular tissue should be discussed.

Minor Comments:

- In figure captions, make sure to define all the acronyms used in each figure. Many abbreviations that are used in the Figure 1 are not defined and explained in the text. Examples: MCC, BPC, CTC, BEC, CPF127, etc. Some of these are extremely important and critical for the reader to understand and follow the content of the article.

- Figure 1 caption is too short and does not really elaborate the highly complex schematics that are presented. Better annotation tools should be used to highlight each component of these schematics and should be described in the caption. Some details in the figure are impossible to read (too small).

- In describing the bioink compositions tested, "B" and "A" should be defined in "1.0B1.0A" and other labels. Sounds like authors meant to label BdECM and Alginate by B and A, but this must be defined and clarified.

- Figure 2: caption is vague and incomplete. Scale bars are missing in several panels. For panels (G and H), it is unclear what cell types are used there, and also there does not seem to be any red signal in neither of groups in panel (H), so the conclusion that one bioink group is 'optimal' for cell compatibility is not well supported.

- In most figure captions the scale bars are not defined.

- Figure 3(A) is missing the scale bar.

- Figure 5C-ii does not show error bars for the quantified data, and no statistical analysis. These should be added.

Response to Reviewer 1

Article ID NCOMMS-23-03632A
Title 3D Bioprinted Multilayered Cerebrovascular Conduits to Study Cancer Extravasation Mechanism Related with Vascular Geometry
Authors Wonbin Park, Jae Sung Lee, Ge Gao, Byoung Soo Kim, Dong-Woo Cho

Summary of response:

We appreciate the constructive comments of the reviewers. We carefully read the valuable comments from the reviewers, and revised the manuscript based on these comments. Please check our specific responses to the reviewer's comments and revisions made in the manuscript as detailed below. The answers to the comments are highlighted in blue, and the modifications in the revised manuscript are indicated with red font.

Reviewer's comment 1:

Park and co-worker demonstrated a mature three-layered cerebrovascular channel with erratic curvatures using in-bath 3D triaxial bioprinting technique and a brain-specific hybrid bioink containing an ionically crosslinkable hydrogel for studying the physical and molecular mechanisms of cancer extravasation in vitro. In this manuscript they showed that more tumor cells are arrested in the curved vascular channels and might cause under low velocity and wall shear stress accelerates the molecular signatures of metastatic potential. The authors claimed that the proposed in-bath 3D triaxial bioprinting technique in conjugation with hybrid bioink technological to develop erratic curvatures of vascular follow that driving brain metastases might help pharmaceutical and medical research.

Response 1:

We are very grateful for the valuable and thoughtful comments from reviewer #1 and have revised our manuscript accordingly.

Reviewer's comment 2:

Overall, this work done in this manuscript will improve the current 3D field in marginal level as there are so many other papers already showed similar approach where they described the 3D printing using dECM from many different organs including brain (<https://doi.org/10.1021/acsbiomaterials.9b01512>). Moreover, the authors claimed that their bioinks provided a structurally supportive environment for hollow structure. On contrary, there are reports to achieve the same using 3D bioprinting of heterogeneous bi- and tri-layered hollow channels within gel scaffolds using scalable multi-axial microfluidic extrusion nozzle (DOI

10.1088/1758-5090/aaf7c7). Considering the outcome of the present work it is hard for the reader to find a substantial improvement in the technological aspects where are currently available in the field.

Response 2:

We acknowledge the reviewer's valid concern regarding the technological advancement compared to the previously reported works of Seo et al. (2019) and Attalla et al. (2018). Nonetheless, our study represents a significant improvement over these previous works in several critical aspects.

Firstly, we have successfully enhanced the printability of tissue-derived decellularized extracellular matrix (dECM) by incorporating alginate, a biopolymer that complements the dECM. The utilization of a hybrid dECM bioink yielded remarkable results during the triaxial printing process. The rapid ionic crosslinking of the hybrid bioink with divalent cations (Ca^{2+}) derived from calcium-added pluronic F-127 played a key role in facilitating the fabrication of dimensionally complex vascular constructs. Of particular importance, this approach resulted in the creation of seamless constructs without any leakage at the curved portions of the tubes, as convincingly demonstrated in Figure 2F.

Furthermore, unlike the previous study that solely employed a material extrusion rate control approach, our novel dimension control strategy for fabricating multilayered tubular structures is based on a comprehensive study regulating the bioink extrusion rate, the diameter of printing nozzles, and the movement speed of the printhead, as illustrated in Figure 3B-F. This integrated approach allows us to directly fabricate multilayered conduits with various diameters and achieve precise control over the wall thickness of each layer. An essential achievement of this approach is the successful formation of extremely thin walls, a critical factor enabling the development of an endothelial monolayer. Through our research efforts, we were able to achieve a vascular wall thickness below 50 μm , facilitating the generation of a fully mature cerebrovascular endothelium, as convincingly demonstrated in Figure 4.

Building upon these significant technological advancements, we have successfully developed an in vitro brain blood vessel model that encompasses various curvatures. This cutting-edge platform has opened up new avenues for conducting multifaceted research aimed at understanding the biomechanical and molecular mechanisms associated with metastatic cancer development. Utilizing this platform for a parametric study, we were able to investigate the correlation between the degree of vascular curvatures and the changes in dissemination patterns of circulating tumor cells.

Considering the reviewer's comment, we have included the relevant contents and references in the revised manuscript as shown below.

Page 7, Line 250

Unfortunately, restoring the properties of the original tissues is difficult during the decellularization and solubilization process because of missing components including intercellular junctions and macromolecular chains. To compensate for this destruction, we utilized alginate and formulated a hybrid BdECM bioink to improve the fine-tuned shape fidelity of the constructs in a brain-tissue-specific microenvironment. As well as its biological and mechanical suitability for the cerebrovascular cells in our platform, its chemical reactivity that allowed the immediate ionic crosslinking between alginate and multivalent cations in CPF-127 for the elaboration of vessel curvatures with sharp angles. Utilizing the in-bath triaxial nozzle printing technique allowed the direct construction of MCCs with cellular complexity with four types of compartmentalized brain cells.

Page 8, Line 269

The impact of cerebrovascular curvatures on metastatic cancer extravasation could be investigated from multiple perspectives. While decades of biomechanical studies on metastatic cancer have primarily focused on physical entrapment in capillaries with a small diameter (<10 μm) and a low fluid velocity (< 1.5 mm s^{-1}), there is a growing need to investigate various hemodynamic factors affecting cancer dissemination^{39, 40}. To comprehend the multifactorial mechanisms involved, we established a physiologically relevant *in vitro* hemodynamic microenvironment that has the potential to facilitate tumor metastasis but no dominant biomechanical influencer. Instead, the variables under our control could accelerate the complex cascade.

Page 8, Line 284

The model replicated a series of metastatic cascades, including tumor circulation, adhesion, and extravasation, demonstrating the interplay of hemodynamic and molecular drivers in brain metastasis events. By bridging the gap between cellular biology and vascular biomechanics, our study offers new insights into how vascular architectural complexity impacts the spread of cancer cells.

Added References

[26] Attalla, R., Puersten, E., Jain, N. & Selvaganapathy, P. R. 3D bioprinting of heterogeneous bi-and tri-layered hollow channels within gel scaffolds using scalable multi-axial microfluidic extrusion nozzle. *Biofabrication* 11, 015012 (2018).

[33] Seo, Y. et al. Development of an anisotropically organized brain dECM hydrogel-based 3D neuronal culture platform for recapitulating the brain microenvironment *in vivo*. *ACS Biomaterials Science & Engineering* 6, 610-620 (2019).

[39] Ivanov, K., Kalinina, M. & Levkovich, Y. I. Blood flow velocity in capillaries of brain and muscles and its physiological significance. *Microvascular research* 22, 143-155 (1981).

[40] Perea Paizal, J., Au, S. H. & Bakal, C. Squeezing through the microcirculation: survival adaptations of circulating tumour cells to seed metastasis. *British Journal of Cancer* 124, 58-65 (2021).

Reviewer's comment 3:

Another philosophical question was not addressed here is that the "seed and soil" hypothesis where the influence in abluminal space is also important. In addition, the overall flow rate as well as oxygen concentration were not addressed.

Response 3

Considering the reviewer's comment, we have supplemented the related contents in the discussion

part with proper references. Please refer below for more details.

Page 2, Line 15

Meanwhile, according to seed-soil hypothesis, the intercellular communication with resident brain cells shapes the brain tropism of metastatic tumor cells^{11, 12}.

Page 1, Line 2

In the native brain, which constitutes only about 2% of the body's weight, it receives approximately 15–20% of the total blood supply in the body. This substantial blood flow allows for the delivery of about 49 mL of oxygen per minute, utilizing 750 mL of blood per minute^{2, 3}.

Page 8, Line 270

While decades of biomechanical studies on metastatic cancer have primarily focused on physical entrapment in capillaries with a small diameter (<10 μm) and a low fluid velocity (< 1.5 mm s^{-1}), there is a growing need to investigate various hemodynamic factors affecting cancer dissemination^{39, 40}.

Page 8, Line 276

In the human circulatory system, small blood vessels have fluid flow velocities ranging from 0.5 to 10 mm s^{-1} and WSS varying from 0 to 30 dynes cm^{-2} ^{222, 41, 42, 43, 44, 45}. Cancer adhesion mainly occurs in the low WSS range below 15 dynes cm^{-2} ⁴⁶.

Page 8, Line 281

A control group was created with straight vessels (0° curvature, a diameter of 700 μm) having a fluid velocity of 6 mm s^{-1} (corresponding to a flow rate of 2.31 $\text{mm}^3 \text{s}^{-1}$) and WSS of $\sim 0.6 \text{ dynes cm}^{-2}$. Subsequently, experimental groups were introduced with right-angle (90°) and obtuse-angle (150°) curves.

Reviewer's comment 4:

Figure 3E and 3F were not mentioned in results and discussion.

Response 4

In response to this valuable feedback, we have made the necessary revisions, as written below.

Page 4, Line 116

A curvilinear motion of the printhead, executed in response to G-code commands, generated vessel curvatures with a spatial frequency of 200–600 cycles m^{-1} (**Figure 3E**). Consequently, straight, narrowed, double-narrowed, and curved vasculatures with geometrical diversity were

created (Figure 3F).

Reviewer's comment 5:

Higher magnification of fluorescent images will substantiate the claims: mature endothelium, HBMECs formed mature vascular tissue, the progenitor cells differentiated into astrocytes and neurons over time.

Response 5

To verify the formation of the mature cerebrovascular tissue, fluorescent images with high magnification were added in the revised manuscript, as shown below.

Page 5, line 135

The encapsulated cells stretched their bodies and formed mature endothelium (Figure 4C, S8). Confluent CD31 and ZO-1 were detected in large, small, and curved models, demonstrating that the encapsulated HBMECs formed mature vascular tissue independent on the diameter and curvature (Figure 4D(i)). Furthermore, the progenitor cells differentiated into astrocytes and neurons over time, expressing GFAP, an astrocytic marker (Figure 4D(ii)), and Tuj1 and MAP2, a neural marker (Figure 4D(iii)). The three-cell structure formed a mature cerebral vasculature composed of four types of brain cells. In addition, vigorously stretched pericytes were observed using the filamentous actin (F-actin) marker and α -SMA, and *de novo* extracellular matrix (ECM) production of collagen IV and laminin was reproduced (Figure 4D(iv)).

Supplementary Figure S8

Figure S8. Tissue maturation in the *in vitro* brain metastasis model over 14 days. A) The triple layers of the cerebrovascular conduits have been maintained on day 14. The scale bar is 500 μ m.

B) The encapsulated cells have stretched their bodies, forming the mature endothelium, the confluent pericytes, and the differentiated neural progenitor cells into the astrocytes and the neurons. The brain endothelial cells, the brain pericytes, the neural progenitor cells, the astrocytes, and the nerve cells are indicated in the figure as BEC, BPC, NPC, AC, and NC, respectively. The scale bars are 50 μm .

Reviewer's comment 6:

Figure 4G and 4 H should be marked as 4E and 4F respectively.

Response 6

We thank the reviewer for pointing out this issue and made the changes, accordingly.

Reviewer's comment 7:

The authors should test diffusion permeability using curved structure developed with triple layered MCCs composed of human brain microvascular endothelial cells (HBMECs), human brain vascular pericytes (HBVPs), and NPCs.

Response 7

Following the suggestion, the diffusion permeability test was conducted. The related results and discussion were included in the revised manuscript as shown below.

Page 5, Line 150

Meanwhile, the straight and curved MCCs showed no significant difference in the fluorescence intensity level of the diffused permeability probes (**Figure S9**). However, the assessment of trans-epithelial electrical resistance (TEER) values demonstrated that the coculture condition significantly enhanced the barrier function of the cerebrovascular conduits (**Figure 4F, S10**). Furthermore, the coculture group showed relatively higher gene expression levels of endothelial markers including the tight junction makers (ZO-1, occluding), the adherens junction marker (VE-cadherin) and the endothelial marker (CD31), indicating that the synergistic interactions among brain cells enhanced the endothelial functions (**Figure 4G**)

Supplementary Figure S9

Figure S9. The relative average fluorescence intensity of the diffused permeability probes in the 3D bioprinted cerebrovascular conduits with various geometries. A) After the fluorescein isothiocyanate (FITC)-dextran particles are perfused, and the diffusion level is imaged, over time. B) The relative fluorescence intensity is increased over time, but there is no significant difference among the groups (0°, 90°, and 150°) (**, $p \leq 0.01$; ns, no significance).

Reviewer’s comment 8:

The authors should define CMM.

Response 8

We were astonished by our typo; it should have been MCCs. The comment was incorporated in the revised manuscript.

Page 5, Line 165

Therefore, MCCs were fabricated at three different angles (0°, straight angle; 90°, right angle; 150°, obtuse angle) and the tumor cells were perfused into the conduits for 24 h at an average velocity of 6 mm s^{-1} under laminar flow condition (Figure 5B).

Reviewer’s comment 9:

The authors should justify how these three different angles (0°, 90°, and 150°) were studied? Why the fluid velocity restricted to 6mm/min?

Response 9

First of all, we were astonished that the fluid velocity was expressed as 6 mm/min in the previous manuscript; it should have been 6 mm/s. We apologize for our typo.

The chosen fluid velocity of 6 mm/s was carefully selected to emulate the flow in the human

vascular system, as this flow condition is essential for demonstrating the correlation between vascular curvatures and cancer extravasation. To ensure accuracy and reliability, the fluid condition was established based on relevant and previously reported research.

Proper references have been included in the revised manuscript to support and justify this choice. We thank the reviewer for mentioning this potential issue. The related contents were also included in the revised manuscript.

Considering this valuable advice, we have incorporated the rationale for studying the angles and included relevant references in the revised manuscript, as written below.

Page 8, Line 269

The impact of cerebrovascular curvatures on metastatic cancer extravasation could be investigated from multiple perspectives. While decades of biomechanical studies on metastatic cancer have primarily focused on physical entrapment in capillaries with a small diameter ($<10\ \mu\text{m}$) and a low fluid velocity ($< 1.5\ \text{mm s}^{-1}$), there is a growing need to investigate various hemodynamic factors affecting cancer dissemination^{39, 40}. To comprehend the multifactorial mechanisms involved, we established a physiologically relevant *in vitro* hemodynamic microenvironment that has the potential to facilitate tumor metastasis but no dominant biomechanical influencer. Instead, the variables under our control could accelerate the complex cascade. In the human circulatory system, small blood vessels have fluid flow velocities ranging from 0.5 to $10\ \text{mm s}^{-1}$ and WSS varying from 0 to $30\ \text{dynes cm}^{-2}$ ^{22, 41, 42, 43, 44, 45}. Cancer adhesion mainly occurs in the low WSS range below $15\ \text{dynes cm}^{-2}$ ²⁴⁶. The adhesion efficiency is relatively high at $\text{WSS} < 0.5\ \text{dynes cm}^{-2}$ but decreases as WSS increases^{47, 48, 49, 50}. Taking these factors into account, our model was designed to investigate how various vascular curvatures influence cancer metastasis within human brain blood vessels. A control group was created with straight vessels (0° curvature, a diameter of $700\ \mu\text{m}$) having a fluid velocity of $6\ \text{mm s}^{-1}$ (corresponding to a flow rate of $2.31\ \text{mm}^3\ \text{s}^{-1}$) and WSS of $\sim 0.6\ \text{dynes cm}^{-2}$. Subsequently, experimental groups were introduced with right-angle (90°) and obtuse-angle (150°) curves. The model replicated a series of metastatic cascades, including tumor circulation, adhesion, and extravasation, demonstrating the interplay of hemodynamic and molecular drivers in brain metastasis events. By bridging the gap between cellular biology and vascular biomechanics, our study offers new insights into how vascular architectural complexity impacts the spread of cancer cells.

Reviewer's comment 10:

Controls for Figure 5D are missing. The authors should provide proteins expression in cases of three different angles without cancer cells.

Response 10

In response to the reviewer's comment, we included the control group and revised the manuscript,

as shown below.

Page 6, Line 173

To verify intercellular communication between tumor cells and brain endothelial cells, adhesion molecules were quantitatively analyzed (**Figure 5D**). After perfusion of CTCs, the expression of cell adhesion molecules (E-selectin, VCAM-1, and ICAM-1) increased, whereas a key marker for the barrier function of blood vessel walls (VE-cadherin) decreased. Significantly, the group of the larger MCC bending angles corresponded to higher measured molecular concentrations (**Table S2**), indicating the inflammatory response of brain endothelial cell.

Revised Figure 5D

Reviewer's comment 11:

It is not clearly articulated in the results, the angle for curved MCC in Figure 6A.

Response 11

It was added accordingly.

Page 6, Line 211

For comparative analysis of gene expression on mRNA level, four types of models were prepared: straight MCCs with endothelial cell monoculture (SE), straight MCCs with endothelial cell–stromal cell coculture (SES), 150°-bent MCCs with endothelial cell monoculture (CE), and 150°-

bent MCCs with endothelial cell–stromal cell coculture (CES).

Reviewer’s comment 12:

The authors should involve CTC from metastatic and non-metastatic tumors and present in Fig 5D, 6B-6E to prove their hypothesis.

Response 12

In response to this valuable advice, additional experiments were proceeded using three different types of tumor cells. We have revised the manuscript including the related contents, as written below.

Page 6, Line 183

Additionally, three different types of lung tumor cells including human lung circulating tumor cells, A549 cells, and H1299 cells were circulated to investigate the variations in metastatic behaviors based on the lung cancer cell types. However, there was no significant difference of the expression level of the adhesion proteins among the cells (**Figure S16**). However, to better understand the reasons for this phenomenon, a different approach was required.

Supplementary Figure S16

Figure S16. Expression of adhesion proteins in multilayered cerebrovascular conduits in response to the different types of cancer cells.

Page 7, Line 238

Additionally, A549 exhibited a similar expression pattern to that of CTCs in the key markers (**Figure S18**). However, more substantial differences in the expression levels of some genes (ICAM-1, MMP-2, MMP-9, IL-6, and Cx-43) were observed, indicating the potential influence of the cancer cell type on secondary tumor progression. Consequently, with comprehensive consideration, it could be concluded that cancer metastasis is a complex outcome, arising from the dynamic interplay between various mechanical and biological factors.

Supplementary Figure S18

Figure S18. Quantitative reverse-transcription PCR results demonstrate the effects of coculture and vascular curvatures on metastatic brain cancer development. A549 cells is circulated in four experimental groups: the straight multilayered cerebrovascular conduits (MCCs) with endothelial cell monoculture (SE), the straight MCCs with endothelial cell–stromal cell coculture (SES), the curved MCCs with endothelial cell monoculture (CE), and the curved MCCs with endothelial cell–stromal cell coculture (CES). The expression of key markers associated with B) endothelial barrier function (ICAM-1, ZO-1, and occludin), C) epithelial–mesenchymal transition (MMP-2, MMP-9, and TWIST), D) neuroinflammatory responses (TNF, IL-6, IL-8, and GFAP), and E) tumorigenesis (HGF, Cx-43) are analyzed (***, $p \leq 0.001$; **, $p \leq 0.01$; *, $p \leq 0.05$; ns, no significance).

Response to Reviewer 2

Article ID NCOMMS-23-03632A
Title 3D Bioprinted Multilayered Cerebrovascular Conduits to Study Cancer Extravasation Mechanism Related with Vascular Geometry
Authors Wonbin Park, Jae Sung Lee, Ge Gao, Byoung Soo Kim, Dong-Woo Cho

Summary of response:

We appreciate the constructive comments of the reviewers. We carefully read the valuable comments from the reviewers, and revised the manuscript based on these comments. Please check our specific responses to the reviewer's comments and revisions made in the manuscript as detailed below. The answers to the comments are highlighted in blue, and the modifications in the revised manuscript are indicated with red font.

Reviewer's comment 1:

Park et al. utilize 3D bioengineered models to analyze the biology of cells that comprise the neurovascular unit and regulate the BBB. In addition, mechanisms of adhesion and communication between neurovascular cells and arrested metastatic cells in the circulation and during the extravasation process are investigated. A particular focus is placed on the effect of curvature in the ex vivo cerebrovasculature and how this impacts BBB permeability as well as metastatic seeding and dissemination into the modeled brain parenchyma. The manuscript is well written. The scientific premise is solid and the data are presented in a rational manner. Overall, the results are of high quality. However, there are several significant conceptual and technical issues that limit enthusiasm for the study.

Response 1:

We are very grateful for the valuable and thoughtful comments from reviewer #2 and have revised our manuscript accordingly.

Reviewer's comment 2:

The major criticism of the manuscript is that, as presented, the data are largely proof-of-concept, confirmatory in nature, and incremental in impact. The data in the first four figures are largely illustrating/confirming the basic premise for design and use the 3D system. The results in Figures 4 and 5 include gene expression data for adhesion and communication between the tumor cells and the vascular endothelial cells, including analysis of effects of vessel curvature on circulating tumor cell arrest and extravasation. However, these cell adhesion molecules and endothelial junction proteins are already established as important for BBB integrity and cell rolling and arrest.

There are very few new findings that add novelty to the study. It is unclear how this 3D system is distinct from other similar published systems for the neurovascular unit.

Response 2:

We acknowledge the reviewer's concern. Nevertheless, we believe that this study represents a significant improvement over other published works in several aspects.

The successful recapitulation of the complex cerebrovascular tissue-specific microenvironment has allowed us to visualize and study the critical processes in the metastatic tumor formation, such as tumor cell adhesion and extravasation, which are often challenging to observe directly in the human brain. This capability holds significant implications for gaining insights into the mechanistic aspects of cancer metastasis in the brain.

Especially, our technology for fabricating the vascular conduits with different bend angles has facilitated a parametric analysis of the hemodynamic effects on cancer dissemination patterns. By shifting the recent research focus towards investigating the underlying mechanisms associated with the dissemination patterns of cancer cells based on the shape of blood vessels, this pioneering approach enables us to study what changes in vascular curvature impact cancer cell behavior, potentially leading to novel discoveries in the field of cancer research. Additionally, the ability to assess molecular cascades within our model highlights its potential as an advanced study platform for the discovery of new biomarkers and testing of therapeutic interventions.

Moreover, our technology also demonstrates potential for future research on various cerebrovascular diseases. Its versatility in fabricating cerebral vasculatures with various shapes, including curved, stenotic, and swollen shapes, highlights its potential applicability in other cerebrovascular disorders, such as aneurysms or atherosclerosis. Namely, our model holds promise for investigating the underlying pathological mechanisms of various diseases. Additionally, the automated fabrication process within a short timeframe demonstrates potential applicability for high-throughput studies.

We have revised the manuscript to include the several discussions to emphasize the strengths, potential applications, and future directions of our research. We believe that these additions enhance the overall contribution and significance of our study.

Page 8, Line 269

The impact of cerebrovascular curvatures on metastatic cancer extravasation could be investigated from multiple perspectives. While decades of biomechanical studies on metastatic cancer have primarily focused on physical entrapment in capillaries with a small diameter ($<10 \mu\text{m}$) and a low fluid velocity ($< 1.5 \text{ mm s}^{-1}$), there is a growing need to investigate various hemodynamic factors affecting cancer dissemination^{39, 40}. To comprehend the multifactorial mechanisms involved, we established a physiologically relevant *in vitro* hemodynamic microenvironment that has the potential to facilitate tumor metastasis but no dominant

biomechanical influencer. Instead, the variables under our control could accelerate the complex cascade.

Page 8, Line 284

The model replicated a series of metastatic cascades, including tumor circulation, adhesion, and extravasation, demonstrating the interplay of hemodynamic and molecular drivers in brain metastasis events. By bridging the gap between cellular biology and vascular biomechanics, our study offers new insights into how vascular architectural complexity impacts the spread of cancer cells.

Page 8, Line 289

Although molecules such as cellular adhesion proteins and endothelial growth factors have been linked to cancer dissemination for decades, there are currently no drugs or therapies targeting cancer extravasation. The model presented in this study may, therefore, hold promise for future biomarker discovery in cancer extravasation, and it has the potential to be applied for transcriptome analysis and high-resolution, real-time imaging in metastatic cancer research. Moreover, the model can be also utilized for the study of other cerebrovascular diseases, such as stroke, aneurysm, atherosclerosis, or even senescence, in which structural, mechanical, and biological changes can be a critical cause. Our technology's capability to fabricate the blood vessels with various structures, such as the curved, stenotic and swollen-shape, opens new possibilities for investigating these diseases. Additionally, the technology can be extended to incorporate diverse immune cell types, such as microglia and monocytes, allowing for a deeper exploration of the underlying mechanisms in these conditions^{28, 51, 52, 53}. Finally, this model has strong potential to provide not only a platform for real-time mechanism study of multiple brain diseases but also as a drug-testing platform for evaluation of the efficacy and safety of biopharmaceuticals.

Added References

[28] Gao, G. et al. Construction of a Novel In Vitro Atherosclerotic Model from Geometry-Tunable Artery Equivalents Engineered via In-Bath Coaxial Cell Printing. *Advanced Functional Materials* 31, 2008878 (2021).

[39] Ivanov, K., Kalinina, M. & Levkovich, Y. I. Blood flow velocity in capillaries of brain and muscles and its physiological significance. *Microvascular research* 22, 143-155 (1981).

[40] Perea Paizal, J., Au, S. H. & Bakal, C. Squeezing through the microcirculation: survival adaptations of circulating tumour cells to seed metastasis. *British Journal of Cancer* 124, 58-65 (2021).

[51] Gonzalez, N. R., Liebeskind, D. S., Dusick, J. R., Mayor, F. & Saver, J. Intracranial arterial stenoses: current viewpoints, novel approaches, and surgical perspectives. *Neurosurgical review*

36, 175-185 (2013).

[52] Rayz, V. L. & Cohen-Gadol, A. A. Hemodynamics of cerebral aneurysms: connecting medical imaging and biomechanical analysis. *Annual review of biomedical engineering* 22, 231-256 (2020).

[53] Bae, M. et al. Neural stem cell delivery using brain-derived tissue-specific bioink for recovering from traumatic brain injury. *Biofabrication* 13, 044110 (2021).

Reviewer's comment 3:

Figures 1 and 2

- The protein composition of the porcine BdECM remains unclear. The authors extract proteins from pieces of decellularized brain. Presumably they are working with ECM proteins from basement membranes as well as the interstitial matrix. However, they are using the bioink/algenate mixtures mostly as a vascular basement membrane model, although there are likely interstitial matrix components present. In Supp Fig. 1 they show GAGs, suggesting interstitial matrix, as well as collagens although the exact collagen composition is unclear. Also, there are likely growth factors and other bound proteins in the BdECM blend. A more detailed experimental characterization of the BdECM is important.

Response 3

To clarify the protein composition in BdECM, we performed the proteomics analysis, and the results and related contents were added in the revised manuscript. Please refer below for more details.

Page 3, Line 54

In addition, the in-depth study through a proteomics analysis of BdECM revealed over 2,000 types of proteins, including basement membrane components, interstitial matrix proteins, and growth factors. This suggests the potential to create a more biologically relevant microenvironment compared to other single materials (**Figure S2**).

Supplementary Figure S2

Figure S2. Protein composition of brain-derived decellularized extracellular matrix (BdECM). A) The pretoemic analysis of BdECM reveals the presence of over 2,000 proteins. B) The top 10 proteins are COL1A1 (17.29%), HSPG2 (7.34%), PLP1 (6.96%), COL4A2 (3.34%), COL1A2 (3.09%), LAMB2 (3.04%), SYN1 (2.90%), FN1 (2.67%), FBN1 (2.47%), and FGA (2.41%). C) Basement membrane components such as HSPG2, COL4A2, laminins (LAMB2 and LAMC1) and nidogen are detected. D) Interstitial matrix proteins including proteoglycans, tenascins, fibrous collagens, and adhesive glycoproteins are identified. E) More than 20 different collagen types, including collagen type I alpha I (COL1A1), collagen type IV alpha II (COL4A2), and collagen type I alpha II (COL1A2), are discovered and the components constitute over 30% of the total composition of BdECM.

Reviewer's comment 4:

Figure 3 and 4

- The authors use neural progenitor cells (presumably neurospheres) in the co-culture system. It is unclear why they do not use differentiated astrocytes, since these cells are important components of the neurovascular unit and regulate BBB development and physiology. The NPCs probably express basal levels of GFAP. Tuj1 is a marker for immature neurons, emphasizing the questionable physiological relevance of using NPCs in the system. As presented, this is more a model for the neural stem cell niche, which makes it distinct from a general model of the neurovascular unit.

Response 4

We acknowledge the reviewer's concern on the cell source used in this study. However, one of the main focuses in this study was to recapitulate the multilayered cerebrovascular conduits composed of various cells considering the main types of cells in native cerebral tissue. Under this platform, it is inevitable to contemplate coculture systems and technologies necessary for studying interaction between different cell types at *in vitro* models. Many efforts have been also made to optimize the co-culture conditions (references: 10.3389/fbioe.2020.00911, <https://doi.org/10.3390/biology10010006>).

To address these issues, our study selected neural progenitor cells as a promising candidate in the development of *in vitro* neurovascular unit model. Such cell sources have already demonstrated their physiological function instead of astrocytes; please refer to following articles (references: <https://doi.org/10.1186/2045-8118-10-2>, <https://doi.org/10.3390/ijms24043608>, <https://doi.org/10.1111/j.1471-4159.2011.07434.x>, and <https://doi.org/10.3389/fncel.2016.00215>). Furthermore, rather than the findings on co-culture conditions, our focus was to recapitulate the hemodynamic microenvironment to investigate the underlying biomechanical mechanisms in secondary tumor formation. Through this study, we could reveal the link among vascular anatomy, hemodynamic flow variance, and secondary tumor formation.

Furthermore, to verify the differentiation of the neural progenitor cells into astrocyte, the additional immunostaining analysis and qRT-PCR assessment were proceeded. As shown below, the expression level of GFAP was significantly upregulated in our system.

Furthermore, the staining results demonstrated the morphological changes and the increased population of GFAP⁺ cells, as shown below.

We also feel that the issue of cell source would make potential readers confused and therefore supplemented the related contents in the revised manuscript, as shown below. We hope for the reviewer's understanding on this experimental difficulty in coculture systems.

Page 8, Line 260

To develop a more physiologically relevant *in vitro* model representing native brain tissue, several factors have been considered. To recapitulate the cellular complexity of brain, coculture with more diverse range of cells is required. However, this raises the issues related to medium composition or cellular duration^{34, 35}. To address these issues, neural progenitor cells have been utilized, and their function in constructing artificial tissue equivalents has been verified^{36, 37, 38}. Based on these techniques utilizing neural progenitor cells, we have fabricated a multilayered cerebrovascular tissue containing four different cell types under a simple culture condition. Furthermore, the design of the crossing medium chamber in the platform supports hybrid perfusion systems including pump-based perfusion and pump-less perfusion, allowing high-throughput studies with better experimental efficiency.

Page 5, line 135

The encapsulated cells stretched their bodies and formed mature endothelium (**Figure 4C, S8**). Confluent CD31 and ZO-1 were detected in large, small, and curved models, demonstrating that the encapsulated HBMECs formed mature vascular tissue independent on the diameter and curvature (**Figure 4D(i)**). Furthermore, the progenitor cells differentiated into astrocytes and neurons over time, expressing GFAP, an astrocytic marker (**Figure 4D(ii)**), and Tuj1 and MAP2, a neural marker (**Figure 4D(iii)**). The three-cell structure formed a mature cerebral vasculature composed of four types of brain cells. In addition, vigorously stretched pericytes were observed using the filamentous actin (F-actin) marker and α -SMA, and *de novo* extracellular matrix (ECM) production of collagen IV and laminin was reproduced (**Figure 4D(iv)**).

Supplementary Figure S8

Figure S8. Tissue maturation in the *in vitro* brain metastasis model over 14 days. A) The triple layers of the cerebrovascular conduits have been maintained on day 14. The scale bar is 500 μm . B) The encapsulated cells have stretched their bodies, forming the mature endothelium, the confluent pericytes, and the differentiated neural progenitor cells into the astrocytes and the neurons. The brain endothelial cells, the brain pericytes, the neural progenitor cells, the astrocytes, and the nerve cells are indicated in the figure as BEC, BPC, NPC, AC, and NC, respectively. The scale bars are 50 μm .

Added References

- [34] Vis, M. A., Ito, K. & Hofmann, S. Impact of culture medium on cellular interactions in *in vitro* co-culture systems. *Frontiers in bioengineering and biotechnology* 8, 911 (2020).
- [35] Kuppusamy, P., Kim, D., Soundharajan, I., Hwang, I. & Choi, K. C. Adipose and muscle cell co-culture system: A novel *in vitro* tool to mimic the *in vivo* cellular environment. *Biology* 10, 6 (2020).
- [36] Lippmann, E. S., Al-Ahmad, A., Palecek, S. P. & Shusta, E. V. Modeling the blood–brain barrier using stem cell sources. *Fluids and Barriers of the CNS* 10, 1-14 (2013).
- [37] de Leeuw, V. C., van Oostrom, C. T., Zwart, E. P., Heusinkveld, H. J. & Hessel, E. V. Prolonged Differentiation of Neuron-Astrocyte Co-Cultures Results in Emergence of Dopaminergic Neurons. *International Journal of Molecular Sciences* 24, 3608 (2023).
- [38] Lippmann, E. S., Weidenfeller, C., Svendsen, C. N. & Shusta, E. V. Blood–brain barrier modeling with co-cultured neural progenitor cell-derived astrocytes and neurons. *Journal of neurochemistry* 119, 507-520 (2011).
-

Reviewer’s comment 5:

Figure 3 and 4

- In Figure 4E the authors show that single cultures of endothelial cells are sufficient to form a barrier to fluorescent tracers, bringing into question the functions of the additional mural cells (pericytes) and glial/neuronal cells. Perhaps an additional assay like TEER would add more insight. They do show that junctional proteins are elevated in the co-culture systems; however, it is uncertain why PECAM/CD31 is included since it is a marker for endothelial cells outside of the brain and is not directly linked to brain endothelial barrier integrity.

Response 5

In response to this valuable advice, we have measured the trans-epithelial electrical resistance (TEER) value of the multilayered cerebrovascular conduits, as shown below. It was also added as Figure 4F in the revised manuscript.

Furthermore, the brain endothelial cells used in this study possess a CD31-positive characteristic (Innoprot, Spain; <https://innoprot.com/product/human-brain-microvascular->

endothelial-cells/). Many research has demonstrated that CD31 is essential for the overall function of the endothelial cells (references: <https://doi.org/10.1038/s41598-019-42439-9>). Therefore, we used this marker as a tissue maturation indicator to demonstrate the endothelial cells were basically functioning in our platform. To avoid any potential confusion arising from the presented CD31 marker expression level in the graph, we modified the graph and figure legend to separately indicate the markers as the tight junction makers (ZO-1, occluding), the adherens junction marker (VE-cadherin) and the endothelial marker (CD31), as shown below.

Therefore, we revised the manuscript to include the related contents. Please refer below for more details.

Page 5, Line 151

However, the assessment of trans-epithelial electrical resistance (TEER) values demonstrated that the coculture condition significantly enhanced the barrier function of the cerebrovascular conduits (**Figure 4F, S10**).

Figure 4F

Page 20, Line 631

F, G) Coculture condition of endothelial cells with pericytes and glial/neural cells enhances trans-

epithelial electrical resistance (TEER) and elevated the expression of the the markers including the tight junction makers (ZO-1, occluding), the adherens junction marker (VE-cadherin) and the endothelial marker (CD31) (***, $p \leq 0.001$; **, $p \leq 0.01$; *, $p \leq 0.05$).

Supplementary Figure S10

Figure S10. Trans-epithelial electrical resistance (TEER) analysis of multilayered cerebrovascular conduits (MCCs). A) To verify the effects of coculture condition, electrodes are placed on either side of the mature MCCs constituted of brain endothelial cells (BECs), brain pericytes (BPCs), and neural progenitor cells (NPCs). B) The lowest TEER value is observed in the monoculture group containing only BECs (BEC⁺ BPC⁻ NPC⁻). When co-cultured with BPCs (BEC⁺ BPC⁺ NPC⁻), the TEER value increases. Furthermore, the TEER value increases even more when the differentiated NPCs are also included in the co-culture (BEC⁺ BPC⁺ NPC⁺) (***, $p \leq 0.001$).

Page 5, Line 153

Furthermore, the coculture group showed relatively higher gene expression levels of endothelial markers including the tight junction makers (ZO-1, occluding), the adherens junction marker (VE-cadherin) and the endothelial marker (CD31), indicating that the synergistic interactions among brain cells enhanced the endothelial functions (Figure 4G).

Revised Figure 4G

F, G) Coculture condition of endothelial cells with pericytes and glial/neural cells enhances trans-epithelial electrical resistance (TEER) and elevated the expression of the the markers including the tight junction makers (ZO-1, occluding), the adherens junction marker (VE-cadherin) and the endothelial marker (CD31) (***, $p \leq 0.001$; **, $p \leq 0.01$; *, $p \leq 0.05$).

Reviewer's comment 6:

Figure 3 and 4

- The authors mention use of pericytes in the co-culture system but contributions by these cells to BBB integrity and tumor cell extravasation are not sufficiently analyzed. There is insufficient inclusion of pericyte markers.

Response 6

To verify the effects of the presence of pericytes on the endothelial integration, additional experiments were conducted through trans-epithelial electrical resistance analysis (references: [10.1016/j.biocel.2011.05.002](https://doi.org/10.1016/j.biocel.2011.05.002). and <https://doi.org/10.1007/s10571-007-9195-4>). In addition, a pericyte marker (α -SMA) was used in additional assessments (references: <https://doi.org/10.1371/journal.pone.0150360> and <https://doi.org/10.1038/s41467-019-13896-7>). Please refer below for more details.

Page 5, Line 151

However, the assessment of trans-epithelial electrical resistance (TEER) values demonstrated that the coculture condition significantly enhanced the barrier function of the cerebrovascular conduits (Figure 4F, S10).

Figure 4F

Page 20, Line 631

F, G) Coculture condition of endothelial cells with pericytes and glial/neural cells enhances trans-epithelial electrical resistance (TEER) and elevated the expression of the the markers including the tight junction makers (ZO-1, occluding), the adherens junction marker (VE-cadherin) and the endothelial marker (CD31) (***, $p \leq 0.001$; **, $p \leq 0.01$; *, $p \leq 0.05$).

Supplementary Figure S10

Figure S10. Trans-epithelial electrical resistance (TEER) analysis of multilayered cerebrovascular conduits (MCCs). A) To verify the effects of coculture condition, electrodes are placed on either side of the mature MCCs constituted of brain endothelial cells (BECs), brain pericytes (BPCs), and neural progenitor cells (NPCs). B) The lowest TEER value is observed in the monoculture group containing only BECs ($\text{BEC}^+ \text{BPC}^- \text{NPC}^-$). When co-cultured with BPCs ($\text{BEC}^+ \text{BPC}^+ \text{NPC}^-$), the TEER value increases. Furthermore, the TEER value increases even more when the differentiated NPCs are also included in the co-culture ($\text{BEC}^+ \text{BPC}^+ \text{NPC}^+$) (***, $p \leq 0.001$).

Page 5, line 135

The encapsulated cells stretched their bodies and formed mature endothelium (**Figure 4C, S8**). Confluent CD31 and ZO-1 were detected in large, small, and curved models, demonstrating that the encapsulated HBMECs formed mature vascular tissue independent on the diameter and curvature (**Figure 4D(i)**). Furthermore, the progenitor cells differentiated into astrocytes and neurons over time, expressing GFAP, an astrocytic marker (**Figure 4D(ii)**), and Tuj1 and MAP2, a neural marker (**Figure 4D(iii)**). The three-cell structure formed a mature cerebral vasculature composed of four types of brain cells. In addition, vigorously stretched pericytes were observed using the filamentous actin (F-actin) marker and α -SMA, and *de novo* extracellular matrix (ECM) production of collagen IV and laminin was reproduced (**Figure 4D(iv)**).

Supplementary Figure S8

Figure S8. Tissue maturation in the *in vitro* brain metastasis model over 14 days. A) The triple layers of the cerebrovascular conduits have been maintained on day 14. The scale bar is 500 μm. B) The encapsulated cells have stretched their bodies, forming the mature endothelium, the confluent pericytes, and the differentiated neural progenitor cells into the astrocytes and the

neurons. The brain endothelial cells, the brain pericytes, the neural progenitor cells, the astrocytes, and the nerve cells are indicated in the figure as BEC, BPC, NPC, AC, and NC, respectively. The scale bars are 50 μm .

Reviewer's comment 7:

Figures 5 and 6

- The authors show that the tumor cells selectively home to bends in the vessel wall. How does this relate to the capillary network in the brain when a tumor cell arrests? Also, there are very few details about the metastatic tumor cells being utilized. More technical specifics should be included. Also, related to the fluorescent tracers in Figure 4E, are the curves/bends in the vessel wall correlative with increased endothelial barrier integrity and how does that relate to tumor cell arrest?

Response 7

Decades of studies have demonstrated that blood flow in the capillary network are also not constant, influenced by various factors such as vascular diameter and shape (reference: <https://doi.org/10.14814/phy2.14067>). Although our study focus on the metastatic progression in the curved microvascular arterio-venous blood vessels, the concept proposed in our research may also be applicable to the capillary network study. We revised the manuscript by adding the several discussions related to this idea.

Additionally, in response to the reviewer's comment, we supplemented the detailed information about the tumor cells.

Lastly, we have performed a perfusion test to verify the difference of endothelial integrity in the curved vascular walls but no significant difference among the groups has been observed. However, considering our results (Figure 5, 6), due to changes in hemodynamic factors may accelerate the cancer dissemination. The related contents have been supplemented as written below.

Page 8, Line 269

The impact of cerebrovascular curvatures on metastatic cancer extravasation could be investigated from multiple perspectives. While decades of biomechanical studies on metastatic cancer have primarily focused on physical entrapment in capillaries with a small diameter ($<10 \mu\text{m}$) and a low fluid velocity ($< 1.5 \text{ mm s}^{-1}$), there is a growing need to investigate various hemodynamic factors affecting cancer dissemination^{39, 40}. To comprehend the multifactorial mechanisms involved, we established a physiologically relevant *in vitro* hemodynamic microenvironment that has the potential to facilitate tumor metastasis but no dominant biomechanical influencer. Instead, the variables under our control could accelerate the complex cascade. In the human circulatory system, small blood vessels have fluid flow velocities ranging

from 0.5 to 10 mm s⁻¹ and WSS varying from 0 to 30 dynes cm⁻², 41, 42, 43, 44, 45.

Page 10, Line 356

The human lung circulating tumor cells (Celprogen, USA) were cultured using a Human Lung Circulating Tumor Cell Extracellular Matrix and a Human Lung Cancer Stem Cell Complete Growth Media according to the manufacturer's instructions. A549 and H1299 cells were cultivated in a growth media composed of RPMI1640 with L-glutamine (Hyclone, USA), 25mM HEPES (Sigma, USA), 25mM sodium bicarbonate (Sigma, USA), and 5% fetal bovine serum (Gibco, USA), following the protocol of Korean Cell Line Bank.

Page 5, Line 150

Meanwhile, the straight and curved MCCs showed no significant difference in the fluorescence intensity level of the diffused permeability probes (**Figure S9**). However, the assessment of trans-epithelial electrical resistance (TEER) values demonstrated that the coculture condition significantly enhanced the barrier function of the cerebrovascular conduits (**Figure 4F, S10**).

Supplementary Figure S9

Figure S9. The relative average fluorescence intensity of the diffused permeability probes in the 3D bioprinted cerebrovascular conduits with various geometries. A) After the fluorescein isothiocyanate (FITC)-dextran particles are perfused, and the diffusion level is imaged, over time. B) The relative fluorescence intensity is increased over time, but there is no significant difference among the groups (0°, 90°, and 150°) (**, $p \leq 0.01$; ns, no significance).

Reviewer's comment 8:

Figures 5 and 6

- The expression of several cell adhesion molecules (ICAM, VCAM, etc.) with known functions

in cancer and immune cell arrest in the circulation are analyzed. However, functional assays are not included. The authors should use function-blocking antibodies or gene silencing efforts in cancer cells to determine which pathways are important for arrest at the endothelial lumen. Also, including all the details for the cell adhesion molecule expression in the Results section is very distracting to the reader. These numbers should be shown in the figure or as supplemental data.

Response 8

Following the reviewer's comment, additional experiments using the function-blocking antibodies were proceeded and the results were supplemented in the revised manuscript.

Additionally, the details for the cell adhesion molecules has been changed as supplemental data.

Page 6, Line 178

Meanwhile, blocking the function of the cell adhesion molecules demonstrated that these surface proteins assist the tumor cell adhesion (**Figure S15**).

Supplementary Figure S15

Figure S15. Blocking cell adhesion molecules and cancer cell adhesion. A) Prior to treating cancer cells and the ICAM-1 function-blocking antibody (C⁻ B⁻), the activated signal of the endothelial cells was not detected. After 24 hours of treating cancer cells without the function-blocking antibody, some cancer cells adhered to the endothelium wall and elongated their bodies. However, when blocking the function of the adhesion molecules, a relatively small number of the

adhered cancer cells with round-shaped bodies was observed. B) In the case of VCAM-1, blocking the adhesion molecules suppressed the tumor metastatic progression.

Supplementary Table S2

Target protein (pg mL ⁻¹)		E-selectin	VCAM-1	ICAM-1	VE-Cadherin
CTC ⁻	0°	3.54 ± 0.31	11.41 ± 2.13	76.55 ± 10.38	553.56 ± 12.09
	90°	3.39 ± 0.56	10.64 ± 3.09	85.16 ± 3.18	540.36 ± 168.89
	150°	3.273 ± 0.23	11.96 ± 1.74	82.81 ± 6.68	574.86 ± 109.44
CTC ⁺	0°	3.33 ± 0.34	14.73 ± 0.62	79.79 ± 5.87	489.17 ± 54.72
	90°	5.42 ± 0.50	17.00 ± 1.81	144.21 ± 12.31	368.24 ± 11.97
	150°	6.71 ± 0.59	21.83 ± 1.25	174.03 ± 19.14	266.86 ± 117.13

Supplementary Table S2. Protein expression level of cell adhesion molecules.

Reviewer's comment 9:

Figures 5 and 6

- The authors show that tumor cells induce a down-regulation in various endothelial cell junction proteins an increase in neuroinflammatory molecules. What does this mean functionally? Is BBB integrity impacted in their system? What new and important mechanistic information could be gained from this system, rather than simply confirming what is known? For example, is the system amenable to transcriptome analysis or high-resolution, real time imaging to analyze the steps in tumor cell extravasation?

Response 9

We appreciate the reviewer for the constructive comments. Our study demonstrated that the circulating tumor cells adhered and interacted with brain cells to form the secondary tumor in vitro by impacting endothelial integrity.

With comprehensive consideration of our results, we could conclude that cancer metastasis is a complex outcome, arising from the dynamic interplay between various mechanical and biological factors.

Furthermore, we have revealed the link among vascular anatomy, hemodynamic flow variance, and secondary tumor formation. As the reviewer mentioned, the model has potential to be applied for transcriptome analysis and high-resolution, real time imaging in metastatic cancer

research, as well.

Moreover, our technology showed the potential to be utilized for a high-throughput fabrication process because the automated printing process could create the entire construct within a short timeframe of less than 10 minutes by simply activating the printing code.

On the other hand, the versatility, which enables to fabricate the cerebral vasculatures with various shapes including the curved, stenotic, and swollen shapes, highlights the potential applicability of our technique in other cerebrovascular disorders, such as aneurism or atherosclerosis (references: <https://doi.org/10.1007/s10143-012-0432-z> and <https://doi.org/10.1146/annurev-bioeng-092419-061429>). Thus, in the future, our model holds promise for exploring the underlying pathological mechanisms of various diseases.

With this regard, we include several discussions about the potential and future works. Please refer below for more details.

Page 8, Line 284

The model replicated a series of metastatic cascades, including tumor circulation, adhesion, and extravasation, demonstrating the interplay of hemodynamic and molecular drivers in brain metastasis events. By bridging the gap between cellular biology and vascular biomechanics, our study offers new insights into how vascular architectural complexity impacts the spread of cancer cells.

Page 8, Line 260

To develop a more physiologically relevant *in vitro* model representing native brain tissue, several factors have been considered. To recapitulate the cellular complexity of brain, coculture with more diverse range of cells is required. However, this raises the issues related to medium composition or cellular duration^{34, 35}. To address these issues, neural progenitor cells have been utilized, and their function in constructing artificial tissue equivalents has been verified^{36, 37, 38}. Based on these techniques utilizing neural progenitor cells, we have fabricated a multilayered cerebrovascular tissue containing four different cell types under a simple culture condition. Furthermore, the design of the crossing medium chamber in the platform supports hybrid perfusion systems including pump-based perfusion and pump-less perfusion, allowing high-throughput studies with better experimental efficiency.

Page 8, Line 289

Although molecules such as cellular adhesion proteins and endothelial growth factors have been linked to cancer dissemination for decades, there are currently no drugs or therapies targeting cancer extravasation. The model presented in this study may, therefore, hold promise for future biomarker discovery in cancer extravasation, and it has the potential to be applied for transcriptome analysis and high-resolution, real-time imaging in metastatic cancer research.

Moreover, the model can be also utilized for the study of other cerebrovascular diseases, such as stroke, aneurysm, atherosclerosis, or even senescence, in which structural, mechanical, and biological changes can be a critical cause. Our technology's capability to fabricate the blood vessels with various structures, such as the curved, stenotic and swollen-shape, opens new possibilities for investigating these diseases. Additionally, the technology can be extended to incorporate diverse immune cell types, such as microglia and monocytes, allowing for a deeper exploration of the underlying mechanisms in these conditions^{28, 51, 52, 53}. Finally, this model has strong potential to provide not only a platform for real-time mechanism study of multiple brain diseases but also as a drug-testing platform for evaluation of the efficacy and safety of biopharmaceuticals.

Response to Reviewer 3

Article ID NCOMMS-23-03632A
Title 3D Bioprinted Multilayered Cerebrovascular Conduits to Study Cancer Extravasation Mechanism Related with Vascular Geometry
Authors Wonbin Park, Jae Sung Lee, Ge Gao, Byoung Soo Kim, Dong-Woo Cho

Summary of response:

We appreciate the constructive comments of the reviewers. We carefully read the valuable comments from the reviewers, and revised the manuscript based on these comments. Please check our specific responses to the reviewer's comments and revisions made in the manuscript as detailed below. The answers to the comments are highlighted in **blue**, and the modifications in the revised manuscript are indicated with **red** font.

Reviewer's comment 1:

The authors utilized a direct 3D bioprinting strategy to construct multilayered cerebrovascular conduits (MCCs) with varying geometries and curvatures to study brain metastasis. First, a hybrid BdECM bioink formulation was optimized for printing fidelity and cell viability. An in vitro brain metastasis model was constructed using triple layered MCCs consisting of HBMECs (endothelial), HBVPs (pericytes) and NPCs (neural progenitor cells) with perfusable and geometrically tunable blood vessels. Using their platform, the authors observed that vascular curvature results in the upregulation of endothelial adhesion molecules and downregulation of endothelial tight junction markers, as well as the upregulation of EMT and tumorigenesis markers in cancer cells. Ultimately, this platform has the potential to be used as a preclinical model for studying the molecular mechanisms involving brain metastases.

Response 1:

We are very grateful for the valuable and thoughtful comments from reviewer #3 and have revised our manuscript accordingly.

Reviewer's comment 2:

The authors state that most tissue engineering strategies still rely on complex techniques to fabricate advanced platforms to study brain vasculature. However, the fabrication process involved in this study still appears quite rigorous. How does this platform compare with advanced 3D microfluidic models in terms of ease of use and technical robustness? Can devices be more reliably established here? How this platform differs from others in this regard should be further specified.

Response 2:

We sincerely appreciate the reviewer for their constructive comments, which have significantly contributed to improving the quality of our work. In our fabrication strategy, we utilized a 3D bioprinter with multiple printheads, enabling the automatic construction of a multilayered cerebrovascular model with multiple cell compositions. This approach allowed us to elaborate the complex brain microenvironment *in vitro* effectively.

By simply activating the printing code, the entire construct can be created within a short timeframe of less than 10 minutes. This remarkable productivity in the *in vitro* biomedical platform development sets our method apart from traditional operator-dependent tissue engineering strategies, such as the soft lithography technique. These approaches often involve the fabrication of molds, casting PDMS for each layer, treating plasma, bonding the parts, and seeding cells, which can be time-consuming and labor-intensive (references: <https://doi.org/10.1021/ar300314s>, and <https://doi.org/10.3390/mi12030319>). In contrast, our approach allows us to produce models with consistent and standardized structures efficiently.

Additionally, the design of the chip with a crossing medium chamber provides flexibility in perfusion methods. We can employ both pump-based perfusion and pump-less perfusion. The pump-less approach allows us to eliminate the sacrificial material and spontaneously provide specific medium for each type of cells. After stabilization, the model can be easily connected to a pump for continuous perfusion, further enhancing the efficiency and manageability of the model. With these refinements, the multi-cellular and multilayered cerebrovascular conduits with various curvatures could be fabricated.

We revised our manuscript to include the related contents, highlighting the advantages and innovations of our fabrication strategy in developing a complex *in vitro* cerebrovascular model. Please refer below for more details.

Page 7, Line 247

By simply activating the printing code, the entire construct can be created within a remarkably short timeframe of less than 10 minutes.

Page 8, Line 260

To develop a more physiologically relevant *in vitro* model representing native brain tissue, several factors have been considered. To recapitulate the cellular complexity of brain, coculture with more diverse range of cells is required. However, this raises the issues related to medium composition or cellular duration^{34, 35}. To address these issues, neural progenitor cells have been utilized, and their function in constructing artificial tissue equivalents has been verified^{36, 37, 38}. Based on these techniques utilizing neural progenitor cells, we have fabricated a multilayered cerebrovascular tissue containing four different cell types under a simple culture condition. Furthermore, the

design of the crossing medium chamber in the platform supports hybrid perfusion systems including pump-based perfusion and pump-less perfusion, allowing high-throughput studies with better experimental efficiency.

Page 10, Line 362

The brain metastasis model has been automatically fabricated by utilizing our in-house built hybrid 3D bioprinting system (ICBS)⁵⁴ in an isolated 3D printing dedicated room. The internal temperature of the room is maintained at 18°C to inhibit the crosslinking of the printed hydrogel, while the humidity is kept at 40-50%.

Added References

[34] Vis, M. A., Ito, K. & Hofmann, S. Impact of culture medium on cellular interactions in in vitro co-culture systems. *Frontiers in bioengineering and biotechnology* 8, 911 (2020).

[35] Kuppusamy, P., Kim, D., Soundharajan, I., Hwang, I. & Choi, K. C. Adipose and muscle cell co-culture system: A novel in vitro tool to mimic the in vivo cellular environment. *Biology* 10, 6 (2020).

[36] Lippmann, E. S., Al-Ahmad, A., Palecek, S. P. & Shusta, E. V. Modeling the blood–brain barrier using stem cell sources. *Fluids and Barriers of the CNS* 10, 1-14 (2013).

[37] de Leeuw, V. C., van Oostrom, C. T., Zwart, E. P., Heusinkveld, H. J. & Hessel, E. V. Prolonged Differentiation of Neuron-Astrocyte Co-Cultures Results in Emergence of Dopaminergic Neurons. *International Journal of Molecular Sciences* 24, 3608 (2023).

[38] Lippmann, E. S., Weidenfeller, C., Svendsen, C. N. & Shusta, E. V. Blood–brain barrier modeling with co-cultured neural progenitor cell-derived astrocytes and neurons. *Journal of neurochemistry* 119, 507-520 (2011).

Reviewer’s comment 3:

The authors stated that their model can recapitulate the cerebral mechanical microenvironment but should compare the mechanical/rheological properties of their platform to that of native tissue if possible in order to come to their conclusion.

Response 3

Although we examined the rheological property of our material (brain-derived decellularized extracellular matrix-based hydrogel), our focus was on creating the cell-friendly microenvironment, rather than mimicking the strength of the native brain tissue.

Considering the reviewer’s comment, the rheological and mechanical properties of the native brain tissue and the crosslinked cell-laden bioink were compared.

We assume that the different mechanical properties between the two systems may arise from variations in the cell density or protein concentration within them.

Additionally, the results were supplemented in the revised manuscript. Please refer below for more details.

Supplementary Figure S5

Figure S5. Comparison of mechanical and rheological properties of native brain tissue and hybrid brain-derived extracellular matrix (BdECM) bioink. A) Mechanical and rheological properties of the native tissue and the multilayered cerebrovascular conduits (MCCs) were measured. B) The compressive modulus of the native tissue and MCCs were 118.28 ± 6.16 Pa and 85.26 ± 11.97 Pa, respectively. C) The storage modulus (G') and loss modulus (G'') of the native tissue were 3104.62 ± 1113.31 Pa and 729.72 ± 379.71 Pa, respectively, while those of MCCs were 1377.51 ± 1031.49 Pa and 266.52 ± 1031.49 Pa, respectively.

Page 4, Line 95

In addition, the cell-laden hybrid BdECM showed lower physical strength than native brain tissue. It might result from the gap in cell density of two systems (Figure S5).

Page 9, Line 339

Rheological and mechanical characterization. The rheological and mechanical performance of BdECM-based bioinks and native brain tissue were validated using an Advanced Rheometric Expansion System (TA Instruments, USA).

Page 10, Line 347

Compressive modulus was measured under shear strain from 0.1 to 1000% at 15 °C (n = 3).

Reviewer's comment 4:

(I could be missing something here but...) The cancer cell line/type as well as the methods to maintain cancer cells are not specified anywhere in the manuscript. Do these cancer cells have an invasive or non-invasive phenotype? What cancer cells (and from what organ) are being used in the metastasis model? Are these patient-derived circulating tumour cells?

Response 4

The detailed information related to the cancer cells was supplemented in the revised manuscript. The human lung circulating tumor purchased from a company (Celprogen, USA), A549 lung cancer cells, and H1299 lung cancer cells were used in this study. The cells have the invasive properties (references: <https://celprogen.com/human-lung-circulating-tumor-cells-frozen-vial/>, <https://doi.org/10.1016/j.bgm.2014.07.002>, and <https://doi.org/10.1002/fsn3.1439>).

Considering the comment, we revised the manuscript to include the related information. Please refer below for more details.

Page 10, Line 356

The human lung circulating tumor cells (Celprogen, USA) were cultured using a Human Lung Circulating Tumor Cell Extracellular Matrix and a Human Lung Cancer Stem Cell Complete Growth Media according to the manufacturer's instructions. A549 and H1299 cells were cultivated in a growth media composed of RPMI1640 with L-glutamine (Hyclone, USA), 25mM HEPES (Sigma, USA), 25mM sodium bicarbonate (Sigma, USA), and 5% fetal bovine serum (Gibco, USA), following the protocol of Korean Cell Line Bank.

Reviewer's comment 5:

(related to the previous comment) With regards to the cancer cell type being used for these studies, does the model capture the behaviour of invasive/non invasive cancer cell phenotypes?

Response 5

We capture the invasive behavior of the cancer cells in our model. Please refer below for more details.

Page 5, Line 168

The introduced metastatic cells adhered to the endothelial monolayer after 3 hours, extravasated from the vasculature, and invaded toward the extracellular matrix, causing slight vasoconstriction

(Figure S11-S14).

Supplementary Figure S11

Figure S11. Tumor cell adhesion in multilayered cerebrovascular conduits (MCCs) with different angles. On the confluent endothelium, cancer cells adhere to initiate the metastasis.

Supplementary Figure S12

Figure S12. Tumor cell adhesion on the confluent cerebrovascular wall. The tumor cells preferentially adhere to the endothelial wall at the curved portion. The scale bars are 500 μm (left) and 50 μm (right).

Supplementary Figure S13

Figure S13. Extravasation of introduced tumor cells in multilayered cerebrovascular conduits over time.

Supplementary Figure S14

Figure S14. Tumor cell extravasation from a multilayered cerebrovascular conduit (MCC). The tumor cells in metastatic stages including adhesion, extravasation, and colonization were observed in the MCC.

Reviewer's comment 6:

The authors demonstrate that changing the vessel curvature influences hemodynamic factors such as wall shear stress and fluid velocity gradients. Do/would these changes in curvature and fluid profile directly affect cancer cell EMT? That is, is cancer cell EMT directly controlled by the expression endothelial cell markers or do mechanical factors, or the mechanical environment play a role in this as well?

Response 6

We considered that the interplay of biomechanical factors, including wall shear stress and fluid velocity, along with biological factors, including cellular interaction, is important in cancer metastasis.

Additional experiments using function-blocking antibodies against cell adhesion molecules showed that the communication between tumor cells and endothelial cells is one of the crucial factors in cancer adhesion and extravasation, as shown below.

Additionally, taking this point into consideration, we revised the results and discussion sections of the manuscript. Please refer to the details below for more information

Page 6, Line 178

Meanwhile, blocking the function of the cell adhesion molecules demonstrated that these surface proteins assist the tumor cell adhesion (**Figure S15**).

Supplementary figure S15

Figure S15. Blocking cell adhesion molecules and cancer cell adhesion. A) Prior to treating cancer cells and the ICAM-1 function-blocking antibody (C⁻ B⁻), the activated signal of the endothelial cells was not detected. After 24 hours of treating cancer cells without the function-blocking antibody, some cancer cells adhered to the endothelium wall and elongated their bodies. However, when blocking the function of the adhesion molecules, a relatively small number of the adhered cancer cells with round-shaped bodies was observed. B) In the case of VCAM-1, blocking the adhesion molecules suppressed the tumor metastatic progression.

Page 6, Line 203

These findings imply the high possibility that various biomechanical and cellular factors can influence cancer metastasis through the prolongation of the residence time of tumor cells.

Page 8, Line 284

The model replicated a series of metastatic cascades, including tumor circulation, adhesion, and extravasation, demonstrating the interplay of hemodynamic and molecular drivers in brain metastasis events. By bridging the gap between cellular biology and vascular biomechanics, our study offers new insights into how vascular architectural complexity impacts the spread of cancer cells.

Reviewer's comment 7:

From fluid mechanics point of view, it is easy to understand the velocity close to convex side is higher compared to velocity close to the concave side. However, as a result of the velocity gradient, given the conservation of mechanical energy, the pressure gradient should be the opposite, which is higher pressure close to the concave side and lower pressure to the convex side, which intuitively should “push” cells towards the convex side, which seems contradictory to what have been observed here. Has the authors quantify/simulate the pressure gradient close to the curved area? How the pressure gradient, shear stress play a role on regulating cancer metastasis?

Response 7

We thank the reviewer for the constructive comments. Under our experimental conditions (fluid velocity, 6 mm s⁻¹; Re, 0~ 4.516), wall shear stress was lower on the concave side. This was due to the Newtonian fluid (cell culture medium) causing reduced contact between the concave side wall and the fluid compared to the convex side, resulting in lower wall shear stress (reference: <https://doi.org/10.20964/2018.09.56> and <https://doi.org/10.3390/fluids6110378>). However, if the size of the curvature was changed, the opposite gradient could be observed, as shown below.

Additionally, in response to the reviewer's suggestion, the related further experiments were proceeded, and the related contents have been supplemented, as shown below.

Supplementary figure S17

Figure S17. Fluid dynamics simulation of pressure in the metastatic cancer model.

Page 6, Line 196

The WSS value towards the center of the curvature was higher than the opposite side. In particular, the range in which the WSS was less than 60 mPa expanded in the curved groups (**Figure 5F(ii)**). In addition, a relatively slow pressure change was formed in the opposite direction of the center of the curvature. (**Figure S17**). This implies that the tumor cells were exposed to the specific pressure level for the extended period. For MCCs with different bending degrees but the same inner diameter and same blood flow velocity, greater curvature may result in the regions of the low shear stress and velocity and the prolonged such pressure, promoting metastasis. These findings imply the high possibility that various biomechanical and cellular factors can influence cancer metastasis through the prolongation of the residence time of tumor cells.

Reviewer's comment 8:

Cancer cell EMT adhesion markers (cadherins) also play a significant role especially in the context of adhesion/extravasation/metastasis. Are these factors also upregulated in addition to the ones already presented (MMP-2, MMP-9, TWIST)?

Response 8

In response to the reviewer's suggestion, the related qRT-PCR experiment was proceeded, and we could obtain the following result.

Although the expression level of N-cadherin was different in the experimental groups (straight MCCs with endothelial cell monoculture (SE), straight MCCs with endothelial cell–stromal cell coculture (SES), 150°-bent MCCs with endothelial cell monoculture (CE), and 150°-bent MCCs with endothelial cell–stromal cell coculture (CES)), there was no significant difference.

Reviewer's comment 9:

The authors should specify how many repeats were performed for each experiment.

Response 9

All the experiments were triplicated. It was marked throughout the revised manuscript

Reviewer's comment 10:

An experimental timeline including all aspects of the experiment involving cancer cells and endothelial cells would be nice to include to allow readers to visualize the procedures performed.

Response 10

In response to the reviewer's suggestion, the experimental timeline was added, as shown below.

Page 5, Line 131

Fully mature MCCs were obtained after 14 days of culture (**Figure S6**).

Supplemented figure S6

Figure S6. The experimental timeline for metastatic cancer progression study. After in-bath 3D triaxial bioprinting of multilayered cerebrovascular conduits (MCCs) using the brain endothelial cells (BECs), brain pericytes (BPCs), and neural progenitor cells (NPCs), the constructs were stabilized and cultured. Subsequently, after the MCCs mature for 14 days, circulating tumor cells are perfused into MCCs to investigate the hemodynamic effects on metastatic cancer progression. On day 1, 7, 14, and 15, the samples were harvested and the functionality of the model for mechanism study is verified.

Reviewer's comment 11:

What environment (temp., humidity(?)) is the bioprinting taking place? Is this performed at room temp. in sterile conditions (that is, in a biosafety cabinet)? Conditions should be specified.

Response 11

In response to the reviewer's suggestion, the Method part in this manuscript was revised to elaborate the printing environments, as written below.

Page 10, Line 362

Construction of *in vitro* brain metastasis model. The brain metastasis model has been automatically fabricated by utilizing our in-house built hybrid 3D bioprinting system (ICBS)⁵⁴ in an isolated 3D printing dedicated room. The internal temperature of the room is maintained at 18°C to inhibit the crosslinking of the printed hydrogel, while the humidity is kept at 40-50%.

Reviewer's comment 12:

Figure 5C(ii.) – Statistical significance between groups in this figure is missing.

Response 12

It was edited as shown below.

Revised Figure 5C(ii)

C) With the increase in the vascular bending angle, larger number of invading tumor cells are detected (***, $p \leq 0.001$; **, $p \leq 0.01$; *, $p \leq 0.05$; ns, no significance).

Reviewer's comment 13:

The authors mention that this model could be used to study other cerebrovascular conditions such as stroke or aneurysm. How would aspects of these diseases be incorporated into the platform?

Response 13

We sincerely appreciate the valuable feedback from the reviewer. As shown in Figure 3C and 3F, our technology allows for the construction of vascular conduits with high geometrical diversity. By controlling the printhead velocity, the diameter of the tubes can be easily adjusted, enabling the fabrication of cerebral vasculatures with stenotic and swollen shapes. This versatility highlights the potential applicability of our model in studying brain metastasis and other cerebrovascular disorders.

In recent years, there has been increasing recognition of the importance of developing in vitro platforms to study the interplay between biomechanical and cellular factors in the development of vascular diseases, such as stroke and aneurysms, for which suitable models have

been lacking (references: <https://doi.org/10.1007/s10143-012-0432-z> and <https://doi.org/10.1146/annurev-bioeng-092419-061429>). Our technology's capability to fabricate stenotic and swollen-shaped vasculature opens new possibilities for investigating these diseases. Moreover, the technology can be extended to incorporate diverse immune cell types, such as microglia and monocytes, allowing for a deeper exploration of the underlying biological mechanisms in these conditions. Our previous research has already demonstrated the potential use of these cells in the platform (references: <https://doi.org/10.1088/1758-5090/ac293f> and <https://doi.org/10.1002/adfm.202008878>). Thus, in the future, our model holds promise for exploring the molecular mechanisms underlying conditions like stroke and aneurysms.

We added the related contents in the revised manuscript, as shown below.

Page 8, Line 293

Moreover, the model can be also utilized for the study of other cerebrovascular diseases, such as stroke, aneurysm, atherosclerosis, or even senescence, in which structural, mechanical, and biological changes can be a critical cause. Our technology's capability to fabricate the blood vessels with various structures, such as the curved, stenotic and swollen-shape, opens new possibilities for investigating these diseases. Additionally, the technology can be extended to incorporate diverse immune cell types, such as microglia and monocytes, allowing for a deeper exploration of the underlying mechanisms in these conditions^{28, 51, 52, 53}.

Added References

[28] Gao, G. et al. Construction of a Novel In Vitro Atherosclerotic Model from Geometry-Tunable Artery Equivalents Engineered via In-Bath Coaxial Cell Printing. *Advanced Functional Materials* 31, 2008878 (2021).

[51] Gonzalez, N. R., Liebeskind, D. S., Dusick, J. R., Mayor, F. & Saver, J. Intracranial arterial stenoses: current viewpoints, novel approaches, and surgical perspectives. *Neurosurgical review* 36, 175-185 (2013).

[52] Rayz, V. L. & Cohen-Gadol, A. A. Hemodynamics of cerebral aneurysms: connecting medical imaging and biomechanical analysis. *Annual review of biomedical engineering* 22, 231-256 (2020).

[53] Bae, M. et al. Neural stem cell delivery using brain-derived tissue-specific bioink for recovering from traumatic brain injury. *Biofabrication* 13, 044110 (2021).

Reviewer's comment 14:

The implications of the shear thinning behaviour is not clear, why it is important and how that compared to native ECM need to be provided.

Response 14

To ensure the printability and fidelity of the bioink, the shear-thinning behavior of the brain-derived decellularized extracellular matrix (BdECM) pre-gel solution was investigated. This property is crucial for the bioinks for the application to the extrusion-based 3D bioprinting (reference: <https://doi.org/10.1021/acs.chemrev.0c00084>).

Both BdECM-based materials, including the pure BdECM hydrogel and hybrid BdECM bioink, showed proper rheological properties. However, we selected the hybrid BdECM bioink considering its rapid crosslinking capacity, as shown in Figure 2F.

Although native ECM is soft, it exists in a solid state. Therefore, additional assessments were conducted to compare mechanical properties of the native brain tissue and the cell-laden hybrid BdECM. Please refer below for more details.

Supplementary Figure S5

Figure S5. Comparison of mechanical and rheological properties of native brain tissue and hybrid brain-derived extracellular matrix (BdECM) bioink. A) Mechanical and rheological properties of the native tissue and the multilayered cerebrovascular conduits (MCCs) were measured. B) The compressive modulus of the native tissue and MCCs were 118.28 ± 6.16 Pa and 85.26 ± 11.97 Pa, respectively. C) The storage modulus (G') and loss modulus (G'') of the native tissue were 3104.62 ± 1113.31 Pa and 729.72 ± 379.71 Pa, respectively, while those of MCCs were 1377.51 ± 1031.49 Pa and 266.52 ± 1031.49 Pa, respectively.

Page 4, Line 95

In addition, the cell-laden hybrid BdECM showed lower physical strength than native brain tissue. It might result from the gap in cell density of two systems (**Figure S5**).

Page 9, Line 339

Rheological and mechanical characterization. The rheological and mechanical performance of BdECM-based bioinks and native brain tissue were validated using an Advanced Rheometric Expansion System (TA Instruments, USA).

Page 10, line 347

Compressive modulus was measured under shear strain from 0.1 to 1000% at 15 °C (n = 3).

Reviewer's comment 15:

Many figures, for example, Figure 2C, 2D, 5D, etc., are poor quality, with very blur images of the curve and the legends, leading to great difficulty to understand the figures.

Response 15

In response to the reviewer's suggestion, the figures and figure legends were to improve readability. Please refer below for more details.

Revised Figure 2C

C) stable sol-gel transition with clear flow points, where the storage modulus (G') and loss modulus (G'') reverse at the threshold shear stress, while 0.5B bioink exhibits fluidic-dominant property.

Page 3, Line 70

A complex modulus assay demonstrated that all groups, except 0.5B, displayed clear yield and

flow points marking the transition from the solid plateau region to the fluid region dependent on the increased strain at 15 °C (Figure 2C, S3A).

Supplementary Figure S3A

Figure S3. Rheological property of brain-derived decellularized extracellular matrix (BdECM) pre-gel. A) The stable sol-gel transition behavior of BdECM pre-gel (G' and G'' represent storage modulus and loss modulus, respectively). B) Thermal gelation kinetics of the BdECM pre-gel.

Revised Figure 2D

D) BdECM-based bioinks including (i) pure BdECM bioink (1.0B) and hybrid BdECM bioink (1.0B0.5A) with concentrations above 1% show rapid shear recovery that the bioinks recovers its shape after the applied stress is removed.

Revised Figure 5D

D

D) The protein expression levels of cell adhesion molecules including E-selectin, VCAM-1, ICAM-1, PECAM-1, and VE-Cadherin are quantified (**, $p \leq 0.01$; *, $p \leq 0.05$; ns, no significance).

Page 6, Line 173

To verify intercellular communication between tumor cells and brain endothelial cells, adhesion molecules were quantitatively analyzed (Figure 5D). After perfusion of CTCs, the expression of cell adhesion molecules (E-selectin, VCAM-1, and ICAM-1) increased, whereas a key marker for the barrier function of blood vessel walls (VE-cadherin) decreased. Significantly, the group of the larger MCC bending angles corresponded to higher measured molecular concentrations (Table S2), indicating the inflammatory response of brain endothelial cell.

Reviewer's comment 16:

Figure 4G and 4H are missing. On page 4 line 152, Fig. 4G is mentioned but this reviewer couldn't find it in Figure 4, instead, it seems the authors are referring to 4E.

Response 16

We were astonished by our typo. The figure labels were carefully corrected in the revised manuscript. Please refer below for more details.

Page 5, Line 144

The generation of functional cerebrovascular tissue was evaluated using the diffusion

permeability test (**Figure 4E(i)**).

Page 5, Line 147

Compared with the bare group ($2.49 \pm 0.10 \text{ cm s}^{-1}$), the endothelial cell-containing groups showed a significantly reduced diffusion permeability (monoculture, $0.52 \pm 0.06 \text{ cm s}^{-1}$; coculture, $0.31 \pm 0.07 \text{ cm s}^{-1}$), although there was no critical difference between the monoculture and coculture group (**Figure 4E(ii)**).

Page 5, line 153

Furthermore, the coculture group showed relatively higher gene expression levels of endothelial markers including the tight junction makers (ZO-1, occluding), the adherens junction marker (VE-cadherin) and the endothelial marker (CD31), indicating that the synergistic interactions among brain cells enhanced the endothelial functions (**Figure 4G**).

Reviewer's comment 17:

In figure 6, SE, SES etc. are not explained, what are these cells?

Response 17

In response to the reviewer's comment, the abbreviations used in Figure 6 were defined and explained in the revised manuscript. Please refer below for more details.

Page 6, Line 211

For comparative analysis of gene expression on mRNA level, four types of models were prepared: straight MCCs with endothelial cell monoculture (SE), straight MCCs with endothelial cell–stromal cell coculture (SES), 150°-bent MCCs with endothelial cell monoculture (CE), and 150°-bent MCCs with endothelial cell–stromal cell coculture (CES).

Page 21, Line 653

Figure 6. Investigation of underlying molecular mechanisms in brain metastasis through comparative analysis, considering coculture conditions and vascular geometries of the multilayered cerebrovascular conduits (MCCs). A) Brain metastasis occurs through stepwise progression from tumor cell adhesion, extravasation, and invasion in brain tissue, accompanied by complex molecular interactions. B–E) Quantitative reverse-transcription PCR results demonstrate the effects of coculture and vascular curvatures on metastatic brain cancer development in four experimental groups: the straight MCCs with endothelial cell monoculture (SE), the straight MCCs with endothelial cell–stromal cell coculture (SES), the curved MCCs with endothelial cell monoculture (CE), and the curved MCCs with endothelial cell–stromal cell coculture (CES). The expression of key markers associated with B) endothelial barrier function

(ICAM-1, ZO-1, and occludin), C) epithelial–mesenchymal transition (MMP-2, MMP-9, and TWIST), D) neuroinflammatory responses (TNF, IL-6, IL-8, and GFAP), and E) tumorigenesis (HGF, Cx-43) **are analyzed** (***, $p \leq 0.001$; **, $p \leq 0.01$; *, $p \leq 0.05$; ns, no significance).

Response to Reviewer 4

Article ID NCOMMS-23-03632A
Title 3D Bioprinted Multilayered Cerebrovascular Conduits to Study Cancer Extravasation Mechanism Related with Vascular Geometry
Authors Wonbin Park, Jae Sung Lee, Ge Gao, Byoung Soo Kim, Dong-Woo Cho

Summary of response:

We appreciate the constructive comments of the reviewers. We carefully read the valuable comments from the reviewers, and revised the manuscript based on these comments. Please check our specific responses to the reviewer's comments and revisions made in the manuscript as detailed below. The answers to the comments are highlighted in blue, and the modifications in the revised manuscript are indicated with red font.

Reviewer's comment 1:

The research article by W. Park et al. introduces a new class of 3D bioprinted in vitro models of cancer, consisting of various three-layered vascular geometries, to study mechanisms of cancer extravasation. The team used an innovative triaxial bioprinting technique in a support bath, as well as a hybrid brain-specific bioink formulation comprising of brain dECM and alginate to better mimic the native tissue. This platform was used to correlate cerebrovascular geometry/curvature, and flow hemodynamics, to the molecular signatures of cancer cells adhesion and metastatic behavior in the brain. The work is innovative, impactful, and offers great potential for clinical/therapeutic translation of the possible mechanisms that could be identified using this platform. There are however a number of major and minor issues that need to be addressed to further improve the flow and the strength of the submitted work:

Response 1:

We are very grateful for the valuable and thoughtful comments from reviewer #4 and have revised our manuscript accordingly.

Reviewer's comment 2:

MCC maturation is not really supported by the minimal data presented on each cell phenotype. For EC maturation, more in-depth data from gene expression profile or more extensive IHC staining will be required to assess maturity of endothelium. Same need exists for other cells/tissue involved. The NPC differentiation certainly needs more in-depth analysis to claim maturation of

brain tissue. The IHC images in Figure 4D are too vague and not really providing much insight to the cells phenotype. I strongly recommend presenting more extensive results, at different magnifications, to clearly show cellular structure and junctions.

Response 2:

In response to the reviewer's suggestion, the IHC images with the high magnifications showing the cerebrovascular tissue formation were added in the revised manuscript. Please refer below for more details.

Page 5, Line 135

The encapsulated cells stretched their bodies and formed mature endothelium (**Figure 4C, S8**). Confluent CD31 and ZO-1 were detected in large, small, and curved models, demonstrating that the encapsulated HBMECs formed mature vascular tissue independent on the diameter and curvature (**Figure 4D(i)**). Furthermore, the progenitor cells differentiated into astrocytes and neurons over time, expressing GFAP, an astrocytic marker (**Figure 4D(ii)**), and Tuj1 and MAP2, a neural marker (**Figure 4D(iii)**). The three-cell structure formed a mature cerebral vasculature composed of four types of brain cells. In addition, vigorously stretched pericytes were observed using the filamentous actin (F-actin) marker and α -SMA, and *de novo* extracellular matrix (ECM) production of collagen IV and laminin was reproduced (**Figure 4D(iv)**).

Supplementary Figure S8

Figure S8. Tissue maturation in the *in vitro* brain metastasis model over 14 days. A) The triple layers of the cerebrovascular conduits have been maintained on day 14. The scale bar is 500 μ m. B) The encapsulated cells have stretched their bodies, forming the mature endothelium, the confluent pericytes, and the differentiated neural progenitor cells into the astrocytes and the

neurons. The brain endothelial cells, the brain pericytes, the neural progenitor cells, the astrocytes, and the nerve cells are indicated in the figure as BEC, BPC, NPC, AC, and NC, respectively. The scale bars are 50 μm .

Reviewer's comment 3:

The perfusion bioreactor set-up used in this study is not fully elaborated. Schematics and photos of the assembly, how they connect to the inlet and outlet of the printed endothelialized channel, the flow parameters (flow rate, wave form, etc.) should be all explained. How the prescribed rate compares with the clinical/in vivo levels of flow in the brain vasculature should be discussed as well. Does the flow rate match with in vivo? This is particularly critical in order to be able to correlate the hemodynamics factors to the cell response. If the WSS levels are not within the same range, the study on cell response does not seem to be logical.

Response 3

In response to the reviewer's suggestion, the manuscript was revised to include the detailed information of the bioreactor set-up. The clinical relevance of the flow was also described in the discussion section.

Page 5, Line 165

Therefore, MCCs were fabricated at three different angles (0° , straight angle; 90° , right angle; 150° , obtuse angle) and the tumor cells were perfused into the conduits for 24 h at an average velocity of 6 mm s^{-1} under laminar flow condition (Figure 5B).

Page 8, Line 273

To comprehend the multifactorial mechanisms involved, we established a physiologically relevant *in vitro* hemodynamic microenvironment that has the potential to facilitate tumor metastasis but no dominant biomechanical influencer. Instead, the variables under our control could accelerate the complex cascade. In the human circulatory system, small blood vessels have fluid flow velocities ranging from 0.5 to 10 mm s^{-1} and WSS varying from 0 to 30 dynes cm^{-2} ^{41, 42, 43, 44, 45}. Cancer adhesion mainly occurs in the low WSS range below 15 dynes cm^{-2} ⁴⁶. The adhesion efficiency is relatively high at $\text{WSS} < 0.5 \text{ dynes cm}^{-2}$ but decreases as WSS increases^{47, 48, 49, 50}. Taking these factors into account, our model was designed to investigate how various vascular curvatures influence cancer metastasis within human brain blood vessels. A control group was created with straight vessels (0° curvature, a diameter of $700 \mu\text{m}$) having a fluid velocity of 6 mm s^{-1} (corresponding to a flow rate of $2.31 \text{ mm}^3 \text{ s}^{-1}$) and WSS of $\sim 0.6 \text{ dynes cm}^{-2}$. Subsequently, experimental groups were introduced with right-angle (90°) and obtuse-angle (150°) curves. The model replicated a series of metastatic cascades, including tumor circulation, adhesion, and extravasation, demonstrating the interplay of hemodynamic and molecular drivers

in brain metastasis events. By bridging the gap between cellular biology and vascular biomechanics, our study offers new insights into how vascular architectural complexity impacts the spread of cancer cells.

Page 10, Line 374

After thermal crosslinking at 37 °C for 30 min in an incubator, a rocker shaker with a tilting angle of 9° and 5 rpm was used to stabilize the chip until the sacrificial material was removed and the cells stretch their body (**Figure S19(A)**). Proper cell culture media were provided from the separated medium chambers in platform during this time. After the stabilization process, the chip has been connected with a peristaltic pump and a steady flow was perfused for tissue maturation (**Figure S19(B)**).

Page 11, Line 390

Tumor cell perfusion analysis. Tumor cell perfusion analysis was conducted using a peristaltic pump to achieve laminar flow with an average velocity of 6 mm s⁻¹ and average wall shear stress of 60 mPa inside the fabricated conduits. Tumor cells suspended in cell culture medium were perfused for 24 hours, and the chip was harvested at 3-hour and 24-hour time points to observe the tumor dissemination pattern.

Supplementary Figure S19

Figure S19. The bioreactor set-up for development of an in vitro metastatic cancer model. A) A

rocker shaker was used for the stabilization of the 3D bioprinted multilayered cerebrovascular conduits (MCCs) model. B) Peristaltic pump was connected for the fully mature brain tissue formation and the generation of the hemodynamic force for the study of the metastatic cancer progression.

Reviewer's comment 4:

The cell maturation is pretty important in this study, especially for the endothelial layer, as authors mainly focus on the extravasation of cancer cells. Not having a full continuous endothelium, could readily 'leak' the CTCs out of the printed channels and this may not have anything to do with the tumor extravasation in vivo. In Figure 5, they assessed CTC migration distance based on the angle of the channel. The formation of the endothelium is typically not ideal in tortuous channel geometries, leaving some fenestrations/gaps in the endothelium, which could in turn increase the CTC escape. This concern is of more significance for this study, since the IHC images shown in Figure 5 do not show a strong and continuous GFP signal from the ECs. So, authors must provide more careful and zoomed-in analysis of endothelial layer in the three angle groups to exclude the potential effect of channel geometry on endothelium formation.

Response 4

To verify the continuous endothelium in our model, additional assessments were proceeded. The zoomed-in images of the conduits were included in the revised manuscript. Please refer below for more details.

Page 5, Line 168

The introduced metastatic cells adhered to the endothelial monolayer after 3 hours, extravasated from the vasculature, and invaded toward the extracellular matrix, causing slight vasoconstriction (**Figure S11-S14**).

Supplementary Figure S11

Figure S11. Tumor cell adhesion in multilayered cerebrovascular conduits (MCCs) with different

angles. On the confluent endothelium, cancer cells adhere to initiate the metastasis.

Supplementary figure S12

Figure S12. Tumor cell adhesion on the confluent cerebrovascular wall. The tumor cells preferentially adhere to the endothelial wall at the curved portion. The scale bars are 500 μm (left) and 50 μm (right).

Reviewer's comment 5:

There is some ambiguity about the specific bioinks that used for each and different parts of the study plan: the ink for BPE and BEC printing (were cells encapsulated in same hybrid ink?) and then the ink used to print the bath with NPCs. Authors must clarify what inks were used for each. Also, did authros print the bath or just cast/inject the NPC/hydrogel into the molding space? There does not seem to be a need for printing that bath material.

Response 5

Hybrid brain-derived decellularized extracellular matrix (BdECM) bioink was used for all parts of our model. The manuscript revised to describe the detailed fabrication steps and the bioink type utilized in our study. To automate the fabrication process of our model, the bath material was also printed. Considering the reviewer's comment, we revised the manuscript to include the related contents, as written below.

Page 4, Line 122

With a single code activation, the entire fabrication process could be automated. Through the gradual printing steps from housing fabrication to cerebral vessel construction, triple layered MCCs composed of human brain microvascular endothelial cells (HBMECs)-, human brain vascular pericytes (HBVPs)-, and NPCs-laden hybrid BdECM bioinks were successfully created (Figure 4A).

Page 10, Line 362

Construction of *in vitro* brain metastasis model. The brain metastasis model has been automatically fabricated by utilizing our in-house built hybrid 3D bioprinting system (ICBS)⁵⁴ in an isolated 3D printing dedicated room. The internal temperature of the room is maintained at 18°C to inhibit the crosslinking of the printed hydrogel, while the humidity is kept at 40-50%. After the first medium chamber was printed with poly (ethylene/vinyl acetate) (PEVA; Polysciences, USA), an NPCs-laden hybrid BdECM bioink was printed at the center part to utilize it as a bath. Afterward, a customized triaxial nozzle (Ramé-Hart Instrument, USA) was submerged in the bath after loading materials into each part: core part, 35% (w/v) Pluronic F127 in 100 mM calcium chloride solution; intermediate part, HBMEC-laden (final cell density: 1×10^7 cells mL⁻¹) hybrid BdECM bioink; shell part, HBVP-laden (final cell density: 1×10^6 cells mL⁻¹) hybrid BdECM bioink. The tubular structures were created in the bath following G-code commands with printhead velocity mentioned previously.

Reviewer's comment 6:

Figure captions and presentation and elaboration of results in the text have some major issues and are hard to follow. See the several minor comments listed below for details. The flow biomechanics simulation and analyses presented in the Figure 5 and in the text are rather superficial and incomplete. The relevance of the predicted WSS to the biological levels in the brain vascular tissue should be discussed.

Response 6

In response to the reviewer's comment, the manuscript was revised to include the physiological relevance of the predicted wall shear stress. Please refer below for more details.

Page 6, Line 189

Accordingly, hemodynamic factors, including fluid velocity and WSS, were analyzed through computational fluid dynamics (CFD) simulations. Different fluid velocity distributions were predicted based on the vascular bending angle (**Figure 5E(i)**). A velocity gradient was formed toward the center of the bend by the centrifugal force owing to the fluid flow in the curved vasculatures. Quantitatively, the range of flow velocity was from 0.1 to 11.02 mm s⁻¹ (**Figure 5E(ii)**). The bias of the radial flow gradient intensified along with the curvature degree; the low velocity (0.5–2.5 mm s⁻¹) area increased, whereas the high-velocity area decreased (**Figure 5E(iii)**). Similarly, an elongated WSS gradient was generated by the curvature of the vessel (**Figure 5F(i)**). The WSS value towards the center of the curvature was higher than the opposite side. In particular, the range in which the WSS was less than 60 mPa expanded in the curved groups (**Figure 5F(ii)**). In addition, a relatively slow pressure change was formed in the opposite

direction of the center of the curvature. **(Figure S17)**. This implies that the tumor cells were exposed to the specific pressure level for the extended period. For MCCs with different bending degrees but the same inner diameter and same blood flow velocity, greater curvature may result in the regions of the low shear stress and velocity and the prolonged such pressure, promoting metastasis. These findings imply the high possibility that various biomechanical and cellular factors can influence cancer metastasis through the prolongation of the residence time of tumor cells.

Page 8, Line 276

In the human circulatory system, small blood vessels have fluid flow velocities ranging from 0.5 to 10 mm s⁻¹ and WSS varying from 0 to 30 dynes cm^{-222, 41, 42, 43, 44, 45}. Cancer adhesion mainly occurs in the low WSS range below 15 dynes cm⁻²⁴⁶. The adhesion efficiency is relatively high at WSS < 0.5 dynes cm⁻² but decreases as WSS increases^{47, 48, 49, 50}. Taking these factors into account, our model was designed to investigate how various vascular curvatures influence cancer metastasis within human brain blood vessels. A control group was created with straight vessels (0° curvature, a diameter of 700 μm) having a fluid velocity of 6 mm s⁻¹ (corresponding to a flow rate of 2.31 mm³ s⁻¹) and WSS of ~ 0.6 dynes cm⁻². Subsequently, experimental groups were introduced with right-angle (90°) and obtuse-angle (150°) curves. The model replicated a series of metastatic cascades, including tumor circulation, adhesion, and extravasation, demonstrating the interplay of hemodynamic and molecular drivers in brain metastasis events.

Reviewer's comment 7:

In figure captions, make sure to define all the acronyms used in each figure. Many abbreviations that are used in the Figure 1 are not defined and explained in the text. Examples: MCC, BPC, CTC, BEC, CPF127, etc. Some of these are extremely important and critical for the reader to understand and follow the content of the article.

Response 7

The abbreviations used in the Figure 1 were defined and explained in the revised manuscript. Please refer below for more details.

Page 2, Line 34

In this study, we propose a direct bioprinting strategy to construct multilayered cerebrovascular conduits (MCCs) with geometrically varying curvatures for brain metastasis studies **(Figure 1)**. The MCCs were developed via an in-bath 3D triaxial bioprinting technique employing an ionically cross-linkable brain-specific bioink laden **with multiple cell sources, including brain endothelial cells (BECs), brain pericytes (BPCs), and neural progenitor cells (NPCs), and a sacrificial material (calcium-added PF-127; CPF-127)** to investigate the effects of vessel curves

and cellular interactions in circulating tumor cells (CTCs) dissemination.

Page 16, Line 580

Figure 1. Schematic illustration for development of in vitro brain metastasis model. A) Brain metastasis usually occurs through blood vessels connecting distant organs to brain. In the brain, circulating tumor cells, disseminating through the complex vascular network, are influenced by cerebrovascular geometry and other cellular components during metastatic progression. B) To recapitulate the pathophysiological mechanisms in brain metastasis of circulating tumor cells (CTCs) *in vitro*, multilayered cerebrovascular conduits are constructed using in-bath 3D triaxial bioprinting technology. Multiple cell types including neural progenitor cells (NPCs), brain endothelial cells (BECs), and brain pericytes (BPCs) are utilized to fabricate triple layered vascular structures with diverse curvatures. C) Through rapid crosslinking between hybrid brain-derived decellularized extracellular matrix bioink and calcium ion from the core sacrificial material (calcium-added Pluronic F-127; CPF-127), the hollow cerebrovascular conduits are constructed with high structural stability. Into a fully mature brain blood vessel, circulating tumor cells are introduced and metastatic pattern changes depending on hemodynamic variants and biological interactions are investigated. The model allows multifaceted investigation of the underlying mechanisms of brain metastasis, demonstrating the interplay of cerebrovascular curvatures and cellular interaction on tumor cell adhesion and extravasation.

Reviewer's comment 8:

Figure 1 caption is too short and does not really elaborate the highly complex schematics that are presented. Better annotation tools should be used to highlight each component of these schematics and should be described in the caption. Some details in the figure are impossible to read (too small).

Response 8

In response to this feedback, the caption of Figure 1 was revised. In addition, Figure 1 was edited using the annotation tools and the abbreviations were defined in the caption. The too small fonts were eliminated for the better readership. Please refer before for more details.

Revised Figure 1

Figure 1. Schematic illustration for development of in vitro brain metastasis model. A) Brain metastasis usually occurs through blood vessels connecting distant organs to brain. In the brain, circulating tumor cells, disseminating through the complex vascular network, are influenced by **cerebrovascular geometry and other cellular components during metastatic progression**. B) To recapitulate the pathophysiological mechanisms **in brain metastasis of circulating tumor cells (CTCs) in vitro**, multilayered cerebrovascular conduits are constructed using in-bath 3D triaxial bioprinting technology. **Multiple cell types including neural progenitor cells (NPCs), brain endothelial cells (BECs), and brain pericytes (BPCs) are utilized to fabricate triple layered vascular structures with diverse curvatures**. C) Through rapid crosslinking between hybrid brain-derived decellularized extracellular matrix bioink and calcium ion from the core sacrificial material (calcium-added Pluronic F-127; CPF-127), the hollow cerebrovascular conduits are constructed with high structural stability. Into a fully mature brain blood vessel, circulating tumor cells are introduced and **metastatic pattern changes depending on hemodynamic variants and biological interactions are investigated**. The model allows multifaceted investigation of the underlying mechanisms of brain metastasis, demonstrating the **interplay** of cerebrovascular curvatures **and cellular interaction** on tumor cell adhesion and extravasation.

Reviewer's comment 9:

In describing the bioink compositions tested, "B" and "A" should be defined in "1.0B1.0A" and other labels. Sounds like authors meant to label BdECM and Alginate by B and A, but this must be defined and clarified.

Response 9

In response to this feedback, the labels were defined in the revised manuscript. Please refer below for more details.

Page 3, Line 60

To combine the two source technologies for fabricating a cerebrovascular tissue equivalent, the optimal condition of the hybrid BdECM bioink was established by comparing five different formulations with BdECM (B) and alginate (A) (0.5B, 0.5 wt. % BdECM; 1.0B, 1.0 wt.% BdECM; 1.5B, 1.5 wt.% BdECM; 1.0B0.5A, 1.0 wt.% BdECM mixed with 0.5 wt.% alginate; 1.0B1.0A, 1.0 wt.% BdECM mixed with 1.0 wt.% alginate).

Page 17, Line 596

Figure 2. Rheological, physiochemical, and biological assessment of hybrid brain-derived extracellular matrix (BdECM) bioink. A) Considering printability and shape fidelity, a hybrid bioink is formulated using BdECM and alginate.

Reviewer's comment 10:

Figure 2: caption is vague and incomplete. Scale bars are missing in several panels. For panels (G and H), it is unclear what cell types are used there, and also there does not seem to be any red signal in neither of groups in panel (H), so the conclusion that one bioink group is 'optimal' for cell compatibility is not well supported.

Response 10

The caption was edited, and the scale bars are added. The information related to the cell type was added and the intensity of red color was increased. Additionally, quantitative data of the cell viability was added. Please refer below for more detail.

Revised Figure 2F

Page 17, Line 607

G, H) Because metabolic activity and cell viability of neural progenitor cells decrease with increasing alginate concentration, 1.0B0.5A is selected as the optimal bioink considering the rheological, chemical, and biological properties (***, $p \leq 0.001$). Scale bars, 200 μm .

Revised Figure 2H

Page 4, Line 93

In addition, the viability of NPCs and brain endothelial cells decreased with increasing alginate concentrations (1.0B, 89.83 ± 1.48%; 1.0B0.5A, 83.73 ± 0.97%; 1.0B1.0A, 83.33 ± 3.43%) (Figure 2H, S4).

Reviewer's comment 11:

In most figure captions the scale bars are not defined. Figure 3(A) is missing the scale bar.

Response 11

The figure captions were revised to include definition of the scale bars. In addition, the scale bar

for Figure 3A has as added. Please refer below for more details.

Page 17, Line 596

Figure 2. Rheological, physiochemical, and biological assessment of hybrid brain-derived extracellular matrix (BdECM) bioink. A) Considering printability and shape fidelity, a hybrid bioink is formulated using BdECM and alginate. BdECM and alginate are labeled as B and A, respectively. B) Most of BdECM-based bioinks show shear-thinning behavior and C) stable sol-gel transition with clear flow points, where the storage modulus (G') and loss modulus (G'') reverse at the threshold shear stress, while 0.5B bioink exhibits fluidic-dominant property. D) BdECM-based bioinks including (i) pure BdECM bioink (1.0B) and hybrid BdECM bioink (1.0B0.5A) with concentrations above 1% show rapid shear recovery that the bioinks recovers its shape after the applied stress is removed. E) The fluidic-dominant hydrogel cannot sustain the hollow tubular structure, but the solid-dominant hydrogels exhibit suitable structural stability for applicability as bath materials and bioinks. Scale bars, 2 mm. F) Rapid ionic crosslinking due to the presence of alginate prevents leakage at the curved portion of tubes. Scale bars, 500 μm . G, H) Because metabolic activity and cell viability of neural progenitor cells decrease with increasing alginate concentration, 1.0B0.5A is selected as the optimal bioink considering the rheological, chemical, and biological properties (***, $p \leq 0.001$). Scale bars, 200 μm .

Page 18, Line 611

Figure 3. Parameter study for geometrical control of a multilayered cerebrovascular conduit using in-bath three dimensional triaxial bioprinting technology. A) For direct fabrication of multilayered conduits, a triaxial nozzle, the hybrid brain-derived extracellular matrix (BdECM) bioink, and calcium-added pluronic F-127 (CPF-127) were utilized. Dimensionally controllable conduits are constructed by managing pneumatic pressure, printhead velocity, and programming commands. Scale bar, 500 μm . B) Using triaxial nozzle with suitable pneumatic pressures, multiple layers of the hollow tube are easily organized. Scale bars, 500 μm . C, D) As the printhead velocity increases, the inner diameter and wall thickness of the construct significantly decrease. Scale bars, 500 μm . E) By appropriate G-code commands, tubes with different angles can be elaborated. Scale bars, 500 μm . F) Based on the parametric study results, fabrication processes are designed for conduits with various geometries, including straight, stenosis, double-stenosis, and curved structures. Scale bars, 2 mm.

Page 19, Line 623

Figure 4. In-bath 3D triaxial bioprinting of fully mature multilayered cerebrovascular conduits (MCCs). A) The strategy to construct an *in vitro* brain metastasis model containing MCC was designed. Through the gradual printing steps, B) perfusable brain blood vessels can be fabricated with diverse vessel curvatures. C) The printed constructs are cultivated for 14 days to generate

mature endothelium and other supportive brain-resident cells. **Scale bars, 200 μm .** D) Immunostaining results show that mature MCCs express (i) endothelial junctional marker (CD31), (ii) glial cell marker (GFAP), and (iii) neuronal marker (Tuj1). In addition, (iv) *de novo* production of extracellular matrix components including collagen IV (Col4) and laminin (Lam) is indicated. **Scale bars, 200 μm .** E) Endothelial barrier function is verified via diffusion permeability assay (***, $p \leq 0.001$; ns, no significance). **Scale bars, 200 μm .** F, G) **Coculture condition of endothelial cells with pericytes and glial/neural cells enhances trans-epithelial electrical resistance (TEER) and elevated the expression of the the markers including the tight junction makers (ZO-1, occluding), the adherens junction marker (VE-cadherin) and the endothelial marker (CD31) (***, $p \leq 0.001$; **, $p \leq 0.01$; *, $p \leq 0.05$).**

Page 20, Line 637

Figure 5. Investigation of effects of vascular geometry on tumor extravasation. A) The introduced tumor cells preferentially adhered to the endothelium at the curved, rather than the straight, portion. **Scale bar, 500 μm .** B) Fluorescence-labeled tumor cells were perfused into multilayered cerebrovascular conduits with three different bending angles (0° , 90° , and 150°), and tumor dissemination was visualized (**, $p \leq 0.01$). **Scale bars, 1 mm and 200 μm .** C) With the increase in the vascular bending angle, larger number of invading tumor cells are detected (***, $p \leq 0.001$; **, $p \leq 0.01$; *, $p \leq 0.05$; ns, no significance). D) The **protein expression levels of cell adhesion molecules including E-selectin, VCAM-1, ICAM-1, PECAM-1, and VE-Cadherin are quantified (**, $p \leq 0.01$; *, $p \leq 0.05$; ns, no significance).** E) Fluid-flow patterns for different vessel curves are investigated by computational fluid dynamics simulation, and (i) velocity contours are obtained. (ii) The **average flow velocity was 6 mm s^{-1} and the biased velocity distribution increased according to the increase of vascular curvatures.** (iii) The area of relatively low velocity ($0.5\text{--}2.5 \text{ mm s}^{-1}$) is enlarged with the increase of vessel curve. F) Similarly, (i) the distribution of wall shear stress **biased and its gradient is generated following the increase of the vascular curvatures.** (ii) **In addition, the area with the low wall shear stress in the elbow portion is increased (***, $p \leq 0.001$; **, $p \leq 0.01$; *, $p \leq 0.05$; ns, no significance).**

Revised Figure 3A

A) For direct fabrication of multilayered conduits, a triaxial nozzle, the hybrid brain-derived extracellular matrix (BdECM) bioink, and calcium-added pluronic F-127 (CPF-127) were utilized. Dimensionally controllable conduits are constructed by managing pneumatic pressure, printhead velocity, and programming commands. Scale bar, 500 μm .

Reviewer's comment 12:

Figure 5C-ii does not show error bars for the quantified data, and no statistical analysis. These should be added.

Response 12

It was added accordingly.

Revised Figure 5C(ii)

C) With the increase in the vascular bending angle, larger number of invading tumor cells are detected (***, $p \leq 0.001$; **, $p \leq 0.01$; *, $p \leq 0.05$; ns, no significance).

REVIEWER COMMENTS

Reviewer #1 (Remarks to the Author):

This revision of the original manuscript by Park and co-workers very thoroughly done and addressed most of the issues raised by the reviewers.

Reviewer #2 (Remarks to the Author):

In this revised manuscript Park and colleagues utilize ex vivo bioengineered models to analyze adhesion and communication mechanisms between metastatic cancer cells and cells that comprise the neurovascular unit. A focus is placed on pathways that promote tumor cell extravasation across the neurovasculature, including the select effects of blood vessel curvature and how this impacts endothelial barrier integrity and metastatic seeding. The scientific premise is solid. The topic of study is fundamentally important and clinically relevant. The field needs 3D models of blood vessels in the brain parenchyma, although it remains uncertain how broadly applicable these systems will be to non-bioengineers. Nonetheless, the data are presented in a rational manner and the experimental results are of high quality and involve adequate rigor. The subject will be of interest to researchers in the neurovascular biology community.

The authors were quite responsive to the multiple points raised by the four Reviewers. New experimental data are included. . For example, the decellularized ECM composition was analyzed by mass spectrometry, revealing thousands of proteins. In addition, function-blocking antibody experiments were included to address pathways that enable tumor cell extravasation across the endothelium in co-culture. Significant modifications have been made to the text as well.

There are a couple of issues that should be addressed by the authors. While additional experiments are not needed, these points should be dealt with my editing the results and conclusions.

1. The authors continue to utilize neurospheres/neural stem and progenitor cells in the 3D models, arguing that these cells are more compatible in co-culture systems and give rise to astrocytes and neuronal cells over time. One way the authors measure astrocyte "differentiation" is by quantitation of GFAP mRNA levels in the co-culture model. It should be noted that many bona fide astrocytes, especially in the cerebral cortical neurovasculature, lack GFAP expression. GFAP up regulation is a marker for activated astrocytes, indicating that the higher GFAP levels in the 3D system may be the result of activation rather than a differentiation event. Indeed, many cultured astrocytes are GFAP+ due to activation status in vitro. This possibility should be highlighted by the authors.

2. The authors continue to refer to the elongated vascular endothelial cells in their 3D models as more "mature and "differentiated." This is based more on a "stretched" morphology rather than the expression of differentiation markers. The authors should edit their results and figure legends to refer to the morphological changes without referencing cellular differentiation or maturation status.

3. At a couple of points in the manuscript Tuj1/Map2+ cells are referred to as "neural," The more precise terminology is neuronal.

Reviewer #3 (Remarks to the Author):

Summary: The authors utilized a direct 3D bioprinting strategy to construct multilayered cerebrovascular conduits (MCCs) with varying geometries and curvatures to study brain metastasis. First, a hybrid BdECM bioink formulation was optimized for printing fidelity and cell viability. An in vitro

brain metastasis model was constructed using triple layered MCCs consisting of HBMECs (endothelial), HBVPs (pericytes) and NPCs (neural progenitor cells) with perfusable and geometrically tunable blood vessels. Using their platform, the authors observed that vascular curvature results in the upregulation of endothelial adhesion molecules and downregulation of endothelial tight junction markers, as well as the upregulation of EMT and tumorigenesis markers in cancer cells. Ultimately, this platform has the potential to be used as a preclinical model for studying the molecular mechanisms involving brain metastases.

The significant consideration and efforts undertaken by the authors to address the comments and clarifications made in the first revision are greatly appreciated. The reviewer also appreciates the measurement of mechanical and rheological properties of the hybrid BdECM compared to native brain tissue, quantification of the expression of endothelial adhesion molecules as a function of bending angles, as well as the appropriate blocking/functional assays, which yielded interesting findings. A few further comments should be addressed.

Major Comments

Comment 1: This reviewer is confused with the authors response on the cause of higher shear stress on the convex side vs concave side, "This was due to the Newtonian fluid (cell culture medium) causing reduced contact between the concave side wall and the fluid compared to the convex side". Isn't shear stress level is determined by the velocity gradient instead of contact area? and how does Newtonian fluid cause reduced contact? Isn't Newtonian fluid is defined by the constant viscosity?

Comment 2: It is very confusing how the pressure simulation is conducted, as this quite contradict to what one would expect to see re the pressure distribution in curved pipe (i.e., low pressure on the intrados and high pressure on the extrados). If the simulation is truly reflecting what's really happening, the authors need to explain why there's almost no pressure gradient along the radius direction at the bent area.

Comment 3: Regarding the comparisons to native brain tissue, the authors further specified the protein composition of the BdECM which is greatly appreciated. This reviewer wonders how the protein composition in the BdECM compares to that of native brain and whether it may also contribute to the observed differences in mechanical/rheological properties. In particular, this reviewer wonders how Figures S2 B (Top 10 proteins), C (Top 5 basement membranes), D (Top 5 interstitial matrix), and E (Top 10 collagens) compares to that of native brain tissue, particular when looking at Figure S1 with a large difference in collagen content in the BdECM?

Comment 4: For the functional blocking assay Figure S15, quantification of the number of cells adhered as well as the cell aspect ratio (to describe cell morphology) to the endothelium with/without blocking antibodies would present a more robust argument for the outcome of these experiments rather than observation or description alone. Moreover, the number of samples per group for these experiments should be specified.

Comment 5: The authors observed a decrease in cell viability with the addition of alginate – however the addition of alginate into collagen-based gels can also affect cell morphology and spreading (Ort et al., ACS Biomater. Sci. Eng., 2021). Does this also occur with NPCs and brain endothelial cells with the addition of alginate?

Minor Comments

Comment 6: Regarding the measurement of native brain rheological and mechanical properties – the authors state that the differences may be due to cell density but should perhaps further specify the difference in cell density in terms of cells per unit volume in native tissue for clarification purposes.

Comment 7: The authors mention use of CTCs throughout the manuscript and in the supplementary

data but should specify the nature or source of the CTCs, and whether the CTC source or cell type changes between experiments.

Comment 8: Figure S17 the colorimetric scale bar for pressure could be enlarged for better visibility.

Comment 9: Figure 2H(ii) on cell viability could be enlarged.

Reviewer #4 (Remarks to the Author):

- The maturation of MCC cells still need more robust and quantitative analysis. The images provided are still not sufficient to support the big claim on "fully mature multilayered cerebrovascular conduits (MCCs)". Authors must provide PCR / micro-array quantitatively to demonstrate significant upregulation of specific maturation markers for each cell type and compare either with control cells in their study, or some reported numbers from the literature.

Response to Reviewer 1

Article ID NCOMMS-23-03632B
Title 3D Bioprinted Multilayered Cerebrovascular Conduits to Study Cancer Extravasation Mechanism Related with Vascular Geometry
Authors Wonbin Park, Jae-Seong Lee, Ge Gao, Byoung Soo Kim, Dong-Woo Cho

Summary of response:

We appreciate the constructive comments of the reviewers. We carefully read the valuable comments from the reviewers, and revised the manuscript based on these comments. Please check our specific responses to the reviewer's comments and revisions made in the manuscript as detailed below. The answers to the comments are highlighted in blue, and the modifications in the revised manuscript are indicated with red font.

Reviewer's comment 1:

This revision of the original manuscript by Park and co-workers very thoroughly done and addressed most of the issues raised by the reviewers.

Response 1:

We appreciate the reviewer's supportive comment.

Response to Reviewer 2

Article ID NCOMMS-23-03632B
Title 3D Bioprinted Multilayered Cerebrovascular Conduits to Study Cancer Extravasation Mechanism Related with Vascular Geometry
Authors Wonbin Park, Jae-Seong Lee, Ge Gao, Byoung Soo Kim, Dong-Woo Cho

Summary of response:

We appreciate the constructive comments of the reviewers. We carefully read the valuable comments from the reviewers, and revised the manuscript based on these comments. Please check our specific responses to the reviewer's comments and revisions made in the manuscript as detailed below. The answers to the comments are highlighted in blue, and the modifications in the revised manuscript are indicated with red font.

Reviewer's comment 1:

In this revised manuscript Park and colleagues utilize ex vivo bioengineered models to analyze adhesion and communication mechanisms between metastatic cancer cells and cells that comprise the neurovascular unit. A focus is placed on pathways that promote tumor cell extravasation across the neurovasculature, including the select effects of blood vessel curvature and how this impacts endothelial barrier integrity and metastatic seeding. The scientific premise is solid. The topic of study is fundamentally important and clinically relevant. The field needs 3D models of blood vessels in the brain parenchyma, although it remains uncertain how broadly applicable these systems will be to non-bioengineers. Nonetheless, the data are presented in a rational manner and the experimental results are of high quality and involve adequate rigor. The subject will be of interest to researchers in the neurovascular biology community.

The authors were quite responsive to the multiple points raised by the four Reviewers. New experimental data are included. For example, the decellularized ECM composition was analyzed by mass spectrometry, revealing thousands of proteins. In addition, function-blocking antibody experiments were included to address pathways that enable tumor cell extravasation across the endothelium in co-culture. Significant modifications have been made to the text as well.

There are a couple of issues that should be addressed by the authors. While additional experiments are not needed, these points should be dealt with my editing the results and conclusions.

Response 1:

We are very grateful for the valuable comments from the reviewer. The manuscript has been

revised in accordance with the reviewer's comments as shown below.

Reviewer's comment 2:

The authors continue to utilize neurospheres/neural stem and progenitor cells in the 3D models, arguing that these cells are more compatible in co-culture systems and give rise to astrocytes and neuronal cells over time. One way the authors measure astrocyte "differentiation" is by quantitation of GFAP mRNA levels in the co-culture model. It should be noted that many bona fide astrocytes, especially in the cerebral cortical neurovasculature, lack GFAP expression. GFAP up regulation is a marker for activated astrocytes, indicating that the higher GFAP levels in the 3D system may be the result of activation rather than a differentiation event. Indeed, many cultured astrocytes are GFAP+ due to activation status in vitro. This possibility should be highlighted by the authors.

Response 2:

In agreement with the reviewer's comment, the GFAP marker is used to show the reactive responses of astrocytes. Yet, the marker has been still used in many studies to reveal the astrocytic differentiation of the neural progenitor cells; please refer to the following articles.

Reference 1: Julia, T. C. W., et al. "An efficient platform for astrocyte differentiation from human induced pluripotent stem cells." Stem cell reports 9.2 (2017): 600-614,

Reference 2: Magistri, Marco, et al. "A comparative transcriptomic analysis of astrocytes differentiation from human neural progenitor cells." European Journal of Neuroscience 44.10 (2016): 2858-2870,

Reference 3: Lippmann, Ethan S., et al. "Blood-brain barrier modeling with co-cultured neural progenitor cell-derived astrocytes and neurons." Journal of neurochemistry 119.3 (2011): 507-520)

Therefore, to the best of our knowledge, the GFAP expression can be utilized as a marker representing astrocyte differentiation. On the other hand, we also used the marker to demonstrate the activation of the astrocytic cells under the condition of the presence of the tumor cells (**Figure 6D**).

Considering the reviewer's comment, we noticed that the GFAP expression in our *in vitro* multicellular model under normal condition could suggest the activation of astrocytic cells, as well as the differentiation. Taken all together, we included the related contents in the revised manuscript to supplement the Discussion part and address the reviewer's concern. We hope for the reviewer's understanding on the changes we have made.

Page 8, Line 280

To evaluate the tissue formation of the 3D bioprinted cerebrovascular system, we used specific molecular markers indicating the endothelial tight junction development, pericyte function, and

neuronal/glial differentiation. In particular, the expression of the GFAP marker suggests that the neural progenitor cells underwent astrocyte differentiation^{46, 47, 48}, but it also possibly indicates astrocytic activation in the *in vitro* culture environment, warranting further investigation⁴⁹.

Added references

[46] Lippmann ES, Weidenfeller C, Svendsen CN, Shusta EV. Blood–brain barrier modeling with co-cultured neural progenitor cell-derived astrocytes and neurons. *Journal of neurochemistry* 119, 507-520 (2011).

[47] Julia T, et al. An efficient platform for astrocyte differentiation from human induced pluripotent stem cells. *Stem cell reports* 9, 600-614 (2017).

[48] Magistri M, et al. A comparative transcriptomic analysis of astrocytes differentiation from human neural progenitor cells. *European Journal of Neuroscience* 44, 2858-2870 (2016).

[49] Escartin C, et al. Reactive astrocyte nomenclature, definitions, and future directions. *Nature neuroscience* 24, 312-325 (2021)

Reviewer's comment 3:

The authors continue to refer to the elongated vascular endothelial cells in their 3D models as more "mature and "differentiated." This is based more on a "stretched" morphology rather than the expression of differentiation markers. The authors should edit their results and figure legends to refer to the morphological changes without referencing cellular differentiation or maturation status.

Response 3

The changes were made according to the reviewer's comment, as follows.

Page 5, Line 146

Original:

The encapsulated cells stretched their bodies and formed mature endothelium (**Figure 4C, S8**). Confluent CD31 and ZO-1 were detected in large, small, and curved models, demonstrating that the encapsulated HBMECs formed mature vascular tissue independent on the diameter and curvature (Figure 4D(i)).

Revised:

The encapsulated cells stretched their bodies and formed **endothelial barriers** (**Figure 4C, S10**). Confluent CD31 and ZO-1 were detected in large, small, and curved models, indicating that the encapsulated HBMECs formed **the selective cellular barriers** regardless of their diameter and curvature (**Figure 4D(i)**).

Page 25, Line 788

Original:

C) The printed constructs are cultivated for 14 days to generate mature endothelium and other supportive brain-resident cells. Scale bars, 200 μm .

Revised:

C) The printed constructs are cultivated for 14 days to generate **confluent cerebrovascular endothelium with the additional mural, glial, and neuronal cells**. Scale bars, 200 μm .

Reviewer's comment 4:

At a couple of points in the manuscript Tuj1/Map2+ cells are referred to as "neural," The more precise terminology is neuronal.

Response 4

We agree with the reviewer's comment and made the necessary changes throughout the revised manuscript.

Page 5, Line 149

Original:

Furthermore, the progenitor cells differentiated into astrocytes and neurons over time, expressing GFAP, an astrocytic marker (**Figure 4D(ii)**), and Tuj1 and MAP2, a neural marker (**Figure 4D(iii)**).

Revised:

Furthermore, the progenitor cells differentiated into astrocytes and neurons over time, expressing GFAP, which is an astrocytic marker (**Figure 4D(ii)**), and Tuj1 and MAP2, which are **neuronal markers (Figure 4D(iii))**.

Response to Reviewer 3

Article ID NCOMMS-23-03632B
Title 3D Bioprinted Multilayered Cerebrovascular Conduits to Study Cancer Extravasation Mechanism Related with Vascular Geometry
Authors Wonbin Park, Jae-Seong Lee, Ge Gao, Byoung Soo Kim, Dong-Woo Cho

Summary of response:

We appreciate the constructive comments of the reviewers. We carefully read the valuable comments from the reviewers, and revised the manuscript based on these comments. Please check our specific responses to the reviewer's comments and revisions made in the manuscript as detailed below. The answers to the comments are highlighted in blue, and the modifications in the revised manuscript are indicated with red font.

Reviewer's comment 1:

The authors utilized a direct 3D bioprinting strategy to construct multilayered cerebrovascular conduits (MCCs) with varying geometries and curvatures to study brain metastasis. First, a hybrid BdECM bioink formulation was optimized for printing fidelity and cell viability. An in vitro brain metastasis model was constructed using triple layered MCCs consisting of HBMECs (endothelial), HBVPs (pericytes) and NPCs (neural progenitor cells) with perfusable and geometrically tunable blood vessels. Using their platform, the authors observed that vascular curvature results in the upregulation of endothelial adhesion molecules and downregulation of endothelial tight junction markers, as well as the upregulation of EMT and tumorigenesis markers in cancer cells. Ultimately, this platform has the potential to be used as a preclinical model for studying the molecular mechanisms involving brain metastases.

The significant consideration and efforts undertaken by the authors to address the comments and clarifications made in the first revision are greatly appreciated. The reviewer also appreciates the measurement of mechanical and rheological properties of the hybrid BdECM compared to native brain tissue, quantification of the expression of endothelial adhesion molecules as a function of bending angles, as well as the appropriate blocking/functional assays, which yielded interesting findings. A few further comments should be addressed.

Response 1:

We thank the reviewer for the constructive comments and have revised our manuscript accordingly. Our point-by-point responses are presented below.

Reviewer's comment 2:

This reviewer is confused with the authors response on the cause of higher shear stress on the convex side vs concave side, “This was due to the Newtonian fluid (cell culture medium) causing reduced contact between the concave side wall and the fluid compared to the convex side”. Isn’t shear stress level is determined by the velocity gradient instead of contact area? and how does Newtonian fluid cause reduced contact? Isn’t Newtonian fluid is defined by the constant viscosity?

Response 2

We apologize for the confusion caused by the unclear response during the previous revision process. In addition, in fact, we misunderstood the previous comment, which might make the reviewer confused.

For clarity, during this revision process, we made the point-by-point responses to the reviewer’s concerns and comments as written below.

The sentence in the previous response, “This was due to the Newtonian fluid (cell culture medium) causing reduced contact between the concave side wall and the fluid compared to the convex side,” must be the revised sentence in the current response, “The low shear stress at concave side may be cause by the centripetal force.” The phrase “reduced contact” was used to explain the centripetal force. We apologize for the confusion caused by the unclear expression, once again.

We totally agree with the reviewer’s comment that the shear stress level is determined by the velocity gradient and Newtonian fluid is defined by the constant viscosity.

On the other hand, we did not simulate the pressure gradient in the original version of the study. However, considering the reviewer’s comment that the pressure is one of the important hemodynamic factor, we simulated the fluid pressure in the curved tube. The pressure gradient along with the longitudinal direction was indicated, but a radial gradient was not observed. It may be cause by the low inertial force resulted from the low fluid velocity under our experimental condition (Reference: *Di Carlo D. Inertial microfluidics. Lab on a Chip 9, 3038-3046 (2009)*). Additionally, when the input fluid velocity increased, the radial pressure gradient was observed, as indicated by the following additional simulation results. In particular, a higher pressure at the extrados was observed.

The related content and references were supplemented in the revised manuscript, as shown below.

Page 6, Line 202

A velocity gradient of the perfused medium (Reynolds number, ~ 4.61) was generated due to the centripetal force of the fluid flow in the curved conduits³³. Quantitatively, the flow velocity ranged from 0.1 to 11.02 mm s⁻¹ (**Figure 5E(ii)**).

Page 6, Line 207

The WSS value at the extrados was lower than that at the intrados, implying that the WSS was proportional to the velocity gradient³⁴. In particular, the range in which the WSS <60 mPa was expanded for the curved groups (**Figure 5F(ii)**).

Supplementary Figure S21

Figure S21. Generation of the pressure gradient at different fluid flow velocities. The radial pressure gradient is induced when the input fluid velocity exceeds ~20 mm s⁻¹ in bent tubes.

Page 6, Line 210

On the other hand, the fluid pressure gradient along with the longitudinal direction was observed, and the pressure change was slower on the extrados compared to the intrados (**Figure S20**). Additionally, no radial pressure gradient was indicated because of the low fluid velocity and low inertial force under the experimental condition (**Figure S21**)³⁵.

Added references

- [33] Wang H, Krüger T, Varnik F. Geometry and flow properties affect the phase shift between pressure and shear stress waves in blood vessels. *Fluids* 6, 378 (2021).
- [34] Hutmacher DW, Singh H. Computational fluid dynamics for improved bioreactor design and 3D culture. *Trends in biotechnology* 26, 166-172 (2008).
- [35] Di Carlo D. Inertial microfluidics. *Lab on a Chip* 9, 3038-3046 (2009).

Reviewer's comment 3:

It is very confusing how the pressure simulation is conducted, as this quite contradict to what one would expect to see re the pressure distribution in curved pipe (i.e., low pressure on the intrados and high pressure on the extrados). If the simulation is truly reflecting what's really happening,

the authors need to explain why there's almost no pressure gradient along the radius direction at the bent area.

Response 3

We perceived that the methods on computational simulation was not included in the previous manuscript and appreciate the reviewer for pointing out this potential ambiguity. The related content was added in the revised manuscript as follows.

Page 12, Line 439

Computational fluid dynamics simulation analysis. The 3D geometries of the models were generated using a commercial software, Solidworks (Dassault Systems, FR). The conduits were modeled with the diameter of 700 μm and the length of 16 mm. Subsequently, fluid flow simulations were performed using a commercial software, ANSYS (2023 R1, ANSYS Inc., USA), following a user guide and established methods. The fluid flow in the channel was assumed to be Newtonian fluid with a density of 1000 kg m^{-3} and a viscosity of 0.00093 Pa·s. The vascular walls were assumed to be rigid and a no-slip boundary condition was applied to them. The initial velocity was $\sim 6 \text{ mm s}^{-1}$ at the inlet and the initial pressure was 0 Pa at the outlet. The length of time step was 0.001 s. The fluid flow velocity, WSS, and pressure gradient were calculated employing the shear stress transport $k\text{-}\omega$ model, according to Navier-Stokes equations,

$$\nabla \cdot \mathbf{u} = 0,$$

$$\rho \frac{\partial \mathbf{u}}{\partial t} + \rho \mathbf{u} + \nabla \mathbf{u} + \nabla p = \mu \nabla^2 \mathbf{u} ,$$

where \mathbf{u} is velocity, ρ is the fluid density, μ is the dynamic viscosity, t is time, and p is pressure^{67, 68}.

Added references

[67] Athani A, et al. Two-phase non-Newtonian pulsatile blood flow simulations in a rigid and flexible patient-specific left coronary artery (LCA) exhibiting multi-stenosis. *Applied Sciences* 11, 11361 (2021).

[68] Zimny M, Kawlewska E, Hebda A, Wolański W, Ładziński P, Kaspera W. Wall shear stress gradient is independently associated with middle cerebral artery aneurysm development: a case-control CFD patient-specific study based on 77 patients. *BMC neurology* 21, 1-10 (2021).

Regarding the reviewer's concerns on the pressure gradient, the pressure gradient along with the longitudinal direction was observed, but a radial pressure gradient was not generated under our experimental condition, which might be due to the low input fluid velocity ($\sim 6 \text{ mm/s}$).

Considering the reviewer's comment, we performed the additional simulations by increasing the input velocities (**Figure S21**). The results showed that the pressure gradient in the

radial direction was generated over 20 mm/s velocity. Unfortunately, this study was highly focused on the effects of vascular geometry (vascular curve) on cancer metastasis. We totally admit the limitation caused by not considering for the various input fluid velocity, which should be overcome under future investigation.

We appreciate the reviewer for such critical concerns and also supplemented the related contents in the revised manuscript for clarity. We hope for the reviewer's understanding on the changes we have made. Additionally, for better understanding, we included an arrow defining the flow direction. The related content was supplemented in the revised manuscript, as follows.

Supplementary Figure S21

Figure S21. Generation of the pressure gradient at different fluid flow velocities. The radial pressure gradient is induced when the input fluid velocity exceeds ~20 mm s⁻¹ in bent tubes.

Page 8, Line 298

Considering these factors, our model was designed to investigate how various vascular curvatures influence cancer metastasis within human brain blood vessels. (...) Finally, this research demonstrated that the vascular geometries influence cancer metastasis. Despite the findings, a parametric study about the other hemodynamic factors, such as fluid velocity, flow rate, and blood pressure have not been fully investigated and the related in-depth study will be considered as a future work.

Revised Supplementary Figure S20

Page 6, Line 210

On the other hand, the fluid pressure gradient along with the longitudinal direction was observed, and the pressure change was slower on the extrados compared to the intrados (**Figure S20**). Additionally, no radial pressure gradient was indicated because of the low fluid velocity and low inertial force under the experimental condition (**Figure S21**)³⁵.

Reviewer’s comment 4:

Regarding the comparisons to native brain tissue, the authors further specified the protein composition of the BdECM which is greatly appreciated. This reviewer wonders how the protein composition in the BdECM compares to that of native brain and whether it may also contribute to the observed differences in mechanical/rheological properties. In particular, this reviewer wonders how Figures S2 B (Top 10 proteins), C (Top 5 basement membranes), D (Top 5 interstitial matrix), and E (Top 10 collagens) compares to that of native brain tissue, particular when looking at Figure S1 with a large difference in collagen content in the BdECM?

Response 4

We thank the reviewer for the interesting questions. In accordance with the comment, the methods of biochemical assay to analyze the glycosaminoglycan and collagen contents were added in the revised manuscript, as follows.

Page 10, Line 349

Biochemical analysis. The GAG contents in native brain tissue and BdECM were compared using a GAG quantification kit (Biocolor, UK) according to the manual. Briefly, to extract GAG components from 5 mg of the lyophilized samples, papain extraction reagent was treated for 3 h at 65 °C. After the sample tubes were centrifuged, the pellets were dissolved in deionized water. The test samples and reference standard aliquots were incubated with dye reagent for 30 min. After centrifugation of the tubes, dissociation reagent was added and the absorbance was measured at 656 nm using a microplate reader (Thermo Fisher Scientific, USA) ($n = 3$). The

collagen content was quantified using a total collagen assay kit (BioVision, USA) according to the developer's protocol. Briefly, 5 mg of native brain tissue and BdECM were homogenized using deionized water, and the samples and the collagen I standard were incubated with ~12 M hydrochloric acid for 3h at 120 °C. 10 min after Chloramine T reagent was added to each solution at room temperature, DMAB reagent was supplemented for 90 min at 60 °C. The absorbance of all samples were measured at 560 nm using a microplate reader (Thermo Fisher Scientific, USA).

Page 9, Line 361

Proteomics analysis. The protein composition was identified by applying the liquid chromatography-mass spectrometry (LC-MS/MS) analysis using Orbitrap Exploris 480 (Thermo Fisher Scientific, US), conducted by a biotechnology company (ebiogen, Korea). Briefly, after quantifying the protein content using the Bicinchoninic Acid Protein (BCA) assay, the proteins were digested using the filter aided sample preparation (FASP) method. The samples were sequentially treated with 500 mM tris (2-carboxyethyl) phosphine, 500 mM iodoacetamide, and 8M urea for 15 min each. The trypsinized proteins was desalted using desalting column, and the remained solutions were evaporated using speed vacuum concentrator. The LC-MS/MS data was qualified and quantified based on spectral library of Uniprot Sus scrofa protein sequence database, and the discovered proteins were selected based on precursor q-value (≤ 0.01) and protein group q-value (≤ 0.01). The relative composition of each protein was calculated as a percentage of the total content.

According to the previous studies, the protein composition governs the matrix organization and influences the stiffness of tissues. In addition, the protein composition and the rheological properties of decellularized extracellular matrix vary depending on the decellularization method; please refer to the following articles and a figure.

Reference 1: Black, Lauren D., et al. "Mechanical and failure properties of extracellular matrix sheets as a function of structural protein composition." Biophysical journal 94.5 (2008): 1916-1929,

Reference 2: Varma, Sameer, Joseph PRO Orgel, and Jay D. Schieber. "Nanomechanics of type I collagen." Biophysical journal 111.1 (2016): 50-56

A related figure: (next page; reference, Fernández-Pérez, Julia, and Mark Ahearne. "The impact of decellularization methods on extracellular matrix derived hydrogels." Scientific reports 9.1 (2019): 14933))

Figure 5. Rheology analysis of ECM-derived hydrogels: (A) Viscosity measurements at increasing shear rates; (B) storage modulus (G') and loss modulus (G''); * $p < 0.05$.

To verify the effects of compositional difference on the rheological properties, we measured the complex modulus of ECM hydrogels which were prepared following an established method (Fernández-Pérez, Julia, and Mark Ahearne. "The impact of decellularization methods on extracellular matrix derived hydrogels." *Scientific reports* 9.1 (2019): 14933), as shown in the following figure.

The complex modulus of BdECM increased according to the increase of the concentration. In addition, the modulus of 2% brain hydrogel was lower than that of 2% BdECM (native brain, 1287.00 ± 513.93 Pa; 2% BdECM Pa, 1635.31 ± 328.01 Pa; no significance). On the other hand, BdECM has higher content of collagen 1, compared to native brain (Figure S1). It raises a possibility that the compositional variance can influence mechanical property of the tissue-derived materials.

According to previous studies, native brain tissue and BdECM have different protein compositions (Reference: Simsa, Robin, et al. "Brain organoid formation on decellularized porcine brain ECM hydrogels." *PLoS One* 16.1 (2021): e0245685, and Cho, Ann-Na, et al.

"Microfluidic device with brain extracellular matrix promotes structural and functional maturation of human brain organoids." Nature communications 12.1 (2021): 4730). Considering the proteomics data of the previous studies, in particular, the native brain tissue has abundant intracellular or signaling molecules such as S100 proteins, but key basement membrane proteins (HSPG2 and type IV collagen) and interstitial matrix proteins (type I collagen and Fibrillin-1) were hardly detected. In contrast, the BdECM contains abundant content of collagen 1, which is consistent with our results (Figure S1 and S2), but it significantly loses fibrous proteins, such as elastin, fibronectin, and laminin, during decellularization process.

Considering all above, we totally admit the importance of the in-depth study to understand the effects of compositional variance on rheological/mechanical properties, which should be explored under future investigation. Unfortunately, however, this study focused on the link between vascular geometries and cancer dissemination patterns. Especially, in this study, we developed a hybrid BdECM bioink with the improved printability, and attempted to create blood vessels with various curvatures to investigate the underlying mechanisms.

We appreciate the reviewer for such interesting point, and the related contents were supplemented in the revised manuscript.

Page 8, Line 262

Unfortunately, restoring the properties of the original tissues is difficult during the decellularization and solubilization process because of missing components including intercellular junctions and macromolecular chains. It demands the technological advancement to maintain the ECM proteins and innate characteristics of the tissue and an in-depth study of the correlation between protein components and the mechanical and biological performances of tissue-derived material^{37, 41}. On the other hand, to compensate for this destruction, we utilized alginate and formulated a hybrid BdECM bioink to improve the fine-tuned shape fidelity of the constructs in a brain-tissue-specific microenvironment. As well as its biological and mechanical suitability for the cerebrovascular cells in our platform, its chemical reactivity that allowed the immediate ionic crosslinking between alginate and multivalent cations in CPF-127 for the elaboration of vessel curvatures with sharp angles. Utilizing the in-bath triaxial nozzle printing technique allowed the direct construction of MCCs with cellular complexity with four types of compartmentalized brain cells.

Page 3, Line 58

However, while the intracellular or signaling molecules were removed during decellularization process, the structural proteins residing at basement membrane (type IV collagens) and interstitial matrix (type I collagens) remained²⁷. Therefore, the amounts of collagen in BdECM were significantly higher than that in native tissue. The increased portion of fibrillary collagens can provide load-bearing mechanical properties to the material²⁸.

Added references

[27] Simsa R, et al. Brain organoid formation on decellularized porcine brain ECM hydrogels. PLoS One 16, e0245685 (2021).

[28] Varma S, Orgel JP, Schieber JD. Nanomechanics of type I collagen. Biophysical journal 111, 50-56 (2016).

[41] Ort C, Chen Y, Ghagre A, Ehrlicher A, Moraes C. Bioprintable, stiffness-tunable collagen-alginate microgels for increased throughput 3D cell culture studies. ACS Biomaterials Science & Engineering 7, 2814-2822 (2021).

Reviewer's comment 5:

For the functional blocking assay Figure S15, quantification of the number of cells adhered as well as the cell aspect ratio (to describe cell morphology) to the endothelium with/without blocking antibodies would present a more robust argument for the outcome of these experiments rather than observation or description alone. Moreover, the number of samples per group for these experiments should be specified.

Response 5

The quantitative data and the information on the number of samples per group were added in the revised manuscript.

Supplementary Figure S18C and S18D

Figure S18. Blocking cell adhesion molecules and cancer cell adhesion. A) Prior to introduction of cancer cells and the treatment of ICAM-1 function-blocking antibody (C⁻B⁻), the activated signal of the endothelial cells was not detected. After 24 h of treating cancer cells without the function-blocking antibody, some cancer cells adhered to the endothelium wall, and elongated their bodies. However, when the function of the adhesion molecules was blocked, a relatively

small number of the adhered cancer cells with round-shaped bodies was observed. B) In the case of VCAM-1, blocking the adhesion molecules suppressed the tumor metastatic progression. C) The number of adhered circulating tumor cells were quantitatively measured ($n = 3$). D) The aspect ratio of the adhered tumor cells was calculated ($n > 10$).

Reviewer's comment 6:

The authors observed a decrease in cell viability with the addition of alginate – however the addition of alginate into collagen-based gels can also affect cell morphology and spreading (Ort et al., ACS Biomater. Sci. Eng., 2021). Does this also occur with NPCs and brain endothelial cells with the addition of alginate?

Response 6

According to the reviewer's comment, the effects of alginate on the cellular morphology were investigated by comparing the morphological features of the cells in pure BdECM and hybrid BdECM bioink.

The features were initially different, but the morphological gap decreased as the encapsulated cells proliferated.

The results and the suggested reference were included in the revised manuscript, as follows.

Page 4, Line 99

Additionally, under the assumption that the presence of alginate can affect the cell morphology, pure BdECM (1.0B) and hybrid BdECM (1.0B0.5A) were compared³¹. Although the cells encapsulated in 1.0B0.5A exhibited relatively thin and long shapes with many pseudopodia on day 7 compared with those in 1.0B, the morphological gap decreased as the cells proliferated (**Figure S5**).

Supplementary Figure S5

Figure S5. Effects of alginate in hybrid brain-derived decellularized extracellular matrix (BdECM) bioink on the cellular morphology. Although A) human brain microvascular endothelial cells and B) neural progenitor cells in hybrid BdECM bioink (1.0B0.5A) have relatively thinner and longer shapes and more pseudopodia than those in pure BdECM bioink (1.0B) (day 7), the morphological gap has decreased as the cells proliferate (day 14).

Added reference

[31] Ort C, Chen Y, Ghagre A, Ehrlicher A, Moraes C. Bioprintable, stiffness-tunable collagen-alginate microgels for increased throughput 3D cell culture studies. *ACS Biomaterials Science & Engineering* 7, 2814-2822 (2021).

Reviewer's comment 7:

Regarding the measurement of native brain rheological and mechanical properties – the authors state that the differences may be due to cell density but should perhaps further specify the difference in cell density in terms of cells per unit volume in native tissue for clarification purposes.

Response 7

Considering the reviewer's concern, the cell densities of the 3D bioprinted construct and native brain tissue were addressed in the revised manuscript with the proper reference.

Because the cell density reported in previous works was expressed in units of cells/mg, conversion to the volume unit was needed. Therefore, the data on the weight and volume ratios of the measured brain tissue were added to the manuscript.

Page 4, Line 103

In addition, the cell-laden hybrid BdECM exhibited lower physical strength than native brain

tissue (**Figure S6**). This may have resulted from the difference in cell density between the two systems (experimental condition, $\sim 2 \times 10^6$ cells mL⁻¹; native brain, $> 5 \times 10^6$ cells mL⁻¹; weight/volume density of porcine brain, 1.05 ± 0.02 g mL⁻¹) (**Figure S7**).

Supplementary Figure S7

Figure S7. Density (weight/volume) of native porcine brain.

Added reference

[32] Kazu RS, Maldonado J, Mota B, Manger PR, Herculano-Houzel S. Cellular scaling rules for the brain of Artiodactyla include a highly folded cortex with few neurons. *Frontiers in neuroanatomy* 8, 128 (2014).

Reviewer's comment 8:

The authors mention use of CTCs throughout the manuscript and in the supplementary data but should specify the nature or source of the CTCs, and whether the CTC source or cell type changes between experiments.

Response 8

We thank the reviewer for pointing out this issue. Details regarding the source of the CTCs used in this study were presented in a supplementary table in the revised manuscript. Additionally, the type of the CTCs was not changed in the entire experiment.

Supplementary table 2

Cell type	Source	Donor
Human Lung Circulating Tumor cells	Human lung peripheral blood	Caucasian, Female, 62-year-old, metastatic lung cancer

Table S2. Cellular information of circulating tumor cells.

Reviewer's comment 9:

Figure S17 the colorimetric scale bar for pressure could be enlarged for better visibility.

Response 9

The change was made accordingly.

Supplementary Figure S20

Reviewer's comment 10:

Figure 2H(ii) on cell viability could be enlarged.

Response 10

The figure was edited accordingly.

Response to Reviewer 4

Article ID NCOMMS-23-03632B
Title 3D Bioprinted Multilayered Cerebrovascular Conduits to Study Cancer Extravasation Mechanism Related with Vascular Geometry
Authors Wonbin Park, Jae-Seong Lee, Ge Gao, Byoung Soo Kim, Dong-Woo Cho

Summary of response:

We appreciate the constructive comments of the reviewers. We carefully read the valuable comments from the reviewers, and revised the manuscript based on these comments. Please check our specific responses to the reviewer's comments and revisions made in the manuscript as detailed below. The answers to the comments are highlighted in blue, and the modifications in the revised manuscript are indicated with red font.

Reviewer's comment 1:

The maturation of MCC cells still need more robust and quantitative analysis. The images provided are still not sufficient to support the big claim on "fully mature multilayered cerebrovascular conduits (MCCs)". Authors must provide PCR / micro-array quantitatively to demonstrate significant upregulation of specific maturation markers for each cell type and compare either with control cells in their study, or some reported numbers from the literature.

Response 1:

We appreciate the reviewer's comment. To verify the formation of mature multi-layered cerebrovascular conduits, qRT-PCR analysis was conducted.

The results indicated the significant upregulation of maturation markers (endothelial tight junction markers, ZO-1 and occludin; pericyte markers, α -SMA and PDGFR- β ; an astrocytic marker, GFAP; neuronal markers, TUJ1 and MAP2) as shown in the supplemented **Figure S11**.

Considering the reviewer's comment, related content was added to the revised manuscript.

Page 5, Line 151

In addition, the expression of the cell maturation markers increased (**Figure S11**).

Supplementary figure S11

Figure S11. Quantitative reverse-transcription polymerase chain reaction results demonstrating the cerebrovascular tissue formation. The expression of maturation markers (endothelial tight junction markers, ZO-1 and occludin; pericyte markers, α-SMA and PDGFR-β; an astrocytic marker, GFAP; neuronal markers, TuJ1 and MAP2) increases over time (**, $p \leq 0.01$; *, $p \leq 0.05$; $n = 3$).

REVIEWERS' COMMENTS

Reviewer #3 (Remarks to the Author):

The significant consideration and efforts undertaken by the authors to address the comments and clarifications made in the previous revisions are greatly appreciated. Moreover, this reviewer particularly appreciates the authors' response to the questions posed in the last revision. As such, there are no further comments or questions to address from this reviewer.

Reviewer #4 (Remarks to the Author):

I'd like to thank the authoring team for their diligent work in responding to all the reviewers' comments. As a result, the manuscript has been significantly improved and merits publication in Nature Comm.

Response to Reviewer 3

Article ID NCOMMS-23-03632B

Title 3D Bioprinted Multilayered Cerebrovascular Conduits to Study Cancer
Extravasation Mechanism Related with Vascular Geometry

Authors Wonbin Park, Jae-Seong Lee, Ge Gao, Byoung Soo Kim, Dong-Woo Cho

Reviewer's comment 1:

The significant consideration and efforts undertaken by the authors to address the comments and clarifications made in the previous revisions are greatly appreciated. Moreover, this reviewer particularly appreciates the authors' response to the questions posed in the last revision. As such, there are no further comments or questions to address from this reviewer.

Response 1:

We appreciate the reviewer's supportive comment.

Response to Reviewer 4

Article ID NCOMMS-23-03632B

Title 3D Bioprinted Multilayered Cerebrovascular Conduits to Study Cancer
Extravasation Mechanism Related with Vascular Geometry

Authors Wonbin Park, Jae-Seong Lee, Ge Gao, Byoung Soo Kim, Dong-Woo Cho

Reviewer's comment 1:

I'd like to thank the authoring team for their diligent work in responding to all the reviewers' comments. As a result, the manuscript has been significantly improved and merits publication in Nature Comm.

Response 1:

We appreciate the reviewer's supportive comment.